

# Deposition, recycling and archival of nitrate stable isotopes between the air-snow interface: comparison between Dronning Maud Land and Dome C, Antarctica

V. Holly L. Winton[1], Alison Ming[1], Nicolas Caillon[2], Lisa Hauge[1], Anna E. Jones[1], Joel Savarino[2], Xin Yang[1], Markus M. Frey[1]

[1]British Antarctic Survey, Cambridge, CB3 0ET, UK
[2]University of Grenoble Alpes, CNRS, IRD, Grenoble INP, IGE, F-38000 Grenoble, France

*Correspondence to*: V. Holly L Winton (vicwin@bas.ac.uk)

## Abstract

The nitrate ($NO_3^-$) isotopic composition $\delta^{15}N\text{-}NO_3^-$ of polar ice cores has the potential to provide constraints on past ultraviolet (UV) radiation and thereby total column ozone (TCO), in addition to the oxidising capacity of the ancient atmosphere. However, understanding the transfer of reactive nitrogen at the air-snow interface in Polar Regions is paramount for the interpretation of ice core records of $\delta^{15}N\text{-}NO_3^-$ and $NO_3^-$ mass concentrations. As $NO_3^-$ undergoes a number of post-depositional processes before it is archived in ice cores, site-specific observations of $\delta^{15}N\text{-}NO_3^-$ and air-snow transfer modelling are necessary in order to understand and quantify the complex photochemical processes at play. As part of the Isotopic Constraints on Past Ozone Layer Thickness in Polar Ice (ISOL-ICE) project, we report new measurements of $NO_3^-$ concentration and $\delta^{15}N\text{-}NO_3^-$ in the atmosphere, skin layer (operationally defined as the top 5 mm of the snow pack), and snow pit depth profiles at Kohnen Station, Dronning Maud Land (DML), Antarctica. We compare the results to previous studies and new data, presented here, from Dome C, East Antarctic Plateau. Additionally, we apply the conceptual one-dimensional model of TRansfer of Atmospheric Nitrate Stable Isotopes To the Snow (TRANSITS) to assess the impact of photochemical processes that drive the archival of $\delta^{15}N\text{-}NO_3^-$ and $NO_3^-$ in the snow pack. We find clear evidence of $NO_3^-$ photolysis at DML, and confirmation of our hypothesis that UV-photolysis is driving $NO_3^-$ recycling at DML. Firstly, strong denitrification of the snow pack is observed through the $\delta^{15}N\text{-}NO_3^-$ signature which evolves from the enriched snow pack (-3 to 100 ‰), to the skin layer (-20 to 3 ‰), to the depleted atmosphere (-50 to -20 ‰) corresponding to mass loss of $NO_3^-$ from the snow pack. Secondly, constrained by field measurements of snow accumulation rate, light attenuation (e-folding depth) and atmospheric $NO_3^-$ mass concentrations, the TRANSITS model is able to reproduce our $\delta^{15}N\text{-}NO_3^-$ observations in depth profiles. We find that $NO_3^-$ is recycled three times before it is archived (i.e., below the photic zone) in the snow pack below 15 cm and within 0.75 years. Archived $\delta^{15}N\text{-}NO_3^-$ and $NO_3^-$ concentration values are 50 ‰ and 60 ng g$^{-1}$ at DML. $NO_3^-$ photolysis is weaker at DML than at Dome C, due primarily to the higher DML snow accumulation rate; this results in a more depleted $\delta^{15}N\text{-}NO_3^-$ signature at DML than at Dome C. Even at a relatively low snow accumulation rate of 6 cm yr$^{-1}$ (water equivalent; w.e.), the accumulation rate at DML is great





enough to preserve the seasonal cycle of $NO_3^-$ concentration and $\delta^{15}N$-$NO_3^-$, in contrast to Dome C where the profiles are smoothed due to stronger photochemistry. TRANSITS sensitivity analysis of $\delta^{15}N$-$NO_3^-$ at DML highlights that the dominant factors controlling the archived $\delta^{15}N$-$NO_3^-$ signature are the snow accumulation rate and e-folding depth, with a smaller role from changes in the snowfall timing and TOC. Here we set the framework for the interpretation of a 1000-year ice core record

of $\delta^{15}N$-$NO_3^-$ from DML. Ice core $\delta^{15}N$-$NO_3^-$ records at DML will be less sensitive to changes in UV than at Dome C, however the higher snow accumulation rate and more accurate dating at DML allows for higher resolution $\delta^{15}N$-$NO_3^-$ records.

# 1   Introduction

Nitrate ($NO_3^-$) is a naturally occurring ion formed by the oxidation of nitrogen, and plays a major role in the global nitrogen cycle. It is one of the most abundant ions in Antarctic snow and is commonly measured in ice cores (e.g. Wolff, 1995). Nitrate

in polar ice provides constraints on past solar activity (Traversi et al., 2012), $NO_3^-$ sources and the oxidative capacity of the atmosphere (Geng et al., 2017;Mulvaney and Wolff, 1993;Hastings et al., 2009;Hastings et al., 2004;McCabe et al., 2007;Savarino et al., 2007;Morin et al., 2008). However, $NO_3^-$ is a non-conservative ion in snow, and due to post-depositional processes (e.g. Mulvaney et al., 1998;Zatko et al., 2016), the interpretation of $NO_3^-$ concentration records from ice core records is challenging (Erbland et al., 2015). The recent development of the analysis of nitrogen isotopic composition of $NO_3^-$ ($\delta^{15}N$-

$NO_3^-$) in snow, ice and aerosol provides a powerful means to understand the sources and processes involved in $NO_3^-$ post-depositional processes, i.e., $NO_3^-$ recycling at the interface between air and snow.

Primary sources of reactive nitrogen species to the Antarctic lower atmosphere and snow pack include the sedimentation of polar stratospheric clouds (PSC) in late winter (Savarino et al., 2007) and to a minor extent advection of oceanic methyl nitrate ($CH_3NO_3$) and peroxyacyl nitrates (PAN) (Jacobi et al., 2000;Jones et al., 1999;Beyersdorf et al., 2010). In the stratosphere,

$NO_3^-$ is produced through the stratospheric oxidation of nitrous oxide ($N_2O$) from extra-terrestrial fluxes of energetic particles and solar radiation, whereas in the troposphere lightning and biomass burning provide background tropospheric reactive nitrogen species to the snow pack (Savarino et al., 2007;Wolff, 1995;Wagenbach et al., 1998). A local secondary source of reactive nitrogen (nitrous acid (HONO), nitrogen oxides ($NO_x$)) originates from post-depositional processes driven by sunlight leading to re-emission from the snow pack and subsequent deposition (Savarino et al., 2007;Frey et al., 2009).

Local nitrogen dioxide ($NO_2$) emissions in Polar Regions are produced from $NO_3^-$ photolysis in the snow pack under sunlit conditions (Jones et al., 2001;Honrath et al., 1999;Oncley et al., 2004). Nitrate photolysis occurs at wavelengths ($\lambda$) = 290-345 nm with a maximum at 320 nm. Photolysis rate J depends on the adsorption cross section of $NO_3^-$, the quantum yield and actinic flux within the snow pack. Photochemical production of $NO_2$ is dependent on the $NO_3^-$ concentration in the snow pack, the snow pack properties, and the intensity of solar radiation within the snow pack. The latter is sensitive to solar zenith angle

and snow optical properties i.e. scattering and adsorption coefficients, which depends on snow density and morphology, and the light absorbing impurity content (France et al., 2011;Erbland et al., 2015;Zatko et al., 2013). Recently, Zatko et al. (2016) found that the range of modelled $NO_x$ fluxes from the snow pack to the overlaying air are similar in both Polar Regions due to



the opposing effects of higher concentrations of both photolabile $NO_3^-$ and light absorbing impurities (e.g. dust and black carbon) in Antarctica and Greenland respectively. At Concordia Station on Dome C in East Antarctica, the light penetration depth (e-folding depth) is ~10 cm for wind pack layers and ~20 cm for hoar layers (France et al., 2011). Based on the propagation of light into the snow pack, the snow pack can be divided into three layers. The first layer is known as the skin layer (a few mm thick) where direct solar radiation is converted into diffuse radiation. The second layer is called the active photic zone (below the skin layer layer), where solar radiation is effectively diffuse and the intensity of the radiation decays exponentially (Warren, 1982). The third layer is called the archived zone (below the active photic zone), where no photochemistry occurs.

Previous research has focused predominantly on the high elevation polar plateau (Dome C). Here, the exponential decay of $NO_3^-$ mass concentrations in the snow pack and thus post-depositional processing of $NO_3^-$ were attributed to either evaporation or ultra-violet (UV)-photolysis (Röthlisberger et al., 2000;Röthlisberger et al., 2002). The open debate of which post-depositional process controlled $NO_3^-$ mass concentrations in the snow pack led to the use of a new isotopic tool, the stable isotopic composition of nitrate ($\delta^{15}N-NO_3^-$) (Blunier et al., 2005). More recently, theoretical (Frey et al., 2009), laboratory (Meusinger et al., 2014;Erbland et al., 2013;Erbland et al., 2015;Shi et al., 2019;Berhanu et al., 2014), and field (Erbland et al., 2013;Frey et al., 2009) evidence show that $NO_3^-$ mass loss from the surface snow to the overlying atmosphere and its associated isotopic fractionation is driven by photolysis. The physical release or evaporation of $NO_3^-$ is negligible (Erbland et al., 2013;Shi et al., 2019).

At Dome C, the large redistribution and net mass loss of $NO_3^-$ below the skin layer and the simultaneous isotopic fractionation of $\delta^{15}N-NO_3^-$ in the snow pack indicate that post-depositional processes significantly modify the original $NO_3^-$ concentration and $\delta^{15}N-NO_3^-$ composition (Frey et al., 2009). Skin layer observations of $\delta^{15}N-NO_3^-$ in the surface snow at Dome C show strong enrichment compared to the atmospheric $\delta^{15}N-NO_3^-$ signature. Furthermore, snow pit profiles show an exponential decrease of $NO_3^-$ concentration and an enrichment in the $\delta^{15}N-NO_3^-$ composition with depth (Erbland et al., 2013). Once $NO_x$ is produced by $NO_3^-$ photolysis, it is expected to have a lifetime in the polar troposphere of <1 day before it is oxidised to nitric acid ($HNO_3$) at Dome C and South Pole (Davis et al., 2004b), and can then be redeposited to the skin layer (e.g. Mulvaney et al., 1998).

This research at Dome C laid the foundation for Erbland et al. (2015) to derive a conceptual model of UV-photolysis induced post-depositional processes of $NO_3^-$ at the air-snow interface. "Nitrate recycling" is the combination of $NO_x$ production from $NO_3^-$ photolysis in snow, the subsequent atmospheric processing and oxidation of $NO_x$ to form atmospheric $HNO_3$, the deposition (dry and/or wet) of a fraction of the $HNO_3$, and the export of another fraction. In $NO_3^-$ recycling, the skin layer is an active component of the atmosphere. This recycling can occur multiple times before $NO_3^-$ is eventually archived below the active photic zone in ice cores (Davis et al., 2008;Erbland et al., 2015;Zatko et al., 2016;Sofen et al., 2014). We refer to atmospheric $NO_3^-$ as the combination (i.e., total) of $HNO_3$ (gas phase) and particulate $NO_3^-$.

Year round measurements of $NO_3^-$ mass concentrations and $\delta^{15}N-NO_3^-$ in the skin layer and atmosphere at Dome C have provided insights into the annual $NO_3^-$ cycle in Antarctica (Fig. 1) (Erbland et al., 2013). In the early winter, the stratosphere





undergoes denitrification via formation of PSC. As PSC sediment slowly, there is a delay between the maximum stratospheric $NO_3^-$ concentration and the maximum $NO_3^-$ concentration deposited in the skin layer in late winter (Mulvaney and Wolff, 1993;Savarino et al., 2007). In spring, surface UV increases and initiates photolysis-driven post-depositional processes, which

redistribute $NO_3^-$ between the snow pack and overlaying air throughout the sunlit summer season. This results in the $\delta^{15}N\text{-}NO_3^-$ isotopic enrichment of the $NO_3^-$ skin layer reservoir, and maximum atmospheric $NO_3^-$ mass concentrations in October-November. In summer, $NO_3^-$ resembles a strongly asymmetric distribution within the atmosphere-snow column with the bulk residing in the skin layer and only a small fraction in the atmospheric column above.

Over longer time scales, UV-driven post-depositional processing of $NO_3^-$ is also driven by changes in the degree of post-

depositional loss of $NO_3^-$ with greater $NO_3^-$ loss during the glacial period relative to the Holocene. The observed glacial-interglacial difference in post-depositional processing of $NO_3^-$ is dominated by variations in snow accumulation rate (Geng et al., 2015).

Nitrate is not preserved in the snow pack at sites with very low snow accumulation rates (i.e., Dome C: 2.5-3 cm yr$^{-1}$) because snow layers remain close to the surface and in contact with the overlaying atmosphere for a relatively long time enhancing the

effect of post-depositional processes. At sites with low snow accumulation rates, the source signature of $\delta^{15}N\text{-}NO_3^-$ is erased by post-depositional process. Therefore, photolysis induced $NO_3^-$ loss and $\delta^{15}N\text{-}NO_3^-$ fractionation is dependent on snow accumulation. Three distinct transects from coastal Antarctica to the East Antarctic Plateau show that $NO_3^-$ fractionation is strongest with decreasing snow accumulation (Shi et al., 2018;Erbland et al., 2013;Noro et al., 2018). Skin layer $NO_3^-$ mass concentrations are significantly higher at low snow accumulation sites, for example ~160 ng g$^{-1}$ (winter) to 1400 ng g$^{-1}$

(summer) at Dome C compared to 50 ng g$^{-1}$ (winter) to 300 ng g$^{-1}$ (summer) at Dumont d'Urville (DDU) on the Antarctic coast. Furthermore, the strong inverse linear relationship between $NO_3^-$ concentration and accumulation rate was revealed in a composite of seven ice cores across Dronning Maud Land (DML) (Pasteris et al., 2014).

Yet, $NO_3^-$ photolysis leaves its own process-specific imprint in the snow pack (Shi et al., 2019;Erbland et al., 2015;Erbland et al., 2013), which opens up the possibility to use $\delta^{15}N\text{-}NO_3^-$ to infer past surface-UV variability (Frey et al., 2009). However,

$NO_3^-$ photolysis rates in snow depend on a number of site-specific factors as does the degree of photolytic isotopic fractionation of $NO_3^-$ eventually preserved in ice cores (Erbland et al., 2013;Berhanu et al., 2014). These factors need to be quantitatively understood at a given ice core site to enable quantitative interpretation of ice core records. Here, we carry out a comprehensive study of the air-snow transfer of $NO_3^-$ at Kohnen Station in DML, East Antarctica through $\delta^{15}N\text{-}NO_3^-$ measurements in the atmosphere, skin layer and snow pits, and compare the observations to Dome C. Due to the previous research outlined above,

we assume that the photolysis is the dominant driver of $NO_3^-$ post-depositional processes, and later assess the validity of this this assumption (section 4.3). We apply the Transfer of Atmospheric Nitrate Stable Isotopes To the Snow (TRANSITS) model (Erbland et al., 2015) to i) understand how $NO_3^-$ mass concentrations and $\delta^{15}N\text{-}NO_3^-$ are archived in deeper snow and ice layers, and ii) investigate the sensitivity of changes in the past snow accumulation rate, snowfall timing, e-folding depth, and TCO on the $\delta^{15}N\text{-}NO_3^-$ signature. In order to interpret this novel UV proxy, it is paramount to understand the air-snow transfer processes

specific to an ice core site, and how $\delta^{15}N\text{-}NO_3^-$ is archived in the deeper snow and ice layers (Geng et al., 2015;Morin et al.,



2009;Erbland et al., 2015). Within the framework of the Isotopic Constraints on Past Ozone Layer Thickness in Polar Ice (ISOL-ICE) project, this study provides a basis for the interpretation of $\delta^{15}$N-NO$_3^-$ from a 1000-year ice core recovered in 2016/17 at Kohnen Station.

## 2    Methods

The ISOL-ICE project aims to understand natural causes of past TCO variability by i) an air-snow exchange study to enable the interpretation of ice core records of NO$_3^-$ and $\delta^{15}$N-NO$_3^-$, ii) reconstructing a 1000-year record of UV using a new ice core proxy based on $\delta^{15}$N-NO$_3^-$ (Ming et al., in prep; Winton et al., 2019a), and iii) numerical modelling of the natural causes of TCO variability. In the air snow-transfer study presented here, we report new atmospheric, skin layer and snow pit NO$_3^-$ and $\delta^{15}$N-NO$_3^-$ observations from DML, and compare them to new and published (Erbland et al., 2015;Erbland et al., 2013;Frey et

al., 2009) observations from Dome C. Published data from Dome C comprises year round atmospheric and skin layer measurements from 2009-2010 (Erbland et al., 2013), and multiple snow pit profiles (Erbland et al., 2013;Frey et al., 2009). We present a new extended time series at Dome C of year round atmospheric and skin layer NO$_3^-$ and $\delta^{15}$N-NO$_3^-$ from 2011-2015.

### 2.1   Study sites

The ISOL-ICE campaign was carried out at the summer only, continental Kohnen Station where the deep European Project for Ice Coring in Antarctica (EPICA) Dronning Maud Land (EDML; 75°00' S, 0°04' E; 2982 m a.s.l.; https://www.awi.de/en/expedition/stations/kohnen-station.html) ice core was recovered in 2001-2006 to a depth of ~2800 m (Wilhelms et al., 2017). As part of the ISOL-ICE campaign, a new ice core (ISOL-ICE;(Winton et al., 2019a)) was drilled 1 km from the EDML borehole (Fig. 2). In addition, the ISOL-ICE air-snow transfer study site was located ~200 m from the

EDML ice core site (Fig. 2). Here we compare two ice core drilling sites in Antarctica: Kohnen Station (referred to as DML henceforth) and EPICA Dome C (75°05'59" S, 123°19'56" E) (Fig. 2). Both sites are similar in terms of the latitude and therefore in terms of radiative forcing at the top of the atmosphere (Table 1). Satellite images of TCO over Antarctica show that the lowest annual TCO values are centred over the South Pole region encompassing DML and usually Dome C although the spatial variability is significant from year to year (https://ozonewatch.gsfc.nasa.gov/). The sites are different in terms of

their location with respect to moisture source, elevation and precipitation regime. The DML site is situated ~550 km from the ice shelf edge, is subject to cyclonic activity and receives ~80 % of its precipitation from frontal clouds (Reijmer and Oerlemans, 2002). While Dome C is more remote (~1100 km from the coast) and diamond dust is the dominant form of precipitation. The annual snow accumulation rate also differs between the sites; while both sites have exceptionally low accumulation compared to the coast, DML (annual mean: 6 cm yr$^{-1}$ (water equivalent; w.e.)) receives more than double that of

Dome C (annual mean: 2.5 cm yr$^{-1}$ (w.e.)) (Le Meur et al., 2018;Hofstede et al., 2004;Sommer et al., 2000). Throughout the study we refer to our sampling site as "DML".



## 2.2 Snow and aerosol sampling

Daily skin layer samples (which we operationally define as the top 5 mm of the snow pack following Erbland et al. (2013)) were collected from the DML site (Fig. 2) in January 2017 during the ISOL-ICE ice core drilling and atmospheric monitoring
campaign. To prevent contamination from the nearby Kohnen Station, snow samples were collected from the "flux site" within the station's designated clean air sector (defined as 45° from both ends of the station building) located ~1 km from the station (Fig. 2). The skin layer samples were was collected in polyethylene bags (Whirl-pak®) using a stainless steel trowel. A total of 45 skin layer samples were collected between 31 December 2016 and 29 January 2017 from a designated sampling site each day during the campaign (75°00.184' S, 000°04.527' E; Fig. 2). To determine the spatial variability of $NO_3^-$ in the skin layer
at the flux site, an additional five skin layer samples were collected in a ~2500 $m^2$ area of the flux site (75°00.161' S - 000°04.441' E, 75°00.175' S - 000°04.518' E; Fig. 2c).

Adjacent to the skin layer samples, snow was sampled from a 1.6 m snow pit at the flux site (snow pit B; Fig. 2c) and a 2 m snow pit at the "ice core" site (snow pit A; Fig. 2b). Two parallel profiles were sampled, i) for major ion mass concentrations (including $NO_3^-$) collected in pre-washed 50 mL Corning® centrifuge tubes at 3 cm resolution by inserting the tube directly
into the snow face, and ii) for stable $NO_3^-$ isotope analysis collected in Whirl-pak® bags at 2 cm resolution using a custom-made stainless steel tool. Exposure blanks (following the same method as the samples by opening the tube/ Whirl-pak® bag at the field site but not filling the sample container with snow) were also collected for both types of samples. Snow density and temperature were measured every 3 cm, and a visual log of snow pit stratigraphy was recorded.

Daily aerosol filters were collected using high-volume aerosol samplers custom-built at the Institute of Environmental
Geosciences (IGE), University of Grenoble-Alpes, France described previously (Frey et al., 2009; Erbland et al., 2013). The high-volume sampler collected atmospheric aerosol on glass fibre filters (Whatman GF/A filter sheets; $20.3 \times 25.4$ cm) at an average flow rate of 1.2 $m^3$ $min^{-1}$ at standard temperature and pressure (STP; temperature: 273.15 K; pressure: 1 bar) to determine the concentration and isotopic composition of atmospheric $NO_3^-$. It is assumed that the atmospheric $NO_3^-$ collected on glass fibre filters represents the sum of atmospheric particulate $NO_3^-$ and $HNO_3$ (gas phase). The bulk of $HNO_3$ present in
the gas phase is most likely adsorbed to aerosols on the filter, as described previously (Frey et al., 2009).

The high-volume sampler was located 1 m above the snow surface at the flux site at the DML site (Fig. 2c), where a total of 35 aerosol filters were sampled daily between 3 and 27 January 2017. In addition, we coordinated an intensive 4-hour sampling campaign in phase with Dome C, East Antarctica (Fig. 2) between 21 and 23 January 2017. At Dome C, high-volume sampler is located on the roof of the atmospheric shelter (6 m above the snow surface), where a total of 12 samples were collected. At
DML, loading and changing of aerosol collection substrates was carried out in a designated clean area. Aerosol laden filters were transferred into individual double zip-lock plastic bags immediately after collection and stored frozen until analysis at the British Antarctic Survey (BAS; major ions) and IGE ($NO_3^-$ isotopic composition). For the atmospheric $NO_3^-$ work, three types of filter blanks were carried out; i) laboratory filter blanks (n = 3; Whatman GF/A filters that underwent the laboratory procedures without going into the field), ii) procedural filter blanks (DML: n = 4; Dome C: n = 1; filters that had been treated



as for normal samples but which were not otherwise used; once a week, during daily filter change-over, a procedural blank
       filter was mounted in the aerosol collector for 5 min without the collector pump in operation – this type of filter provides an
       indication of the operational blank associated with the sampling procedure), and iii) 24 h exposure filter blanks sampled at the
       beginning and end of the field campaign (DML: n = 2; Dome C: n = 1; filters treated like a procedural blank but left in the
       collector for 24 h without switching the collector on). All samples were kept frozen below -20 ℃ during storage and transport
prior to analysis.

       In addition, skin layer and aerosol samples have been sampled continuously at Dome C over the period 2009-2015 following
       Erbland et al. (2013);Frey et al. (2009). The sampling resolution for skin layer is every 2-4 days, and weekly for aerosol
       samples. Data from 2009-2010 have previously been published by Erbland et al. (2013), and we report the 2011-2015 data
       here (Fig. 1).

**2.3 Major ion mass concentrations in snow and aerosol**

       Aerosol $NO_3^-$ and other major ions were extracted in 40 mL of ultra-pure water (resistivity of 18.2 MΩ; Milli-Q water) by
       centrifugation using Millipore Centricon® Plus-70 Filter Units (10 kD filters) in a class-100 clean room at the BAS. Major ion
       mass concentrations in DML snow samples were determined in an aliquot of melted snow from skin layer and snow pit
       samples, and aerosol extracts by suppressed ion chromatography (IC) using a Dionex™ ICS-4000 Integrated Capillary HPIC™
System ion chromatograph. A suite of anions, including $NO_3^-$, chloride ($Cl^-$), methanesulfonic acid (MSA) and sulphate ($SO_4^{2-}$
       ), were determined using an AS11-HC column and a CES 500 suppressor. Cations, including sodium ($Na^+$), were determined
       using a CS12A column and a CES 500 suppressor. During the course of the sample sequence, instrumental blank solutions
       and certified reference materials (CRM; ERM-CA616 groundwater standard and ERM-CA408 simulated rainwater standard;
       Sigma-Aldrich) were measured regularly for quality control and yielded an accuracy of 97 % for $NO_3^-$. Nitrate mass
concentrations in Dome C samples were determined by colorimetry at IGE following the procedure described in Frey et al.
       (2009). Blank concentrations for exposure blank, procedural blank and laboratory blank and detection limits are reported in
       Table S1. The non-sea-salt sulphate (nss-$SO_4^{2-}$) fraction of $SO_4^{2-}$ was obtained by subtracting the contribution of sea-salt-
       derived $SO_4^{2-}$ from the measured $SO_4^{2-}$ mass concentrations (nss-$SO_4^{2-}$ = $SO_4^{2-}$ - 0.252 × $Na^+$, where $Na^+$ and $SO_4^{2-}$ are the
       measured concentrations in snow pit samples and 0.252 is the $SO_4^{2-}$/ $Na^+$ ratio in bulk seawater.

**2.4 Nitrate isotopic composition in snow and aerosol**

       Samples were shipped frozen to IGE where the $NO_3^-$ isotope analysis was performed. The denitrifier method was used to
       determine the stable $NO_3^-$ isotopic composition in samples at IGE following Morin et al. (2008). Briefly, samples were pre-
       concentrated due to the low $NO_3^-$ mass concentrations found in the atmosphere and snow over Antarctica. To obtain 100 nmol
       of $NO_3^-$ required for $NO_3^-$ isotope analysis, the meltwater of snow samples and aerosol extracts were sorbed onto 0.3 mL of
anion exchange resin (AG1-X8 chloride form; Bio-Rad) and eluted with 5 x 2 mL of 1 M NaCl (high purity grade 99.0 %;
       American Chemical Society (ACS grade); AppliChem Panreac) following Silva et al. (2000). Recovery tests yielded 100 %





recovery of NO$_3^-$ (Frey et al., 2009;Erbland et al., 2013). Once pre-concentrated, NO$_3^-$ is converted to N$_2$O gas by denitrifying bacteria, *Pseudomonas aureofaciens*. The N$_2$O is split into O$_2$ and N$_2$ on a gold furnace heated to 900 °C followed by gas chromatographic separation and injection into the isotope ratio mass spectrometer (IRMS) for duel O and N analysis using a

Thermo Finnigan™ MAT 253 IRMS equipped with a GasBench II™ and coupled to an in-house-built NO$_3^-$ interface (Morin et al., 2009).

Certified reference materials (IAEA USGS-32, USGS-34 and USGS-35; Böhlke et al., 1993; Böhlke et al., 2003) were prepared (matrix match 1 M NaCl in identical water isotopic composition as samples; ACS grade) and subject to the same analytical procedures as snow and aerosol samples. The nitrogen isotopic ratio was referenced against N$_2$-Air (Mariotti, 1983).

We report $^{15}$N/$^{14}$N of NO$_3^-$ ($\delta^{15}$N-NO$_3^-$) as δ-values following Eq. (1).

$$\delta^{15}\text{N-NO}_3^- = (^{15}\text{N}/^{14}\text{N}_{sample} / {}^{15}\text{N}/^{14}\text{N}_{standard} - 1) \qquad (1)$$

For each batch of 60 samples, the overall accuracy of the method is estimated as the reduced standard deviation of the residuals from the linear regression between the measured reference materials (n = 16) and their expected values. For the snow (n = 118) and aerosol samples (n = 35), the average uncertainty values obtained for $\delta^{15}$N was 0.5 ‰ for both datasets.

### 2.5 Light attenuation through the snow pack (e-folding depth)

Measurements of light attenuation through the snow pack were made at the two snow pit sites during the ISOL-ICE campaign following a similar approach of previous studies (France and King, 2012;France et al., 2011). Vertical profiles of down-welling irradiance in the top 0.4 m of the snow pack were measured using a high-resolution spectrometer (HR4000; Ocean Optics) covering a spectral range of 280 to 710 nm. To do this, a fiber optic probe attached to the spectrometer and equipped with a

cosine corrector with spectralon diffusing material (CC-3-UV-S; Ocean Optics) was inserted into the snow to make measurements at approximately 0.03 m depth intervals. The fiber optic probe was either inserted horizontally into pre-cored holes, at least 0.5 m in length to prevent stray light, into the side wall of a previously dug snow pit or pushed gradually into the undisturbed snow pack starting at the surface at a 45° angle, which was maintained by a metal frame. Most measurements with integration time ranging between 30 and 200 ms were carried out at noon to minimise changing sky conditions, and each

vertical snow profile was completed within 0.5 hr. The spectrometer was calibrated against a known reference spectrum from a Mercury Argon calibration source (HG-1; Ocean Optics), dark spectra were recorded in the field by capping the fibre optic probe and spectral irradiance was then recorded at depth relative to that measured right above the snow surface.

The e-folding depth was then calculated according to the Beer-Bouguer Lambert law. Stratigraphy of the snow pack recorded at each site showed presence of several thin (10 mm) wind crust layers over the top 0.4 m of snow pack. However, calculating

e-folding depths for each layer in between wind crusts yielded inconclusive results. Therefore, reported e-folding depths (Fig. S1, Table S2) are based on complete profiles integrating potential effects from wind crust layers. Resulting e-folding depths relevant for the photolysis of NO$_3^-$ (Table S2) show significant standard deviations, and also considerable variability (0.9-4.0 cm) between profiles, which reflect both systematic experimental errors as well as spatial variability of snow optical properties. They are lower than at Dome C but similar to previous model estimates for South Pole (France et al., 2011;Wolff et al., 2002).





The origin of the reduced e-folding depth relative to Dome C is not known but is likely due to greater HUmic-LIke Substances (HULIS) content or different snow morphology (Libois et al., 2013;Zatko et al., 2013). We use e-folding depths observed in this study at DML and those reported previously at Dome C as guidance for our model sensitivity study to quantify the impact of the variability of e-folding depth on archived $\delta^{15}N\text{-}NO_3^-$ in snow.

## 2.6 Nitrate photolysis rate coefficient

Hemispheric or $2\pi$ spectral actinic flux from 270 to 700 nm was measured at 2.1 m above the snow surface using an actinic flux spectroradiometer (Meteorologieconsult GmbH; Hofzumahaus et al. (2004). $2\pi$ $NO_3^-$ photolysis rate coefficients J($NO_3^-$) were then computed using the $NO_3^-$ absorption cross section and quantum yield on ice estimated for -30 ºC from Chu and Anastasio (2003). The mean $2\pi$ J-$NO_3^-$ value at DML during January 2017 was 1.02 x $10^{-8}$ $s^{-1}$, and 0.98 x $10^{-8}$ $s^{-1}$ during the 1 to 14 January 2017 period. The observed $2\pi$ J($NO_3^-$) at DML was a factor of three lower than Dome C (2.97 x $10^{-8}$ $s^{-1}$; 1 to 14

January 2012) which was previously measured using the same instrument make and model, and at the same latitude (Kukui et al., 2013). Only ~5 % of the apparent inter-site difference can be attributed to TCO being ~25 DU larger at DML (306 DU) than at Dome C (287 DU) during the comparison period. The remainder was possibly due to greater cloudiness at DML and differences in calibration. In this study, the observed $2\pi$ J($NO_3^-$) is used to estimate the snow emission flux of $NO_2$.

## 2.7 Air-snow transfer modelling

In order to evaluate the driving parameters of isotope air-snow transfer at DML we used the TRANSITS model (Erbland et al., 2015) to simulate snow profiles of $NO_3^-$ concentration and $\delta^{15}N\text{-}NO_3^-$ and compare them to our observations. TRANSITS is a conceptual multi-layer 1D model which aims to represent $NO_3^-$ recycling at the air-snow interface including processes relevant for $NO_3^-$ snow photochemistry (UV-photolysis of $NO_3^-$, emission of $NO_x$, local oxidation, deposition of $HNO_3$) and explicitly calculates $NO_3^-$ mass concentrations and $\delta^{15}N\text{-}NO_3^-$ in snow. Due to the reproducible depth profile of $\delta^{15}N\text{-}NO_3^-$

within 1 km (section 3.3), we assume $\delta^{15}N\text{-}NO_3^-$ composition is spatially uniform at DML and thus a 1D model is appropriate for our site. The atmospheric boundary layer in the model is represented by a single box above the snow pack. The 1 m snow pack is divided into 1000 layers of 1 mm thickness. Below the photic zone of the snow pack, the $NO_3^-$ mass concentrations and $\delta^{15}N\text{-}NO_3^-$ values are assumed to be constant and thus archived during the model run. The model is run for 25 years, which is sufficient to reach steady state. The input data is provided in Table S3.

Photolysis rate coefficients of $NO_3^-$ (J($NO_3^-$)) above and within the snowpack are used by the TRANSITS model runs as input for this study, and are modelled off-line using the tropospheric ultraviolet and visible (TUV)-snow radiative transfer model (Lee-Taylor and Madronich, 2002). The following assumptions were made: i) a clear aerosol-free sky, ii) extra-terrestrial irradiance from Chance and Kurucz (2010), and iii) a constant Earth-Sun distance as that on 27 December 2010 (Erbland et al., 2015). The TUV-snow radiative transfer model was constrained by optical properties of the Dome C snow pack (France et

al., 2011), notably an e-folding depth of i) 10 cm in the top 0.3 m, and ii) 20 cm below 0.3 m (Erbland et al., 2015), to compute J($^{14/15}NO_3^-$) profiles as a function of solar zenith angle (SZA) and TCO (Erbland et al., 2015) (Fig. S2; dashed lines).





The set up used in this paper is similar to Erbland et al. (2015) except for the following modifications. We use the TCO from the NIWA Bodeker combined dataset version 3.3, at the location of the snow pit site, averaged from 2000 to 2016

(http://www.bodekerscientific.com/data/total-column-ozone). The year round atmospheric $NO_3^-$ concentration is taken from Weller and Wagenbach (2007), and the meteorology data is taken from Utrecht University automatic weather Station (AWS) at DML05/Kohnen (AWS9; https://www.projects.science.uu.nl/iceclimate/aws/files_oper/oper_20632). The snow accumulation rate is set to 6 cm yr$^{-1}$ (w.e.) (Sommer et al., 2000). We carried out a sensitivity analysis to evaluate the impact of variable accumulation rate, timing of snowfall, and e-folding depth on the snow profile of $NO_3^-$ and $\delta^{15}N\text{-}NO_3^-$. The

sensitivity tests were as followed: the snow accumulation rate was varied between the bounds seen in the last 1000-years at DML; the snow accumulation rate was varied from year to year according to our snow pit profile; the timing of the snow accumulation was varied throughout the year; and the e-folding depth was varied within the range of observations from this study and previously at Dome C. To evaluate the sensitivity of archived $\delta^{15}N\text{-}NO_3^-$ to e-folding depth, changes the $J(^{14/15}NO_3^-)$ profiles for Dome C (Erbland et al., 2015) were recalculated and used as TRANSITS input by scaling the surface value of

$J(^{14/15}NO_3^-)$ with a new e-folding depth (2, 5, 10, 20 cm). An example is shown in Fig. S2a for SZA = 70º, TCO = 300 DU and an e-folding depth of 5 cm. The top 2 mm are retained from the Dome C base case to account for non-linearities in snow radiative transfer in snow, which are strongest in the non-diffuse zone right below the snow surface (Fig. S2b). It is noted that TUV-snow model estimates of down-welling or $2\pi\ J(NO_3^-)$ above the snow surface at the latitude of Dome C or DML (= 75º S) compare well to observations at Dome C in January 2012, whereas they are a factor three higher than measurements at

DML in January 2017 (Table S4 and section 2.6). This should not affect the results of the sensitivity study, which aims to explore relative changes of archived $\delta^{15}\text{-}NO_3^-$ due to a prescribed change in e-folding depth.

## 3 Results

### 3.1 Snow pit dating

Dating of the snow pits was based on the measured concentrations of $Na^+$, MSA, and $nss\text{-}SO_4^{2-}$ following previous aerosol and

ice core studies at DML (Göktas et al., 2002;Weller et al., 2018). Here, $Na^+$ mass concentrations have a sharp, well-defined peak in the austral spring/late winter, while MSA and $nss\text{-}SO_4^{2-}$, primarily derived from the biogenic production of dimethylsulfide (DMS), record maximum concentrations in the austral autumn. Non-sea salt $SO_4^{2-}$ ($nss\text{-}SO_4^{2-}$) often displays a second peak corresponding to late austral spring/summer sometimes linked to MSA. Spring seasons were defined as 1 September and positioned at the $Na^+$ peak, while autumn seasons were defined as 1 April and positioned where a MSA and

$nss\text{-}SO_4^{2-}$ peak aligned (Fig. S3). Annual layer counting of $Na^+$ layers shows snow pit A spans 8 years from autumn 2009 to summer 2017 and snow pit B spans 9 years from summer 2008 to summer 2017 with an age uncertainty of ± 1 year at the base of the snow pit. The mean snow accumulation rate for the snow pits is estimated to be 6.3 ± 1.4 cm yr$^{-1}$ (w.e.), consistent with



accumulation rates of 6.0 - 7.1 cm yr$^{-1}$ (w.e.) from snow pits and ice cores from DML (Sommer et al., 2000;Hofstede et al., 2004;Oerter et al., 2000).

**3.2 Nitrate mass concentrations**

Atmospheric $NO_3^-$ mass concentrations ($C_{aerosol}$) were estimated from high-volume aerosol filters by the ratio of total $NO_3^-$ mass loading to the total volume of air pumped through the filter at STP conditions following Eq. (2), and assuming a uniform loading of the aerosol filter.

$$C_{aerosol} = NO_3^- \text{ mass loading / air volume (STP)} \tag{2}$$

Aerosol mass concentrations range from 0.5 to 19 ng m$^{-3}$ and show a downward trend throughout January 2017 ($R^2$=0.55; p= <0.001; Fig. 3). In contrast, $NO_3^-$ mass concentrations in the skin layer increase during the month from 136 to 290 ng g$^{-1}$. Nitrate mass concentrations in both snow pits, which range from 23 to 142 ng g$^{-1}$, are substantially lower than those in the skin layer. Compared to Dome C, average annual atmospheric, skin layer and snow pit mass concentrations are lower at DML (Table 2), in agreement with higher $NO_3^-$ mass concentrations found at lower snow accumulation sites (Erbland et al., 2013). The $NO_3^-$ mass concentration profile in the upper 50 cm of the snow pack at Dome C shows an exponential decrease with depth and becomes relatively constant at 35 ng g$^{-1}$ at 20 cm compared to 160-1400 ng g$^{-1}$ in the skin layer (Figs. 1 and 4; (Erbland et al., 2013;Frey et al., 2009). While the highest $NO_3^-$ mass concentrations in the snow pack at DML are also found in the skin layer, the concentration profile exhibits a different pattern. The sharp decrease in $NO_3^-$ mass concentration occurs in the top ~5 mm at which point the snow pit records inter-annual variability in the $NO_3^-$ mass concentration. Nitrate mass concentrations at DML exhibit a maximum in summer and winter minimum.

Although the Dome C depth profiles of $NO_3^-$ mass concentration do not record seasonal variability, year-round measurements of skin layer and atmospheric $NO_3^-$ mass concentrations exhibit sharp maximum during sunlit conditions in spring and summer and low mass concentrations in winter. This annual cycle is consistent both i) spatially across Antarctica (McCabe et al., 2007;Wolff et al., 2008;Erbland et al., 2013;Frey et al., 2009), and ii) temporally over last 7 years (Fig. 1) (Erbland et al., 2015;Erbland et al., 2013;Frey et al., 2009).

While the precision of the IC measurement of $NO_3^-$ is better than 2 %, the spatial variability at DML of $NO_3^-$ in the skin layer exceeds this. During the sampling campaign, five skin layer samples were taken from an area of ~2500 m$^2$ at the flux site (snow surface had sastrugi up to 10 cm) to understand how representative the snow pit mass concentrations are of the greater study area. We found that the spatial variability of $NO_3^-$ mass concentrations and $\delta^{15}$N-$NO_3^-$ at DML was 10 % and 17 % respectively. At Dome C, the spatial variability of $NO_3^-$ mass concentrations is between 15 and 20 %. We note that this variability includes the natural spatial variability and the operator sampling technique.

**3.3 Isotopic composition of nitrate**

Atmospheric $\delta^{15}$N-$NO_3^-$ ranges from -49 to -20 ‰ at DML and -9 to 8 ‰ at Dome C during the January campaign, and is depleted with respect to the skin layer, which ranges from -22 to 3 ‰ at DML (Fig. 3). Similar to the $NO_3^-$ mass concentrations,





the $\delta^{15}$N-NO$_3^-$ in the depth profile at DML exhibits large variability between seasons (-3 to 99 ‰) with more enriched values in spring and summer with respect to winter (Fig. 4). The $\delta^{15}$N-NO$_3^-$ in both snow pits at DML show extremely good reproducibility with depth indicating there is little spatial variability within 1 km at the site (Fig. 4). The $\delta^{15}$N-NO$_3^-$ in snow pits at Dome C do not preserve a seasonal cycle. However, in parallel with the exponential decay of NO$_3^-$ mass concentrations with depth at Dome C, there is a strong increase in the $\delta^{15}$N-NO$_3^-$ with depth. At Dome C, $\delta^{15}$N-NO$_3^-$ increases up to 250 ‰

in the top 50 cm, this increase is weaker at DML (up to 80 ‰ in the top 10 cm at which point seasonal cycles are evident). At Dome C, although no annual cycle is preserved in the snow pack, the year-round measurements of $\delta^{15}$N-NO$_3^-$ in the atmosphere decrease during sunlit conditions in spring and summer (Fig. 1). While the $\delta^{15}$N-NO$_3^-$ in the skin layer has a spring minimum that increases to a maximum at the end of summer (Fig. 1). Skin layer $\delta^{15}$N-NO$_3^-$ is about 25 ‰ higher than atmospheric $\delta^{15}$N-NO$_3^-$. Nitrate mass concentration and $\delta^{15}$N-NO$_3^-$ composition data for aerosol, skin layer and snow pit samples are available

in Winton et al. (2019b).

### 3.4 Archived nitrate mass concentration and isotopic composition

We calculate archived values of NO$_3^-$ mass concentration and $\delta^{15}$N-NO$_3^-$ which represent the archived mass fraction and isotopic composition reached below the photic zone. Archived values were calculated by averaging the NO$_3^-$ and $\delta^{15}$N-NO$_3^-$ values below the photic zone, i.e., 30 cm (section 4.4). The archived NO$_3^-$ mass concentration and $\delta^{15}$N-NO$_3^-$ values for snow

pit A were 60 ng g$^{-1}$ and 50 ‰, the and archived NO$_3^-$ mass concentration for snow pit B was 50 ng g$^{-1}$. Note that $\delta^{15}$N-NO$_3^-$ values were measured below 30 cm in snow pit B. These measured values are half of those expected for a site with a snow accumulation rate of 6 cm yr$^{-1}$ (w.e.) in the spatial survey from Erbland et al. (2013) (Table 2).

### 3.5 Nitrate mass flux estimates

The total deposition flux (F) of NO$_3^-$ is partitioned into wet and dry deposition fluxes (F$_{wet}$ and F$_{dry}$ respectively; Eq. (3)), and

can be estimated using the measured mass concentration of NO$_3^-$ in the snow pack (C$_{snow}$) and the local snow accumulation rate (A; Eq. (4)). Estimates of the dry deposition rate (F$_{dry}$) of NO$_3^-$ were calculated using Eq. (5) using the atmospheric mass concentrations of NO$_3^-$ (C$_{aerosol}$) and a dry deposition velocity (V$_{dry\ deposition}$) of 0.8 cm s$^{-1}$, and are reported in Table S5. This deposition velocity is based on the dry deposition of HNO$_3$ at South Pole (Huey et al., 2004) which has a similar snow accumulation rate (6.4 cm yr$^{-1}$ (w.e.); Mosley-Thompson et al. (1999)) to DML. Other estimates of dry deposition velocities

include 0.05-0.5 cm s$^{-1}$ for HNO$_3$ over snow (Hauglustaine et al., 1994; Seinfeld and Pandis, 1998), 1.0 cm s$^{-1}$ for NO$_3^-$ over the open ocean (Duce et al., 1991), and an apparent deposition velocity of 0.15 cm s$^{-1}$ for summer HNO$_3$ at Dome C (Erbland et al., 2013). The estimated apparent NO$_3^-$ deposition velocity at Dome C is low because of the strong recycling of NO$_3^-$ on the polar plateau in summer, i.e., reactive nitrogen is re-emitted from the skin layer to the atmosphere. Thus, the dry deposition velocity at DML is likely to lie between 0.15 and 0.8 cm s$^{-1}$. We assume that a constant deposition velocity throughout the

campaign is appropriate for DML.

$$F = F_{wet} + F_{dry} \tag{3}$$



$$C_{snow} = F / A \qquad (4)$$

$$F_{dry} = C_{aerosol} V_{dry\ deposition} \qquad (5)$$

390 Note that Eq. (4) does not take into account post-depositional processes of non-conservative ions, such as $NO_3^-$. We follow the approach of Erbland et al. (2013) who use an archived $NO_3^-$ flux ($F_a$) to represent the downward $NO_3^-$ flux which escapes the photic zone towards deeper snow layers. Using simple mass balance, we can then estimate the flux of $NO_3^-$ ($F_{re-emit}$), which is re-emitted from the snow pack to the overlaying atmosphere (Eq. (6)).

$$F_{re-emit} = F - F_a \qquad (6)$$

395 Taking a simple mass balance approach, a schematic of $NO_3^-$ mass fluxes are illustrated in Fig. 5a. As the atmospheric campaign did not cover an entire annual cycle, we use estimates of atmospheric $NO_3^-$ fluxes at DML reported by Pasteris et al. (2014) and Weller and Wagenbach (2007) of 43 and 45 pg m$^{-2}$ s$^{-1}$, respectively, as a year round dry deposition fluxes. Due to the linear relationship of ice core $NO_3^-$ mass concentrations with the inverse accumulation, the authors assume that the magnitude of the dry deposition flux is homogenous over the DML region. Mean annual mass concentrations of $NO_3^-$ in our

400 snow pits suggest a total $NO_3^-$ deposition mass flux of 110 pg m$^{-2}$ s$^{-1}$ and therefore a wet deposition mass flux of 65 pg m$^{-2}$ s$^{-1}$.

However, at relatively low snow accumulation sites where photolysis drives the fractionation of $NO_3^-$ from the surface snow to atmosphere (Frey et al., 2009), it is necessary to take into account the skin layer in the $NO_3^-$ flux budget as this air-snow interface is where air-snow transfer of $NO_3^-$ takes place. We use the available $NO_3^-$ mass concentrations measured in aerosol,

405 skin layer, and snow pits from the ISOL-ICE campaign to estimate the mass flux budget for January 2017 (Fig. 5b). The dry deposition mass flux of atmospheric $NO_3^-$ during January 2017 at DML averages $64 \pm 38$ pg m$^{-2}$ s$^{-1}$ (Table S5). The $NO_3^-$ mass flux to the skin layer is 360 pg m$^{-2}$ s$^{-1}$, however only 110 pg m$^{-2}$ s$^{-1}$ of $NO_3^-$ is archived. Considering the active skin layer, only 30 % of deposited $NO_3^-$ is archived in the snow pack while 250 pg m$^{-2}$ s$^{-1}$ is re-emitted to the overlaying atmosphere.

### 3.6 Fractionation constants

410 Fractionation constants were calculated following the approach of Erbland et al. (2013) which assumes a Rayleigh single loss and irreversible process of $NO_3^-$ removal from the snow. As this approach may oversimplify the processes occurring at the air-snow interface, Erbland et al. (2013) referred to the quantity as an "apparent" fractionation constant. Thus, the apparent fractionation constant represents the integrated isotopic effect of the processes involving $NO_3^-$ in the surface of the snow pack and in the lower atmosphere. The apparent fractionation constant is denoted as $^{15}\varepsilon_{app}$ and calculated using Eq. (7).

415 $$\ln(\delta^{15}Nf + 1) = {}^{15}\varepsilon \times \ln f + \ln(\delta^{15}N_0 + 1) \qquad (7)$$

where $\delta^{15}Nf$ and $\delta^{15}N_0$ are the δ-values in the initial and remaining $NO_3^-$, and $f$ is the remaining $NO_3^-$ mass concentration. The ε values are related to the commonly used fractionation factor α by $\varepsilon = \alpha - 1$. The $^{15}\varepsilon_{app}$ derived for snow pits in the photic zone is 12 ‰.



### 3.7 Simulated nitrate mass concentrations and isotopic ratios from TRANSITS modelling

Simulated TRANSITS results for the air snow interface are illustrated in Fig. 6. In the atmosphere, the TRANSITS model is forced with the smoothed profile of year-round atmospheric $NO_3^-$ measurements from the DML site (Weller and Wagenbach, 2007) which has the highest mass concentrations in spring and summer with a maximum of 80 ng m$^{-3}$ in November and a winter minimum of 2 ng m$^{-3}$ (Fig. 6b). Although we only have measurements of $\delta^{15}N$-$NO_3^-$ in January, the simulated atmospheric $\delta^{15}N$-$NO_3^-$ values for January are greater than the measurements available from this study. The annual cycle of

simulated atmospheric $\delta^{15}N$-$NO_3^-$ shows a 40 ‰ dip in spring to -32 ‰ from winter values which coincides with the simulated atmospheric $NO_3^-$ mass concentration increase in spring. The highest atmospheric $\delta^{15}N$-$NO_3^-$ values (7 ‰) occur in winter. In the skin layer, the simulated $NO_3^-$ mass concentrations are an order of magnitude greater than our observations in January and we outline possible reasons for this discrepancy in the discussion (section 4.1). The simulated annual cycle of $NO_3^-$ mass concentrations in the skin layer steadily rise in spring and reach a peak in January when they begin to decline to the lowest

concentration in winter. Simulated skin layer $\delta^{15}N$-$NO_3^-$ values in January are ~10 ‰ higher than our highest observations for that month. They begin to decrease by 24 ‰ in spring at the same time as atmospheric $\delta^{15}N$-$NO_3^-$ values decrease. In October and November, the skin layer $\delta^{15}N$-$NO_3^-$ values begin to rise up to 14 ‰ in February.

The seasonality of $NO_3^-$ mass concentrations and $\delta^{15}N$-$NO_3^-$ values in the atmosphere and skin layer at DML is consistent with Dome C (Fig. 1). Similar to Dome C, $NO_3^-$ mass concentrations in the skin layer start to rise two months earlier than

atmospheric $NO_3^-$ mass concentrations and the summer maximum is later. While the seasonality of $\delta^{15}N$-$NO_3^-$ in the skin layer and atmosphere co-vary, simulated skin layer $\delta^{15}N$-$NO_3^-$ values are enriched relative to atmospheric values on average by 80 ‰.

The simulated $NO_3^-$ mass concentrations and $\delta^{15}N$-$NO_3^-$ values in the snow pit are illustrated in Fig. 7. Both the simulated depth profile of $NO_3^-$ mass concentration and $\delta^{15}N$-$NO_3^-$ for an accumulation rate of 6 cm yr$^{-1}$ (w.e.) show seasonal variability in the

first year with a range of of 380 ng g$^{-1}$ and 20 ‰, which decreases with depth to a range of 95 ng g$^{-1}$ and 10 ‰ in the fourth year. Also plotted are the simulated $NO_3^-$ and $\delta^{15}N$-$NO_3^-$ depth profiles for accumulation rates of 2.5 cm yr$^{-1}$ (w.e.) and 100 cm yr$^{-1}$ (w.e.). As the accumulation rate increases, the annual layers of $\delta^{15}N$-$NO_3^-$ become thicker, the seasonal amplitude increases, the mean annual $\delta^{15}N$-$NO_3^-$ value decreases, and the $\delta^{15}N$-$NO_3^-$ values in the top 10 cm decrease. At very low snow accumulation rates, the seasonal cycle is smoothed, as in the case of Dome C. A similar pattern is observed for the simulated

$NO_3^-$ mass concentrations with depth: seasonal cycles of $NO_3^-$ mass concentrations are more pronounced at higher snow accumulation rates, while inter-annual variability is smoothed at Dome C. The simulated archived (i.e., annual average of the first year below 1 m) $NO_3^-$ mass concentration, $\delta^{15}N$-$NO_3^-$, and $NO_3^-$ mass flux values are 120 ng g$^{-1}$, 130 ‰, and 210 pg m$^{-2}$ yr$^{-1}$, respectively. The simulated annual average $^{15}\varepsilon_{app}$ is -19 ‰ for the top 30 cm (i.e., active photic zone with an e-folding depth of 10 cm).



## 4 Discussion

### 4.1 Validation of results

Our January 2017 $NO_3^-$ measurements agree well with values reported in the literature, and largely with the simulated results from the TRANSITS model with the exception of skin layer $NO_3^-$ mass concentration. While we made the first measurements of atmospheric, skin layer and snow pit $\delta^{15}N$-$NO_3^-$, and skin layer $NO_3^-$ mass concentrations at DML, there are published measurements of $NO_3^-$ mass concentrations in snow pits and our concentrations agree well with those (Weller et al., 2004). Our atmospheric mass $NO_3^-$ concentrations in January 2017 are lower than those observed in 2003 by Weller and Wagenbach (2007) which could be due to inter-decadal variability of atmospheric $NO_3^-$ (which varied by 30 ng $g^{-1}$ over summer between 2003 and 2005) or reflect the different filter substrates used (Teflon/nylon versus glass fibre).

Overall, the simulated results are greater than our January observations, particularly the skin layer $NO_3^-$ mass concentrations (Fig. 6d). The discrepancy between the significantly higher simulated $NO_3^-$ mass concentrations than observations in the skin layer was also found at Dome C. Erbland et al. (2015) suggested that this discrepancy could be related to either a sampling artefact, snow erosion or a modelled time response to changes in past primary inputs. We provide an alternative explanation for the extremely high simulated $NO_3^-$ mass concentrations in the skin layer using measurements of $NO_3^-$ mass concentration in diamond dust and hoar frost collected daily from Polyvinyl chloride (PVC) sheets at Dome C in summer 2007/08, i.e. new deposition. New deposition of diamond dust had $NO_3^-$ mass concentrations up to 2000 ng $g^{-1}$, which is four times greater than that observed in natural snow from the skin layer at the same time (Fig. S4). Similarly, new deposition of hoar frost had $NO_3^-$ concentrations up to 900 ng $g^{-1}$, which is three times greater than the skin layer snow. The formation of surface hoar frost occurs by co-condensation, i.e. the simultaneous condensation of water vapour and $NO_3^-$ at the air-snow interface. Recent modelling suggests that co-condensation is the most important process explaining $NO_3^-$ incorporation in snow undergoing temperature gradient metamorphism at Dome C (Bock et al., 2016). Diamond dust can also scavenge high concentrations of $HNO_3$ at Dome C (Chan et al., 2018). Furthermore, the top layer of the snow pack is only 1 mm thick in the TRANSITS model, which is where we would expect the highest concentrations due to the exponential decay of $NO_3^-$ with depth (Fig. S4). If indeed the higher simulated values in the skin layer can be explained by hoar frost and diamond dust, then we can have greater confidence in the depth profile of $NO_3^-$ concentration. It is interesting to note that these higher simulated values in the skin layer do not impact the simulated depth profiles (Fig. 5). In summary, it is likely that we do not measure such high hoar frost and diamond dust values in the skin layer because of sampling artefacts or blowing snow, which can dilute or remove the diamond dust and hoar frost.

While not yet observed elsewhere on the Antarctic continent, over the short intensive sampling period at DML we observe significant variability in $NO_3^-$ mass concentrations and $\delta^{15}N$-$NO_3^-$ values that resembles a diurnal cycle. Over 4 hours, the skin layer $NO_3^-$ mass concentrations varied by 46 ng $g^{-1}$, the skin layer $\delta^{15}N$-$NO_3^-$ by 21 ‰, and the atmospheric $\delta^{15}N$-$NO_3^-$ by 18 ‰. Other coastal studies have attributed daily variability to individual storm events (Mulvaney et al., 1998;Weller et al., 1999).



We note that the sampling duration is too short to confirm any diurnal patterns but it would be interesting to investigate this further in future work.

## 4.2 Nitrate deposition

Here we discuss the various processes in which $NO_3^-$ can be deposited to the skin layer at DML. As we have just one month of atmospheric and skin layer data, our ability to look at the deposition on seasonal scales is limited, however we provide new insights into the austral summer deposition processes.

While it is common to measure nitrogen species in snow and aerosol samples as the $NO_3^-$ ion using ion chromatography, nitrogen species can be deposited in various forms either by wet or dry deposition to the skin layer. We note that organic $NO_3^-$
plays are little role in determining snow concentrations (Jones et al., 2007;Wolff et al., 2008), and as such we focus our discussion on inorganic $NO_3^-$. The various nitrogen species include, i) a neutral salt ($NO_3^-$ co-deposition with sea salt or mineral dust; Wolff et al. (2008)), ii) $NO_3^-$ in air ($HNO_3$ in gas-phase plus particulate $NO_3^-$). Following the terminology of Erbland et al. (2013), this is referred to as "atmospheric $NO_3^-$" and is represented by the $NO_3^-$ mass concentrations measured on our aerosol filters. Atmospheric $NO_3^-$ can either be deposited as dry deposition by adsorption to the snow surface as $HNO_3$ has a
strong affinity for ice surfaces (Abbatt, 2003;Huthwelker et al., 2006) or scavenged by precipitation as wet deposition, and iii) co-condensation of $HNO_3$ and water vapour onto snow crystals (Thibert and Domine, 1998).

Depending on the deposition pathway, $NO_3^-$ can either be predominantly incorporated into the bulk snow crystal or be adsorbed onto the surface of the snow crystal. Deposition pathways include co-condensation (formation of surface hoar frost), riming (deposition of supercooled fog droplets), and adsorption of $HNO_3$ onto the snow crystal surface (dry deposition) (Röthlisberger
et al., 2002). Both co-condensation (Bock et al., 2016) and dry deposition of $HNO_3$, at very cold temperatures, can elevate $NO_3^-$ mass concentrations in the skin layer. Furthermore, trace nitrogen impurities present in the interstitial air in the porous snow pack may be incorporated in snow crystals. While scavenging of $NO_3^-$ by snow (wet deposition) occurs sporadically throughout the year, dry deposition of particulate $NO_3^-$ or surface adsorption may take place continuously throughout the year. We see both of these deposition processes taking place during January 2017.

## 4.2.1 Wet deposition

Precipitation at DML can occur either through sporadic cyclonic intrusions of marine air masses from the adjacent ocean associated with large amounts of precipitation, or clear sky diamond dust that contributes smaller amounts to the total precipitation (Schlosser et al., 2010). Overall, extreme precipitation events dominate the total precipitation (Turner et al., 2019). In austral summer, the transport of marine aerosol to DML is mediated by two synoptic situations, i) low-pressure
systems from the eastern South Atlantic associated with high marine aerosol concentrations, and ii) persistent long-range transport that provides background aerosol deposition during clear sky conditions (Weller et al., 2018). Weller et al. (2018) suggest that dry deposition of marine aerosol is dominant over wet deposition at DML. In contrast, Dome C receives predominantly diamond dust, and thus aerosol deposition is different there.





More specifically, precipitation during our sampling campaign in January 2017 was relatively low compared to previous years.
Modelled daily precipitation at the nearest Regional Atmospheric Climate Model (RACMO2; Van Meijgaard et al. (2008)) grid point (75.0014°S, 0.3278°W) is illustrated in Fig. 3a. The largest precipitation event of the month was on 1 January (0.27 mm) resulting from a low-pressure system in the South Atlantic (Fig. S5). For the rest of the month, half of the days had zero precipitation and the other half had very little precipitation (~0.05 mm per day).

We use the RACMO2 daily precipitation data to identify whether the cyclonic intrusions of marine air masses provide wet
deposition of $NO_3^-$ to the site in January. In the skin layer, we observe that $NO_3^-$ mass concentrations and other sea salt ions co-vary (Fig. S6) suggesting similar deposition pathways of these ions. Some peaks in the skin layer $NO_3^-$ concentration are accompanied by fresh snow laden with relatively high sea salt aerosol concentrations and atmospheric $NO_3^-$ mass concentrations, for example on 1, 13, and 18 January 2017. Such deposition events have also been observed on the Antarctic coast (Wolff et al., 2008). During the formation of precipitation, essentially all $HNO_3$ is removed from the gas-phase due to
its high solubility in liquid clouds (Seinfeld and Pandis, 1998). Therefore, $HNO_3$ can be scavenged from the atmosphere and deposited as $NO_3^-$ in the skin layer. The uptake of $HNO_3$ onto the snow and ice crystal surface during and after precipitation can also contribute further to the $NO_3^-$ mass concentrations found in the skin layer. On some precipitation days, we observe lower atmospheric $NO_3^-$ mass concentrations and higher skin layer $NO_3^-$ mass concentrations that could be a result of $HNO_3$ scavenging. Mulvaney et al. (1998) observed higher skin layer concentrations in days when there was little snow accumulation
and concluded that $NO_3^-$ is directly up taken onto the surface by dry deposition of particulate $NO_3^-$ and surface adsorption of $HNO_3$ (gas-phase) (Mulvaney et al., 1998). With only one month of data it is difficult to see the impact of wet deposition on the $NO_3^-$ concentration in the skin layer; i.e. whether fresh snowfall dilutes the $NO_3^-$ concentration in the skin layer or whether it scavenges $HNO_3$ (gas-phase) resulting in higher concentrations of $NO_3^-$ in the skin layer. Most likely both processes are occurring. We note that due to post-depositional processes (section 3) any short-term signals observed in the skin layer are
unlikely to be preserved. Even at the South Pole where the snow accumulation rate is slightly higher (8.5 cm $yr^{-1}$ (w.e.); (Mosley-Thompson et al., 1999) than DML deposition, $NO_3^-$ peaks are substantially modified after burial (Dibb and Whitlow, 1996).

### 4.2.2 Dry deposition

In order to investigate dry deposition of $NO_3^-$, we first look at atmospheric $NO_3^-$ in relation to the wind direction and air mass
back trajectories. The mean annual wind direction at the site is 65°, and January 2017 is no exception (Figs. 3 and S7). There is an excursion from the predominant wind direction between 19-22 January, where the wind direction switches to the southwest. Although there are no studies indicating fractionation of $\delta^{15}N$-$NO_3^-$ in the atmosphere during atmospheric transport from the plateau to the coast, we do not see elevated $NO_3^-$ mass concentrations during this period nor do we see a marked difference in isotopic signature that is similar to Dome C at this time (Fig. 4). This, in line with air mass back trajectories (not
shown) suggests that long-range transport of $NO_3^-$ re-emitted from inland sites of the Antarctic did not reach DML during our campaign. We can also rule out any downwind contamination from the station.


High concentrations of sea salt and mineral dust can promote the conversion of $HNO_3$ (gas-phase) to aerosol, as well as trapping $NO_3^-$ (gas-phase) on salty snow surfaces. We see a relationship between sea salt aerosol and atmospheric $NO_3^-$ ($R^2$= 0.59; p=<0.001) suggesting that even 550 km inland from the coast sea salt could promote the conversion of $HNO_3$ to atmospheric

$NO_3^-$, although we acknowledge that our filters capture both aerosol $NO_3^-$ and $HNO_3$, and sea salt concentrations are much higher at Halley and coastal Antarctica where this mechanism sporadically occurs (Wolff et al., 2008).

Scavenging of atmospheric $NO_3^-$ is largely responsible for the high mass concentrations observed in the skin layer. Variation in the concentration and isotopic signature of aerosol and surface snow at DML over January 2017 suggests atmospheric $NO_3^-$ is the source of $NO_3^-$ to the skin layer. Throughout the month, the increase in the skin layer concentration of summer $NO_3^-$

appears to be closely related to the decrease in the atmospheric $NO_3^-$ mass concentrations (Fig. 3). There is a lag between atmospheric and skin layer $NO_3^-$ i.e. atmospheric $NO_3^-$ mass concentrations precede skin layer $NO_3^-$ mass concentrations by day or two, however a longer time series is required to confirm this. The lag suggests that atmospheric $NO_3^-$ is a source of $NO_3^-$ to the skin layer, in line with Dome C where the snow pack is the dominant source of $NO_3^-$ to the skin layer via the overlying air in summer. Furthermore, as atmospheric $NO_3^-$ is deposited to the snow surface, [15]N is preferentially removed

first leaving the air isotopically depleted relative to the isotopically enriched snow (Frey et al., 2009). Fig. 3 illustrates that the $\delta^{15}N$-$NO_3^-$ in the atmosphere is depleted with respect to the $\delta^{15}N$-$NO_3^-$ in the skin layer snow. In the short time series, there are some periods where the $\delta^{15}N$-$NO_3^-$ in the snow and atmosphere are in phase, for example, 3-13 January 2017. During other periods, the $\delta^{15}N$-$NO_3^-$ in the snow and atmosphere switch to being out of phase emphasising $NO_3^-$ isotopic fractionation during those periods. Both $HNO_3$ and peroxynitric acid ($HNO_4$) can be adsorbed to the snow surface in tandem (Jones et al., 2014),

and although we have no direct measurements of these during the campaign, based on previous studies we suggest that $HNO_3$ is the most likely form of nitrogen to skin layer (Jones et al., 2007;Chan et al., 2018).

Furthermore, the adsorption of $HNO_3$ on ice surfaces is temperature dependent with higher uptake at lower temperatures (Abbatt, 1997;Jones et al., 2014). However, there is only a relatively small temperature difference between Dome C and DML (summer mean temperature -30 °C and -25 °C respectively) which is not enough to drive a large difference in $HNO_3$ uptake

(Jones et al., 2014). In addition, the uptake is not dependent on the $HNO_3$ concentration in the air (Abbatt, 1997). However, the seasonal temperature difference at an individual site (i.e., DML or Dome C) is far greater, which could allow a seasonal dependence on the uptake and loss of $NO_3^-$ in the skin layer, which results in the retention of a greater proportion of $NO_3^-$ in summer (Chan et al., 2018).

### 4.2.3 Annual cycle of nitrate deposition

We use the simulated annual cycle of $NO_3^-$ from TRANSITS model to describe the seasonal evolution of $NO_3^-$ deposition to DML. While $NO_3^-$ deposited to DML can be sourced from the sedimentation of polar stratospheric clouds in winter and we assume the atmospheric $NO_3^-$ loading is uniform under the polar vortex, in spring and summer $NO_3^-$ net deposition is related to local photochemistry and subsequent post-depositional processing rather than primary $NO_3^-$ sources. At this time, deposition





of $NO_3^-$ can be through the transport of re-emitted $NO_3^-$ from the surface snow at low accumulation regions of the polar plateau,
or $NO_3^-$ produced *in situ* at DML in spring and summer.

The annual cycle of atmospheric $NO_3^-$ deposition (Weller and Wagenbach, 2007) indicates how much $NO_3^-$ is deposited to the skin layer from the atmosphere (Figs. 5 and 6). Year-round $NO_3^-$ mass concentrations have been measured in surface snow at the coastal sites of Halley (Mulvaney et al., 1998;Jones et al., 2011) and Neumayer Stations (Wagenbach et al., 1998), and the low snow accumulation site at Dome C (Fig. 1). An agreement with our simulated results, at all Antarctic sites the highest
atmospheric $NO_3^-$ mass concentrations are found during summer when the solar radiation is close to its annual maximum and $NO_3^-$ photolysis is strongest. The summer maximum at Dome C results from co-condensation of $NO_3^-$ (Bock et al., 2016). This intense uptake in the skin layer in summer is driven by the strong temperature gradient in the upper few centimetres of the snow pack, highlighting that both physical (deposition; Bock et al. (2016); Chan et al., 2018) and chemical ($NO_3^-$ re-emission; Erbland et al. (2015)) processes explain the cycling of $NO_3^-$ between the air and snow. The lowest $NO_3^-$ mass concentrations
in the skin layer are found in winter.

Year-round atmospheric $NO_3^-$ data at DML and Dome C shows atmospheric $NO_3^-$ is at a minimum in April to June and reaches a maximum in late November, slightly out of phase with skin layer $NO_3^-$ (Wagenbach et al., 1998;Erbland et al., 2013) (Figs. 1 and 6). The fact that the seasonality of simulated skin layer and atmospheric $NO_3^-$ at DML matches observations at other sites in Antarctica gives confidence in our TRANSITS model results (Fig. 6).

**4.2.4 Nitrate mass fluxes**

Our two $NO_3^-$ mass flux scenarios in Fig. 5 highlight the importance of the skin layer in the air-snow transfer of $NO_3^-$. Like Dome C, the greatest deposition flux of $NO_3^-$ is to the skin layer. The January dry deposition flux is greater than the annual mean flux estimated by Pasteris et al. (2014) and Weller and Wagenbach (2007) which is to be expected given the higher atmospheric $NO_3^-$ mass concentrations in summer (Fig. 6). The wet deposition flux, calculated for the greater DML region by
Pasteris et al. (2014), falls within our two scenarios. Furthermore, the simulated archived $NO_3^-$ flux at DML of 210 pg m$^{-2}$ s$^{-1}$ over predicts the observed $NO_3^-$ archived flux of 110 pg m$^{-2}$ s$^{-1}$ due to the higher simulated archived $NO_3^-$ mass concentrations. Interestingly, the simulated archived flux at Dome C (88 pg m$^{-2}$ s$^{-1}$) is lower than DML, yet the $NO_3^-$ deposition flux to the skin layer in January at Dome C is similar to DML. We continue our discussion focusing on the recycling and redistribution of $NO_3^-$ that occurs in the active skin layer emphasising its importance.

**4.3 Recycling and redistribution**

Post-depositional loss and redistribution of $NO_3^-$ is the dominant control on snow pack mass concentrations and $\delta^{15}$N-$NO_3^-$ isotopic signature on the Antarctic Plateau. Recycling of $NO_3^-$ at the air-snow interface comprises the following processes. Nitrate on the surface of a snow crystal can be lost from the snow pack (Dubowski et al., 2001), either by UV photolysis or evaporation. UV-photolysis produces NO, $NO_2$ and HONO while only $HNO_3$ can evaporate. Both of these processes produce
reactive nitrogen that can be released from snow crystal into the interstitial air and rapidly transported out of the snow pack to





the overlaying air via wind pumping (Zatko et al., 2013;Jones et al., 2000;Honrath et al., 1999;Jones et al., 2001). Here, $NO_2$ is either oxidised to $HNO_3$, which undergoes wet or dry deposition back to skin layer within a day, or transported away from the site (Davis et al., 2004a). If $HNO_3$ is re-deposited on the snow skin layer, it is available for $NO_3^-$ photolysis and/or evaporation again. Nitrate can be recycled multiple times between the boundary layer and the skin layer before it is buried in

deeper layers of the snow pack. Photolysis and/or evaporation of $NO_3^-$ and subsequent recycling between the air and snow alters the concentration and $\delta^{15}N$-$NO_3^-$ that is ultimately preserved in ice cores. Nitrate recycling therefore redistributes $NO_3^-$ from the active snow pack column to the skin layer via the atmosphere. Any locally produced $NO_2$ that is transported away from the site of emission represents a loss of $NO_3^-$ from the snow pack.

### 4.3.1 Evaporation

The desorption of $HNO_3$ from the snow crystal reduces the $NO_3^-$ concentration in the snow in coastal Antarctica (Mulvaney et al., 1998). The evaporation of $HNO_3$ is a two-step process, which involves the recombination of $NO_3^- + H^+ \longrightarrow HNO_3$ followed by a phase change to $HNO_3$ (gas-phase). First, theoretical estimates indicated that evaporation of $HNO_3$ should preferentially remove $^{15}N$ from the snow and release to the atmosphere leading to depletion in $\delta^{15}N$-$NO_3^-$ in the residual snow pack (Frey et al., 2009). Furthermore, recent laboratory experiments showed that evaporation imposes a negligible fractionation of $\delta^{15}N$-

$NO_3^-$ (Erbland et al., 2013;Shi et al., 2019). However, we find that the snow pack is enriched in $\delta^{15}N$-$NO_3^-$ relative to the atmosphere at DML (Figs. 3 and 6) and at Dome C (section 4.3.2). This fractionation observed in field studies cannot therefore be explained by evaporation, and must be attributed to different processes. It therefore follows that evaporation must be only a minor process in the redistribution of $NO_3^-$ between atmosphere and the snow pack above the Antarctic plateau.

### 4.3.2 Photolysis

We focus our discussion on photolysis, which is the dominant process responsible for $NO_3^-$ loss and redistribution and associated $\delta^{15}N$-$NO_3^-$ isotopic fractionation at low accumulation sites in Antarctica (Erbland et al., 2013;France et al., 2011). Nitrate photolysis occurs in the photochemically active zone of the snow pack, known as the snow photic zone. Below this, $NO_3^-$ is buried. Nitrate photolysis in the active snow pack results in the production of $NO_2$ leading to a reduction in the $NO_3^-$ concentration with depth in the snow pack (Fig. 4). In the photolysis-induced fractionation of $NO_3^-$, $^{14}N$ is preferentially

removed first resulting in an enrichment of $\delta^{15}N$-$NO_3^-$ in the snow pack. An individual snow layer is enriched when it is near the surface during sunlit conditions, i.e. spring and summer. Therefore, spring snow layers undergo strong $\delta^{15}N$-$NO_3^-$ enrichment as they are exposed to UV near the surface for the longest; late summer and autumn layers experience less $\delta^{15}N$-$NO_3^-$ enrichment as they are exposed for less time before sunlight disappears at the start of polar winter, during which new precipitation buries existing snowfall.

We provide five lines of evidence that photolysis is the dominant process for $NO_3^-$ recycling and redistribution at DML. Firstly, the highly enriched $\delta^{15}N$-$NO_3^-$ values of snow at DML and other Antarctic sites are among the most extreme observed on earth (Fig. S8) (Savarino et al., 2007), and cannot be explained by any known anthropogenic, marine or other natural sources. The





$\delta^{15}$N-NO$_3^-$ source signature of the main natural NO$_x$ sources (biomass burning, lightning, soil emissions is lower; $\delta^{15}$N-NO$_3^-$ <0 ‰) than anthropogenic NO$_x$ sources, which generally have positive $\delta^{15}$N-NO$_3^-$ values (-13< $\delta^{15}$N-NO$_3^-$ < 13 ‰) (Hastings
et al., 2013;Kendall et al., 2007 and references therein). The $\delta^{15}$N-NO$_3^-$ observations of aerosol, skin layer and snow pit at DML (-49< $\delta^{15}$N-NO$_3^-$ <99 ‰) lie outside of the range of natural and anthropogenic source end members, and thus cannot be explained by mixing of sources. Thus, our measurements at DML are unrelated to seasonal variations in NO$_x$ sources e.g. increased springtime agricultural emissions, which has been observed in the mid-latitudes. The contribution of natural sources to the Greenland snow pack $\delta^{15}$N-NO$_3^-$ signature has also been discarded (Geng et al., 2014;Geng et al., 2015). Furthermore,
the negative atmospheric $\delta^{15}$N-NO$_3^-$ values at DML (-20 to -49 ‰) are extremely low. Such low atmospheric $\delta^{15}$N-NO$_3^-$ values have only been observed in Antarctica, and show marked difference to other mid-latitude tropospheric aerosol (-10< $\delta^{15}$N-NO$_3^-$ <10 ‰; Freyer (1991). We acknowledge that stratospheric NO$_3^-$ contributes to NO$_3^-$ mass concentrations in snow in Antarctica. Although its isotopic signature is uncertain, estimates of stratospheric $\delta^{15}$N-NO$_3^-$ are 19 ± 3 ‰ (Savarino et al., 2007), and fall well outside of atmospheric observations at DML. The unique $\delta^{15}$N-NO$_3^-$ signature of low accumulation
Antarctic snow and aerosol is thus related to post-depositional processes specific to low accumulation sites in Antarctica.

Secondly, denitrification of the snow pack is seen through the $\delta^{15}$N-NO$_3^-$ signature which evolves from the enriched snow pack (-3 to 99 ‰), to the skin layer (-22 to 3 ‰), to the depleted atmosphere (-49 to -20 ‰) corresponding to mass loss from the snow pack (Fig. 4). Denitrification causes the $\delta^{15}$N-NO$_3^-$ of the residual snow pack NO$_3^-$ to increase exponentially as NO$_3^-$ mass concentrations decrease. Additionally, although not the focus of the study, denitrification causes the $\delta^{18}$O-NO$_3^-$ values to
increase in the residual NO$_3^-$ snow pack.

Thirdly, the application of TRANSITS to DML observations show that our observed atmospheric, skin layer and snow depth profiles of $\delta^{15}$N-NO$_3^-$ are similar to the simulated values where photolysis is the driving process (Figs. 6-7). Sensitivity analysis with TRANSITS is able to explain the observed snow pit $\delta^{15}$N-NO$_3^-$ variability (section 4.5). Nitrate isotope enrichment takes place in the top 25 cm, which is consistent with an e-folding depth of 10 cm. Here, the $\delta^{15}$N-NO$_3^-$ observations closely match
the simulated $\delta^{15}$N-NO$_3^-$ values and show enrichment to this depth indicating NO$_3^-$ photolytic redistribution at DML in the active photic zone of the snow pack (Fig. 7). Below the photic zone, $\delta^{15}$N-NO$_3^-$ values oscillate around a mean of ~125 ‰. The mean values of the $\delta^{15}$N-NO$_3^-$ observations are lower than the simulated values, which could be related to uncertainties in a number of factors, for example: i) a shallower e-folding depth than modelled. During our field measurements, we derived a lower e-folding depth of 2-5 cm (Fig. S1) at DML which could explain the lower enrichment in $\delta^{15}$N-NO$_3^-$ (section 4.5.2), ii)
lower JNO$_3^-$ values (NO$_3^-$ photolysis rate), which are related to a lower e-folding depth, and would lead to less enrichment of $\delta^{15}$N-NO$_3^-$ in the snow pack, iii) higher atmospheric NO$_3^-$ input, however $\delta^{15}$N-NO$_3^-$ values are not sensitive to variable atmospheric NO$_3^-$ mass concentrations (Erbland et al., 2015), and/or iv) variable accumulation which would shift the oscillations to the correct depth and lower the mean $\delta^{15}$N-NO$_3^-$ values below the photic zone (section 4.5.1). The difference between the simulated and snow pit values shows that DML site is less sensitive to photolysis than we expected from
TRANSITS modelling of $\delta^{15}$N-NO$_3^-$ along an accumulation gradient (Erbland et al., 2015).



Fourthly, we use Rayleigh isotopic fractionation to calculate apparent fractionation constants ($^{15}\varepsilon_{app}$) associated with $NO_3^-$ fractionation between phases during evaporation-condensation processes. Nitrate evaporation from the snow pack has a $^{15}\varepsilon_{app}$ of ~0 as determined by two independent studies (Erbland et al., 2013;Shi et al., 2019). This indicates that during $NO_3^-$ evaporation, the air above the snow is not replenished and thus there is only a small $NO_3^-$ mass loss. The isotopic fractionation
of $NO_3^-$ evaporation is negligible across most of Antarctica at cold temperatures of <-24 °C (Shi et al., 2019) which is the case for DML. However, evaporation of $NO_3^-$ at warmer temperatures (-4 °C) depletes the heavy isotopes of $NO_3^-$ remaining in the snow, and decreases the $\delta^{15}N\text{-}NO_3^-$ and the remaining snow by a few ‰ contrary to isotope effects of photolysis. In comparison, fractionation constants associated with laboratory studies and field observations of $NO_3^-$ photolysis are large: $^{15}\varepsilon_{app}$ = -34 ‰ (Berhanu et al., 2014;Meusinger et al., 2014) and -54 < $^{15}\varepsilon_{app}$ < -60 ‰ (Frey et al., 2009;Erbland et al., 2013), respectively.
The negative fractionation constant obtained from photolysis implies that the remaining $NO_3^-$ in the skin layer snow is enriched in $\delta^{15}N\text{-}NO_3^-$. In turn, the atmosphere is left with the source of $NO_x$ that is highly depleted in $\delta^{15}N\text{-}NO_3^-$. This enrichment (depletion) is exactly what we observe in the snow pack (atmosphere) at DML (Figs. 4 and 6). The marked difference in values from the evaporation experiments and those observed in snow at Dome C allows us to separate out the isotopic signature of evaporation and photolysis processes.

Assuming a Rayleigh-type single loss process, we calculate a $^{15}\varepsilon_{app}$ at DML of 12 ‰ using the active photic zone section of the snow pack (top 30 cm). We also calculate a $^{15}\varepsilon_{app}$ of -19 ‰ using our simulated results from the TRANSITS model. This simulated $^{15}\varepsilon_{app}$ nicely matches the expected $^{15}\varepsilon_{app}$ values (-59< $^{15}\varepsilon_{app}$< -16 ‰) within the "transition zone" of 5-20 cm yr$^{-1}$ (w.e.) modelled by Erbland et al. (2015). As the two loss processes of evaporation and photolysis have different isotopic fractionation signatures, an $^{15}\varepsilon_{app}$ of -19 ‰ cannot be explained by evaporation ($^{15}\varepsilon_{app}$ of 0) but rather photolysis albeit implying
a weaker photolytic loss of $NO_3^-$ than Dome C ($^{15}\varepsilon_{app}$ < -59 ‰) (Erbland et al., 2013). The discrepancy between our observed and simulated $^{15}\varepsilon_{app}$ is due to the larger snow accumulation rate, which preserves seasonality, and with a noisy signal, there is no pure separation of the loss processes assuming Rayleigh isotopic fractionation. The single-process Raleigh model does not work well at sites with annual signal in $\delta^{15}N\text{-}NO_3^-$.

Lastly, we estimate the potential snow emission flux of $NO_2$ ($F_{NO2}$) from $NO_3^-$ photolysis in snow using Eq. (8).

$$F_{NO2} = \int_{z=0\,m}^{z=1\,m} [NO_3^-]_z \, J(NO_3^-) dz \qquad\qquad\qquad \text{Eq. (8)}$$

where $J_z(NO_3^-)$ is the photolysis rate coefficient of reaction $NO_3^- + h\nu \rightarrow NO_2 + O^-$ at depth, z, in the snowpack, and is derived by scaling surface measurements (section 2.6) with e-folding depth (= 2-10 cm), and $[NO_3^-]_z$ is the amount of $NO_3^-$ per unit volume of snow at depth, z, in the snowpack. The calculated $F_{NO2}$ is a potential emission flux assuming that all $NO_3^-$ within the snow grain is photo-available, no cage effects are present and $NO_2$ is vented immediately after release from the snow grain
to the air above the snowpack without undergoing any secondary reactions. For the 1 to 14 January 2017 period, model estimates of $F_{NO2}$ scaled approximately linearly with e-folding depth were 0.4, 1.0 and 1.9 x 10$^{11}$ molecule m$^{-2}$ s$^{-1}$ for e-folding depths of 2, 5 and 10 cm, respectively. Spatial variability of $NO_3^-$ in the top 30 cm of surface snow at DML based on snow pit A and B is on the order of 13 % inducing similar variability in the model estimates of $F_{NO2}$. Estimates of $F_{NO2}$ at Dome C,





based on the same model during 1 to 14 January 2012, were larger with 1.2-7.3 x $10^{11}$ molecule m$^{-2}$ s$^{-1}$ (Frey et al., 2013),
mostly due to larger J(NO$_3^-$) values observed above the surface (section 2.6) as well as a larger e-folding depth (= 10 cm near
the surface). It should be noted that the observed F$_{NOx}$ was found to be up to 50 times larger than model estimates, which is
attributed to the poorly constrained quantum yield of NO$_3^-$ photolysis in natural snow (Frey et al., 2015;Frey et al., 2013). In
summary, the weakened air-snow recycling at DML is due to i) the shallower e-folding depth compared to Dome C which
implies reduced emission flux of NO$_x$, and ii) the reduced UV exposure time of surface snow due to higher annual accumulation
compared to Dome C. We estimate that NO$_3^-$ has a mean lifetime in the skin layer of 12 days to 3 years before it is photolysed
back to atmosphere.

### 4.3.3 Evidence for weaker recycling at DML

Only two studies have attempted to quantify the degree of NO$_3^-$ recycling between the air and snow (Davis et al., 2008;Erbland
et al., 2015). Erbland et al. (2015) use the TRANSITS model to estimate that NO$_3^-$ is recycled 4 times on average before burial
beneath the photic zone at Dome C, similar to the findings of Davis et al. (2008) for the same site. Using the approach of
Erbland et al. (2015), we find that NO$_3^-$ is recycled 3 times before it is archived at DML. A lower recycling factor than Dome
C is consistent with spatial patterns of NO$_3^-$ recycling factors across Antarctica reported by Zatko et al. (2016). As Dome C
and DML lie on the same latitude (75° S), incoming UV-radiation (except for cloud cover) should not impact the efficiency of
photolysis and thus recycling at the two sites. While photolysis-driven NO$_3^-$ recycling can occur at all polar sites, the most
intense enrichment of δ$^{15}$N-NO$_3^-$ in the depth profile is seen at Dome C and Vostok (Erbland, 2011). Below we provide some
explanations for the weakened recycling at DML.

  i.   Higher snow accumulation rate

The TRANSITS modelling shows the influence of the snow accumulation rate on the depth profile of NO$_3^-$ concentration and
δ$^{15}$N-NO$_3^-$, including the preservation of a seasonal cycle at higher snow accumulation rates (Fig. 7). At low accumulation
sites, i.e. Dome C, NO$_3^-$ in the skin layer and thinner snow layers is exposed to sunlight (and the actinic flux) for longer
allowing more photochemistry and thus a very active snow pack with strong NO$_3^-$ recycling and δ$^{15}$N-NO$_3^-$ enrichment. At
DML, which has a higher snow accumulation rate, the skin layer is buried more rapidly, leaving less time to adsorb additional
HNO$_3$ from the atmosphere and less time for photolysis to redistribute snow pack NO$_3^-$ to the overlying air for re-adsorption
to the skin layer. Following photolysis at DML, the recycling of NO$_3^-$ at the air snow interface alters the depth profile of δ$^{15}$N-
NO$_3^-$ in the top skin layer but below the skin layer δ$^{15}$N-NO$_3^-$ in snow remains intact as there is less redistribution and a lower
loss of NO$_3^-$ than at Dome C.

  ii.   Shallower e-folding depth

Based on measurements we derived an e-folding depth for DML ranging between 2 and 5 cm (Fig. S1). This estimate is similar
to a modelled value at South Pole (3.7 cm; Wolff et al. (2002) which has a similar accumulation rate, and Alert, Canada (5-6
cm; King and Simpson, 2001). The e-folding depth at Dome C is considerably deeper, ranging between 10 cm to 20 cm
depending on the snow properties (France et al., 2011). The e-folding depth depends on the density and grain size of snow





crystals, and the concentration of impurities. The larger e-folding depth at Dome C is due to the larger grain sizes and low impurity content. The impact of impurities in the range of observed polar snow concentrations on e-folding depth is small compared to the contribution from scattering by snow grains (France et al., 2011;Zatko et al., 2013). At Dome C, a larger e-

folding depth corresponds to a greater depth over which photochemistry occurs and thus stronger recycling and redistribution of $NO_3^-$. At DML, the lower e-folding depth of 2 to 5 cm lowers the mean $\delta^{15}N$-$NO_3^-$ value.

iii. Lower nitrate uptake at warmer temperatures

Temperature can control skin layer $NO_3^-$ uptake and loss. At colder snow temperatures, there is greater adsorption of $HNO_3$ to the skin layer (Abbatt, 1997;Jones et al., 2014). Although the difference in the mean annual temperature at Dome C compared

to DML (~5 °C) is not large enough to explain the significantly higher skin layer $NO_3^-$ mass concentrations there. Compounding this, $NO_3^-$ loss by evaporation is also dependent on temperature with maximum $NO_3^-$ loss at higher temperatures, i.e., diffusion of $NO_3^-$ in ice is slower at colder temperatures (Thibert and Domine, 1998). A compilation of $NO_3^-$ concentration data from Greenland and Antarctic ice cores showed that at very low accumulation rates lower temperatures lead to higher $NO_3^-$ mass concentrations preserved in the snow (Röthlisberger et al., 2000). Although the snow accumulation rate is closely linked to

temperature, photolysis is the dominant $NO_3^-$ loss process at low snow accumulation sites in Antarctica. Therefore, any differences in temperature between DML and Dome C could partly explain the greater uptake of $HNO_3$ to the skin layer, higher mass concentrations of $NO_3^-$ in the skin layer, and stronger recycling at Dome C compared to DML.

iv. Lower photolysis rate

At DML, $NO_3^-$ photolysis produces a lower $NO_x$ flux to the atmosphere and lower $^{15}\varepsilon_{app}$ highlighting that the photolysis rate

is lower thus the recycling strength is reduced (section 4.3.2). Furthermore, the large $^{15}\varepsilon_{app}$ associated with $NO_3^-$ photolysis has been determined for snow at Dome C (Berhanu et al., 2014;Frey et al., 2009;Erbland et al., 2013) and DML. At both sites, $\delta^{15}N$-$NO_3^-$ is enriched in the remaining skin layer snow. However, at DML, the $^{15}\varepsilon_{app}$ is lower due to less active photochemistry associated with a higher snow accumulation rate. Our results are consistent with Zatko et al. (2016) who suggest that the large fractionation constant associated with photolysis is greatest on the polar plateau where strong winds are most efficient at

exporting $NO_3^-$ away from the site.

v. Lower export of locally produced nitrate

Export of locally produced $NO_x$ on the Antarctic Plateau leads to greater enrichment in the depth profiles of $\delta^{15}N$-$NO_3^-$ relative to the coast (Savarino et al., 2007;Zatko et al., 2016). Zatko et al. (2016) modelled the export of snow sourced $NO_x$ away from the original site of $NO_3^-$ photolysis, and found that the largest loss of $NO_3^-$ occurs in central Antarctica where most $NO_3^-$ is

transported away by katabatic winds. At the coast, photolysis driven loss of $NO_3^-$ from the snow is minimal due to high snow accumulation rates. Here, observations of enriched atmospheric $\delta^{15}N$-$NO_3^-$ show that $NO_x$ has been transported away from the location of its production on the Antarctic Plateau to the coast (Savarino et al., 2007;Morin et al., 2009). The greater export of $NO_3^-$ from Dome C allows efficient removal of recycled $NO_3^-$ from that site, resulting in a lower archived $NO_3^-$ mass flux and enriched $\delta^{15}N$-$NO_3^-$ signature in the surface snow. The enrichment of $\delta^{15}N$-$NO_3^-$ is due to the isotopic mass balance rather than

an increase for photolysis intensity. With less export of $NO_3^-$ away from the DML site, locally sourced $NO_x$ is redeposited back



to the skin layer at the site of the emission and the depth profile of the $\delta^{15}$N-NO$_3^-$ is not as dramatically impacted as Dome C where there is substantial loss of NO$_3^-$. Therefore, the degree of NO$_3^-$ recycling is also determined by the transport patterns across Antarctica.

Based on field, laboratory and theory, we conclude that NO$_3^-$ photolysis is the dominant post-depositional process on the
Antarctic plateau controlling NO$_3^-$ mass concentrations and $\delta^{15}$N-NO$_3^-$ values in the snow and atmosphere. Nitrate photolysis in snow causes $\delta^{15}$N-NO$_3^-$ fractionation of the magnitude needed to explain field and lab observations. The development of TRANSITS allows us to model the archived $\delta^{15}$N-NO$_3^-$ values taking into account all parameters in the air-snow system.

### 4.4 Preservation and archival

The photolysis-driven recycling of NO$_3^-$ is largely dependent on the time that NO$_3^-$ remains in the snow photic zone. Post-
depositional loss of NO$_3^-$ at DML was quantified in a number of firn cores and snow pits by Weller et al. (2004) who found that ~26 % of the NO$_3^-$ originally deposited to the snow pack was lost. The e-folding time for NO$_3^-$ at the site was reported as ~20 years, and NO$_3^-$ was archived after 5 to 6 years of deposition (or 1.1 – 1.4 m depth) which is the time it takes for the NO$_3^-$ mean concentration to become representative of the last 100 years. At this point, the authors considered post-depositional loss of NO$_3^-$ to be negligible, and therefore archived. However, no skin layer measurements were made in the study and given how
active the skin layer is NO$_3^-$ redistribution and recycling, we use our skin layer measurements to provide new constraints on the archival values and time of NO$_3^-$ at DML.

Taking the high skin layer NO$_3^-$ mass concentrations into account (average of 230 ng g$^{-1}$ in January for DML), we calculate a NO$_3^-$ loss of 60 ng g$^{-1}$ (or 75 %) and enrichment of 170 ‰ from the snow pack. Assuming all NO$_3^-$ is archived below the photic zone, i.e., an e-folding depth of 5 cm, archival occurs below a depth of 15 cm, where NO$_3^-$ has a residence time of 0.75 years
in the photic zone corresponding to one summer. At this point, the amplitude of the annual cycle of $\delta^{15}$N-NO$_3^-$ at DML does not vary. Our archived values of 50 ‰ and 60 ng g$^{-1}$ agree well with the mean values of the snow pit below the photic zone (30 cm), and are lower than the simulated archived values from TRANSITS (120 ng g$^{-1}$ and 130 ‰) due to the stronger photochemistry in the model. Due to the larger e-folding depth and hence larger photic zone at Dome C, NO$_3^-$ has a longer residence time of 3 years (3 summers) in the photic zone. Here, archival of NO$_3^-$ occurs below a depth of 30 cm. Compared to
Dome C, the archived values at DML have a similar concentration (Dome C: 35 ng g$^{-1}$) but lower $\delta^{15}$N-NO$_3^-$ value (Dome C: 300 ‰), due to the thicker photic zone, stronger redistribution and recycling there.

### 4.5 Sensitivity of $\delta^{15}$N-NO$_3^-$ to deposition parameters and implications for interpreting ice core records of $\delta^{15}$N-NO$_3^-$ at DML

As first proposed by Frey et al. (2009) and later confirmed by field and lab studies (Erbland et al., 2015;Berhanu et al., 2014;Shi
et al., 2019) it is UV-photolysis of NO$_3^-$ that dominates post-depositional fractionation of $\delta^{15}$N-NO$_3^-$ in snow and firn. Yet the extent of photolytic fractionation and the $\delta^{15}$N-NO$_3^-$ ultimately preserved in firn and ice depends on the UV-spectrum of down-welling irradiance, on the time snow layers are exposed to incoming UV-radiation as well as on the snow optical properties.



Previous studies showed that $\delta^{15}N\text{-}NO_3^-$ is sensitive to TCO but also to deposition parameters such as the annual accumulation rate (Shi et al., 2018;Noro et al., 2018;Erbland et al., 2013). Thus, if all deposition parameters remained constant or are well-constrained it should be theoretically possible to use $\delta^{15}N\text{-}NO_3^-$ as an ice core proxy for past surface UV-radiation and stratospheric ozone. Understanding the depositional parameters and their impact on $\delta^{15}N\text{-}NO_3^-$ is paramount for the interpretation of $\delta^{15}N\text{-}NO_3^-$ signals preserved in ice cores. As the interpretation of $\delta^{15}N\text{-}NO_3^-$ is site-specific, we investigate the sensitivity of the $\delta^{15}N\text{-}NO_3^-$ signature at DML to snow accumulation rate, e-folding depth and TCO. As the mean annual snow accumulation rate at DML is 6 cm (w.e.) $yr^{-1}$, we take this simulation as our base case.

### 4.5.1 Sensitivity of the ice core $\delta^{15}N\text{-}NO_3^-$ signal to accumulation rate

The $\delta^{15}N\text{-}NO_3^-$ signal is indeed sensitive to the snow accumulation rate at DML. Here, the accumulation rate varied between 2.5 and 11 cm $yr^{-1}$ (w.e.) over the last 1000 years (Sommer et al., 2000). Figs. 7a-b shows the potential impact of this variability in the snow accumulation rate on the $NO_3^-$ concentration and $\delta^{15}N\text{-}NO_3^-$ signature at DML calculated with the TRANSITS model. Considering that the actual snow accumulation rate varied between 3.5 and 7.1 cm $yr^{-1}$ (w.e.) in our snow pit, our $\delta^{15}N\text{-}NO_3^-$ measurements fall within the simulated $\delta^{15}N\text{-}NO_3^-$ depth profile for the accumulation rates over the past 1000 years. Although the mean snow pit $\delta^{15}N\text{-}NO_3^-$ is ~50 ‰ lower, the snow pit depth profile parallels the base case profile for the top 30 cm. Here, there is a clear enrichment of $\delta^{15}N\text{-}NO_3^-$ in both the snow pit and base case profiles corresponding to the depth of the photic zone (30 cm), and demonstrating that $NO_3^-$ photolysis is taking place in this section of the snow pack. Below the photic zone, the seasonal variability of the base case $\delta^{15}N\text{-}NO_3^-$ depth profile is constant between 100-153 ‰ indicating that no further enrichment or $NO_3^-$ redistribution is taking place in the archived section of the snow pack.

Despite the relatively high $NO_3^-$ mass concentrations and enriched $\delta^{15}N\text{-}NO_3^-$ in the skin layer at DML, clear seasonal cycles remain in the depth profile in contrast to the lower snow accumulation site of Dome C where the depth profile is relatively constant below the photic zone. Figs. 7a-b indicate that at higher snow accumulation rates, the seasonality of atmospheric $NO_3^-$ and $\delta^{15}N\text{-}NO_3^-$ is preserved due to faster burial. Even at 6 cm $yr^{-1}$ (w.e.), the snow layers remain in the active photic zone for 0.75 years and the weaker recycling factor is low enough to conserve the seasonality. Whereas at Dome C, snow layers remain within the photic zone for longer (about 3 years), and $NO_3^-$ loss and redistribution continues until the seasonal cycle becomes smoothed (Figs. 7a-b). Thus, $NO_3^-$ recycling is strongest in the lowermost snow accumulation regions.

Below the active photic zone, there is an offset between the base case and snow pit $\delta^{15}N\text{-}NO_3^-$ depth profile in terms of i) the amplitude of the summer and winter $\delta^{15}N\text{-}NO_3^-$ values, and ii) the mean $\delta^{15}N\text{-}NO_3^-$ value (Fig. 7). To account for this offset we investigated how the timing of snow deposition altered the $\delta^{15}N\text{-}NO_3^-$ depth profile. Rather than assuming a constant accumulation rate of 6 cm $yr^{-1}$ (w.e.), as in the base case, we find that a variable snow accumulation rate, based on our observations from the snow pit, alters the depth of the summer and winter $\delta^{15}N\text{-}NO_3^-$ peaks (Fig. 7b.). Using the actual annual accumulation rates improves the model fit (~10 cm depth; Fig 7a). Furthermore, the timing of the snow accumulation throughout the year has a significant control on the amplitude of the seasonal $\delta^{15}N\text{-}NO_3^-$ cycle. Snowfall at DML has a bimodal distribution with higher accumulation in austral autumn and early austral summer (Fig. S9). In Fig. 7c, we modified the timing



of the snow accumulation during the year by depositing 90 % of the annual snowfall in i) the first week of winter, and ii) the first week of summer, which represents the upper bound for snow accumulation in winter and summer respectively. The remaining 10 % of the annual snowfall is distributed evenly across the rest of the weeks of the year. Summer snow accumulation results in a higher $\delta^{15}N$-$NO_3^-$ enrichment compared to winter snow accumulation, as the exposure of summer

layers to UV is longer and thus $NO_3^-$ photolysis is stronger. Therefore, the timing and rate of snowfall can explain the misalignment between snow pit observations and base case simulation, which shifts the depth and amplitude of the $\delta^{15}N$-$NO_3^-$ peaks in the depth profile.

On centennial to millennial timescales, the snow accumulation rate has varied in regions of Antarctica (Thomas et al., 2017), which could potentially modify the degree of post-depositional processing and thus impact the archival and temporal variability

of $\delta^{15}N$-$NO_3^-$ in ice cores. Interestingly, Geng et al. (2015) found that post-depositional loss of $NO_3^-$ in Greenland could fully account for the large difference between the glacial and Holocene $\delta^{15}N$-$NO_3^-$ signature. At DML, higher snow accumulation rates would result in lower $NO_3^-$ mass concentrations and more depleted $\delta^{15}N$-$NO_3^-$ values in the skin layer, thus reducing the recycling strength and lowering the sensitivity of the UV proxy recorded in the ice over time, and vice versa. TRANSITS modelling predicts that the upper and lower bounds of $\delta^{15}N$-$NO_3^-$ values in a 1000-year ice core from DML that has an

accumulation rate between 2.5 and 11 cm yr$^{-1}$ (w.e.) to be 70 - 360 ‰. Furthermore, $\delta^{15}N$-$NO_3^-$ values could range between 90-110 ‰ depending the timing of snowfall and extreme precipitation events, which are known to play a dominant role in snowfall variability across Antarctica (Turner et al., 2019). At DML, snow pit observations suggest that the variation of $\delta^{15}N$-$NO_3^-$ between the polar day and polar night is 20 ‰. This seasonality is within the range of values expected for changes in snow accumulation rates over time (Fig. 7). Therefore, any seasonal variation in ice core $\delta^{15}N$-$NO_3^-$ will need to be accounted

for in order to observe decadal, centennial and millennial scale trends in $\delta^{15}N$-$NO_3^-$.

### 4.5.2 Sensitivity of the ice core $\delta^{15}N$-$NO_3^-$ signal to e-folding depth

We measured an e-folding depth at DML (2-5 cm) which is lower than that employed in the TRANSITS model (10 cm). Furthermore, a range of e-folding depth values, between 3.7 and 20 cm, have been reported for Antarctica. The positive bias of the TRANSITS simulation in archived $\delta^{15}N$-$NO_3^-$ at DML may be due to e-folding depth being smaller than at Dome C as

indicated by direct observations. In order to test this assumption, the sensitivity of archived $\delta^{15}N$-$NO_3^-$ to the parameter e-folding depth needs to be quantified, which has not been done before as far as we know. Zatko et al. (2016) modelled the e-folding depth over Antarctica and investigated the impact of snow-sourced $NO_x$ fluxes but not on $\delta^{15}N$-$NO_3^-$. The e-folding depth has a large influence on the $\delta^{15}N$-$NO_3^-$ depth profile in terms of i) depth of the photic zone and thus depth of the $\delta^{15}N$-$NO_3^-$ enrichment, and ii) the mean archived $\delta^{15}N$-$NO_3^-$ value below the photic zone (Fig. 7d). A larger e-folding depth

strengthens the $\delta^{15}N$-$NO_3^-$ enrichment in the photic zone and archived mean $\delta^{15}N$-$NO_3^-$ value. For example, an e-folding depth of 10 cm at DML gives $\delta^{15}N$-$NO_3^-$ enrichment down to 25 cm and an archived mean $\delta^{15}N$-$NO_3^-$ value of 125 ‰ in the snow pack compared to an e-folding depth of 20 cm, which enriches the snow pack to 45 cm and more than doubles the archived mean $\delta^{15}N$-$NO_3^-$ value to 320 ‰. Meanwhile, an e-folding depth of 2 cm gives minimal enrichment and a low archived mean





$\delta^{15}$N-NO$_3^-$ value of 25 ‰. In comparison to the base case simulation, which has an e-folding depth of 10 cm, a lower e-folding
depth of 5 cm decreases the archived mean $\delta^{15}$N-NO$_3^-$ in the snow pack to ~50 ‰, closely matching our snow pit observations.
Hence, a shallower e-folding depth of 5 cm can explain the more depleted $\delta^{15}$N-NO$_3^-$ snow pit profile, relative to the base case
simulation, as NO$_3^-$ photolysis occurs in a shallower depth. Therefore, e-folding depth knowledge is required to understand
the sensitivity of archived $\delta^{15}$N-NO$_3^-$ at specific sites. A lower e-folding depth and variable snowfall throughout the year can
explain the misalignment between the snow pit observations and simulated $\delta^{15}$N-NO$_3^-$ depth profiles.

**4.5.3 Sensitivity of ice core $\delta^{15}$N-NO$_3^-$ signal to TCO**

Fig. 8 shows the strong sensitivity of $\delta^{15}$N-NO$_3^-$ to variations in decreasing TCO. A decrease in TCO will increase UV radiation
reaching the surface at an ice core site. As a result, stronger photolysis enhances NO$_3^-$ loss, redistribution and recycling from
the snow pack and ultimately decreases the archived NO$_3^-$ concentration. Furthermore, a decrease in TCO enriches the $\delta^{15}$N-
NO$_3^-$ signature as the snow is exposed to a greater UV dose. We expect that a change of 100 Dobson Units (DU), i.e. the
amount that ozone now decreases each spring as a result of stratospheric ozone destruction processes, will result in a 22 ‰
change in $\delta^{15}$N-NO$_3^-$ at DML. The variability in $\delta^{15}$N-NO$_3^-$ induced by TCO is similar to the seasonal variability of $\delta^{15}$N-NO$_3^-$
recorded in the snow pit (20 ‰) and less than the predicted variability of $\delta^{15}$N-NO$_3^-$ due to variability in snow accumulation
(340 ‰), thus the development of a large ozone hole is unlikely to be observed above the natural background $\delta^{15}$N-NO$_3^-$
variability in the ice core at this site. The sensitivity of $\delta^{15}$N-NO$_3^-$ to TCO is greater at Dome C than DML.

**4.5.4 Implications for interpreting ice core $\delta^{15}$N-NO$_3^-$**

Site-specific air-snow transfer studies provide an understanding of the mechanisms that archive $\delta^{15}$N-NO$_3^-$ in ice cores, thus
allowing for the interpretation of longer records of $\delta^{15}$N-NO$_3^-$ from the site. Ice core records of archived NO$_3^-$ mass
concentrations and $\delta^{15}$N-NO$_3^-$ at DML are a result of three uptake and loss cycles that occur in the top 30 cm during sunlit
conditions. While we do not observe further redistribution of NO$_3^-$ in layers deeper than the photic zone, we cannot rule out
any further NO$_3^-$ diffusion within the firn or ice sections of an ice core. This redistribution unlikely results in a loss of NO$_3^-$
but could migrate NO$_3^-$ to different layers, for example in acidic layers around volcanic horizons (Wolff, 1995).

There are a number of factors that will control the variability of the archived $\delta^{15}$N-NO$_3^-$ signature in ice cores recovered from
DML. The $\delta^{15}$N-NO$_3^-$ signature in the snow pack is most sensitive to changes in the snow accumulation rate and e-folding
depth, with snowfall timing and TCO also playing a smaller role. The snow accumulation rate and e-folding depth could
influence the archived $\delta^{15}$N-NO$_3^-$ composition by up to 300 % over the last 1000-years. This magnitude is comparable to
modelled enrichment in ice-core $\delta^{15}$N-NO$_3^-$ (0 to 363 ‰) due photolysis-driven loss of NO$_3^-$ at low accumulation sites in
Antarctica by Zatko et al. (2016). While the timing of snowfall and changes in TCO will have a smaller impact of 20 ‰ on
archived $\delta^{15}$N-NO$_3^-$. Ice core $\delta^{15}$N-NO$_3^-$ records at DML will be less sensitive to changes in UV than those at Dome C (Fig.
8), however the higher snow accumulation rate and more accurate dating at DML allows for higher resolution $\delta^{15}$N-NO$_3^-$
records. We acknowledge that in addition, other factors such as light absorbing impurities (Geng et al., 2015), local




meteorology, source of emissions and transport of $NO_x$ and $NO_3^-$, atmospheric oxidant concentrations, and polar $NO_3^-$ formation can influence the rate of recycling and export of snow sourced $NO_x$. We discussed above that atmospheric $\delta^{15}N$-$NO_3^-$ values are unlikely to be influenced or sourced from snow exported up wind from the polar plateau due to the local meteorology at DML at least for the duration of the campaign. Yet these factors may have changed over time.

Given a variable accumulation rate and smaller e-folding depth, which we provide evidence for at DML, the TRANSITS model is able to reproduce our snow pit observations, justifying our previous assumption that photolysis is the main driver of $NO_3^-$ post-depositional processes at DML. In fact, TRANSITS does such a good job at simulating $NO_3^-$ recycling in Antarctica that we recommend that this tool is employed before the commencement of future ice core $\delta^{15}N$-$NO_3^-$ studies to understand the sensitivity of the signal to various factors. Taking changes snow accumulation into account, it may be possible to reconstruct

past UV and TCO from the $\delta^{15}N$-$NO_3^-$ signal in DML ice cores provided other factors such as the e-folding depth have remained the same.

**5 Conclusions**

Our key findings are:

- Isotopes are a powerful tool for unpicking post-depositional processes affecting ice core signals of $NO_3^-$ at low
accumulation sites;
- At DML, post-depositional loss of $NO_3^-$ is controlled predominantly by photolytic loss;
- Photolysis redistributes $NO_3^-$ between the snow pack and atmosphere resulting an enrichment of $\delta^{15}N$-$NO_3^-$ in the skin layer;
- TRANSITS, a photolysis driven model, modelling suggests that $NO_3^-$ is recycled three times before it is archived in
the snow pack below 15 cm and within 0.75 years;
- Once archived, the seasonal variability of $\delta^{15}N$-$NO_3^-$ values and $NO_3^-$ mass concentrations oscillate between -1 to 80 ‰ and 30 to 80 ng $g^{-1}$, respectively;
- TRANSITS can explain the observed snow depth profiles of $\delta^{15}N$-$NO_3^-$ constrained by an e-folding depth of 5 cm, the observed snow accumulation rate, and variable snowfall timing.
- TRANSITS sensitivity analysis showed that the $\delta^{15}N$-$NO_3^-$ signature in the snow pack is most sensitive to changes in the snow accumulation rate (up to 300 %) and e-folding depth (up to 300 %), with snowfall timing (~20 %) and total column ozone (~20 %) also playing a smaller role;
- Constraints on e-folding depth are critical for calculating photolytic loss of snow pack $NO_3^-$ and for interpreting $\delta^{15}N$-$NO_3^-$ preserved in ice cores;
- Additional studies of e-folding depth across a range of Antarctic sites would help determine key factors influencing this parameter;



- The $NO_3^-$ recycling process at DML is weaker than Dome C, largely because of the higher snow accumulation rate and lower e-folding depth;

- TRANSITS has now been tested at two sites in Antarctica, namely DML and Dome C, and we recommend applying this model to new ice core sites to understand the sensitivity of the $\delta^{15}N$-$NO_3^-$ signal before embarking on new ice core projects;

- By accounting for variability in the snow accumulation rate and assuming a constant e-folding depth, it may be possible to reconstruct past UV-radiation at ice core sites with very a low accumulation rate and low accumulation variability, as low accumulation variability will have little effect on $\delta^{15}N$-$NO_3^-$ in comparison to the UV dose reaching

ground.

**Acknowledgments**

This project was funded by a National Environment Research Council (NERC) Standard Grant (NE/N011813/1) to M.F. V.H.L.W would like to thank the University of Cambridge Doctoral Training Program (DTP) for funding a NERC Research Experience Project (REP) that contributed to this manuscript. We would like to thank British Antarctic Survey (BAS) and

Alfred Wegener Institute (AWI) staff for their field and logistics support at Halley Station and Kohnen Station, respectively. Technical support for nitrate isotope analysis at the Institut des Géosciences de l'Environnement (IGE), Grenoble was provided by Joris Leglise, Ines Ollivier and Ilan Bourgeois. We thank Joseph Erbland for providing the TRANSITS model. Field samples collected at Dome C was possible through the program SUNITEDC/CAPOXI (grant 1011/1177) funded by the Institut Polaire Français IPEV. J.S and N.C thank the ANR (Investissements d'avenir ANR-15-IDEX-02 and EAIIST grant ANR—16-CE01-

0011-01) and the INSU program LEFE-CHAT for supporting the stable isotope laboratory. This is publication 1 of PANDA platform on which isotope analysis were performed. PANDA was partially funded by the LabEx OSUG@2020 (ANR10 LABX56). All winter over personal who collected the year-round Dome C samples in extreme conditions, years after years, are deeply acknowledged. In addition, we thank Emily Ludlow, Shaun Miller, Catriona Sinclair, Rebecca Tuckwell, and Neil Brough for technical support at BAS. Thanks to James France for discussions around the e-folding depth measurements and

interpretation, and to John Turner for discussions of the local meteorology. We acknowledge Utrecht University who supplied the AWS data for AWS9 at DML05/Kohnen (https://www.projects.science.uu.nl/iceclimate/aws/files_oper/oper_20632), and the precipitation data from the RACMO2 model (https://doi.org/10/c2pv). We would like to thank Bodeker Scientific, funded by the New Zealand Deep South National Science Challenge, for providing the combined NIWA-BS total column ozone database. Wind roses were plotted using the openair package in R. The data set for the DML nitrate isotopic ratios and nitrate

mass concentrations in aerosol, skin layer and snow pits is available through the Polar Data Centre https://doi.org/10.5285/1467b446-54eb-45c1-8a31-f4af21e60e60, and supporting data are also included as figures and tables in the supplement.



## Author contributions

V.H.L.W, J.S and M.F designed the research. V.H.L.W, M.F and J.S, N.C collected samples at DML and Dome C, respectively. V.H.L.W analysed the major ion data. V.H.L.W, L.H, and N.C analysed the nitrate isotope data. A.M and V.H.L.W designed the TRANSITS experiments. A.M performed the TRANSITS experiments. M.F did e-folding depth and flux calculations. V.H.L.W prepared the manuscript with contributions from all co-authors.

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



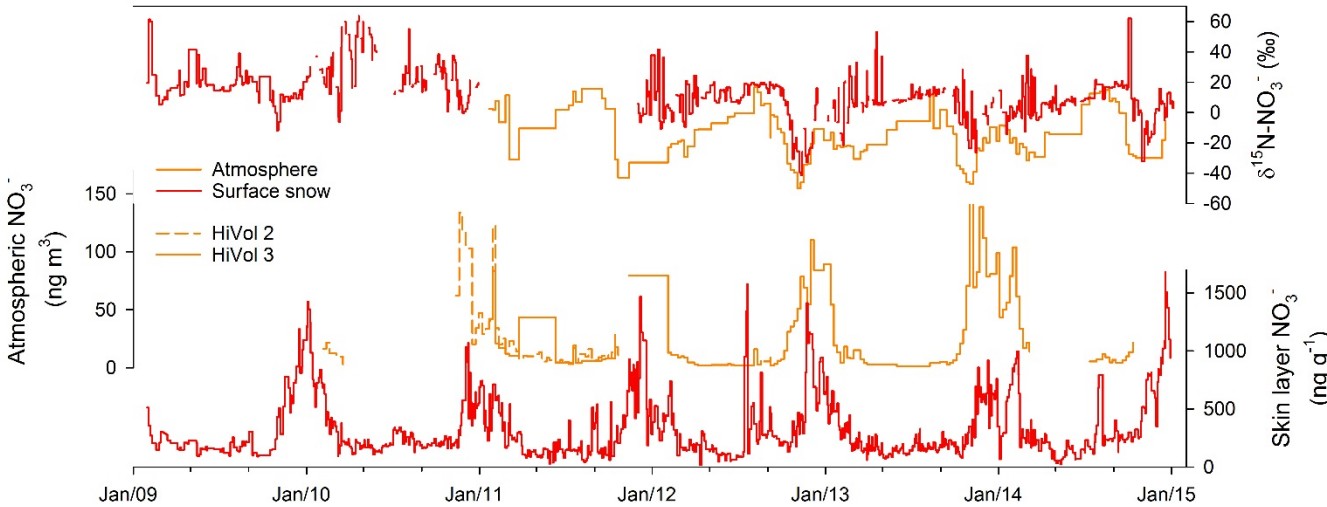

**Figure 1: Year round atmospheric and skin layer NO₃⁻ concentration and δ¹⁵N-NO₃⁻ at Dome C. Data source: years 2009-2010: Erbland et al. (2013); 2011-2015: this study.**




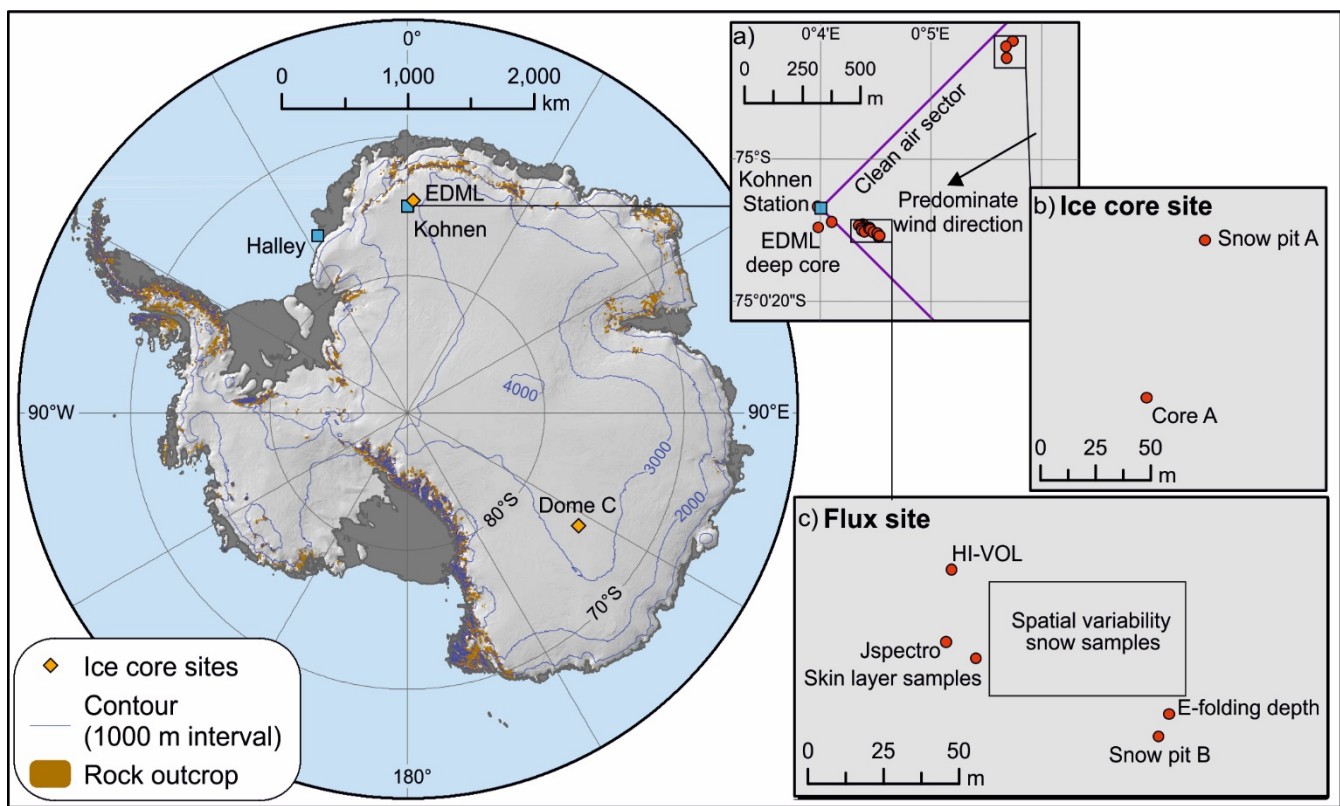

**Figure 2: Map of ISOL-ICE ice core drilling and atmospheric campaign, and ice core sites and Antarctica stations mentioned in this study. a) Insert of Kohnen Station in Droning Maud Land (DML) highlighting the predominate wind direction, deep EDML ice core site and the ISOL-ICE "flux" and "ice cores" sites, b) ISOL-ICE "ice core site" showing ice core, firn core and snow pit A locations, and c) ISOL-ICE "flux" site showing location of *in situ* atmospheric instruments, surface snow, snow pit and aerosol sampling locations and e-folding depth measurements.**





**Figure 3: January 2017 time series at Dronning Maud Land (DML) of a) precipitation, b) wind direction and wind speed, c) atmospheric and skin layer δ¹⁵N-NO₃⁻, and d) atmospheric and skin layer NO₃⁻ concentration. Meteorological data source: University of Utrecht (AWS9; DML05/Kohnen; 75°00'S, 00°00' E/W; ~2900 m.a.s.l.). Precipitation data source: RACMO2 (https://doi.org/10/c2pv).**









**Figure 4: Comparison of NO₃⁻ concentration and δ¹⁵N-NO₃⁻ at Dronning Maud Land (DML) and Dome C in January 2017. NO₃⁻ concentration in a) atmosphere, b) skin layer, and c) depth profiles. Insert: Depth profile of NO₃⁻ concentration highlighting seasonal variability. δ¹⁵N-NO₃⁻ in d) atmosphere, e) skin layer, and f) depth profiles.**



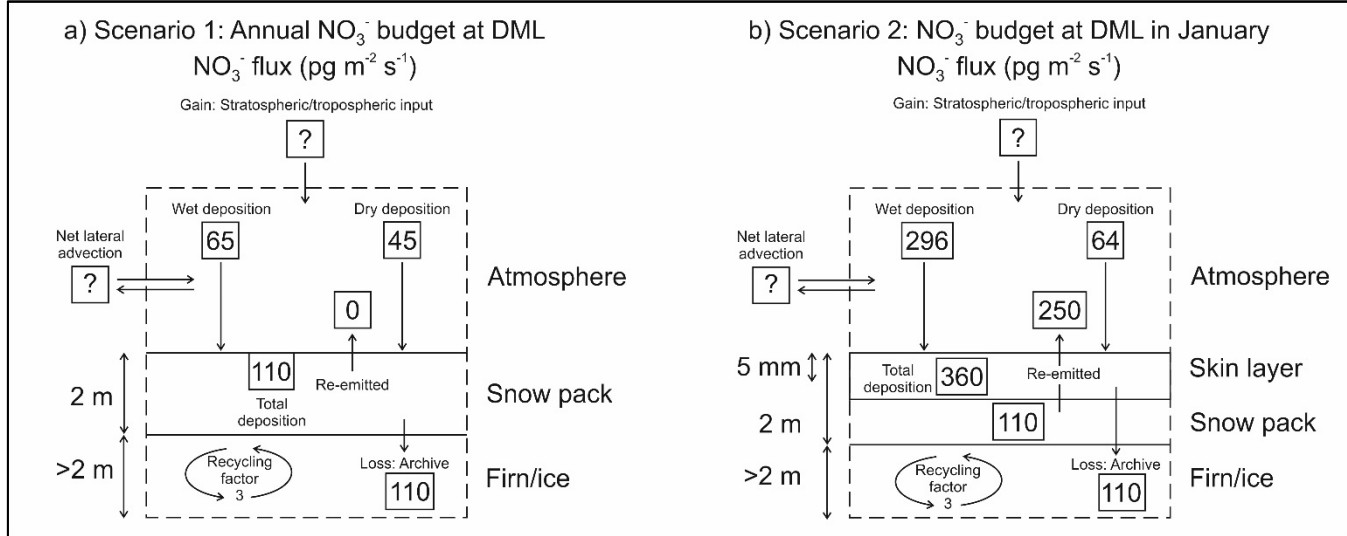

**Figure 5: Schematic of NO₃⁻ mass fluxes at Dronning Maud Land (DML) for a) annual mean scenario and b) January scenario.**



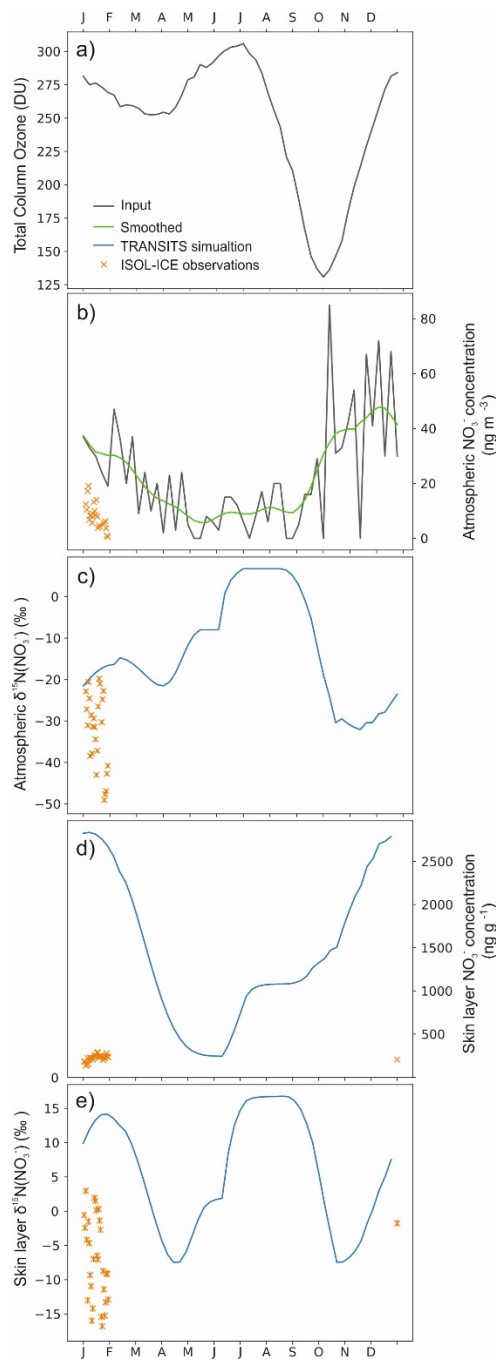


**Figure 6: ISOL-ICE observations and simulated annual cycle of skin layer and atmospheric NO₃⁻ concentration and δ¹⁵N-NO₃⁻ at Dronning Maud Land (DML) from the TRANSITS model for January 2017. a) Total column ozone: NIWA Bodeker combined dataset version 3.3 at DML averaged from 2000 to 2016 (http://www.bodekerscientific.com/data/total-column-ozone). b) Atmospheric NO₃⁻ concentration data are observations from Kohnen Station (Weller and Wagenbach, 2007) that are used as input into the model. ISOL-ICE observations and TRANISTS simulations of c) atmospheric δ¹⁵N-NO₃⁻, d) skin layer NO₃⁻ concentration and e) skin layer δ¹⁵N-NO₃⁻.**






**Figure 7: Snow pit depth profiles of observations and simulations from TRANSITS. Sensitivity of a) $\delta^{15}$N-NO$_3^-$ and b) NO$_3^-$ concentration to the upper and lower bounds of accumulation rates observed over the last thousand years at Dronning Maud Land (DML). Also shown are our snow pit observations, and the depth profiles of the simulated $\delta^{15}$N-NO$_3^-$ values and NO$_3^-$ concentration using the observed accumulation rate in our snow pits. Sensitivity of c) $\delta^{15}$N-NO$_3^-$ to the timing of snow accumulation, d) $\delta^{15}$N-NO$_3^-$ to the e-folding depth. Blue is the base case simulation, in which we refer to throughout the study. Note that panels a-b) have the same legend, and the nominal date refers to the base case simulation.**




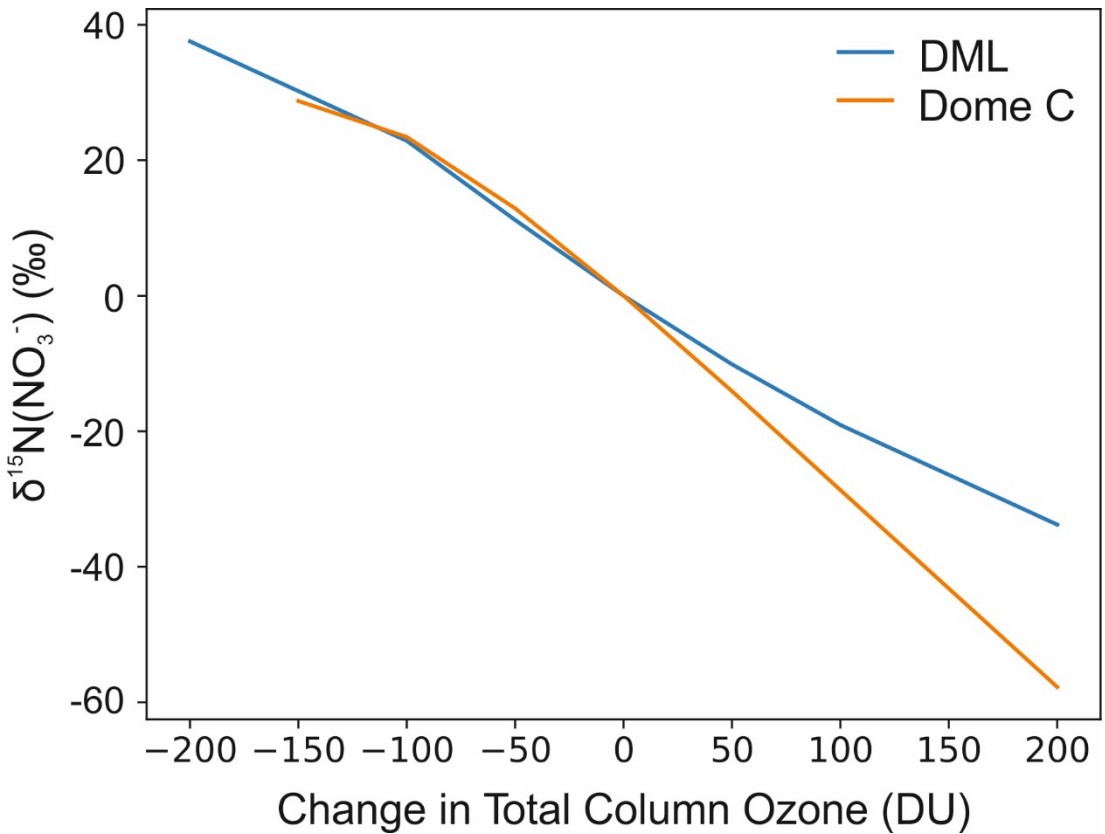

**Figure 8: Expected response of archived δ¹⁵N-NO₃⁻ to changes in total column ozone at Dronning Maud Land (DML) and Dome C. Archived DML δ¹⁵N-NO₃⁻ values were simulated using a fixed accumulation rate of 6 cm yr⁻¹ (w.e.) and e-folding depth of 10 cm. Dome C data source: Erbland et al. (2015).**





**Table 1: Site characteristics of Dronning Maud Land (DML) and Dome C ice core sites.**

|  | DML | Dome C |
|---|---|---|
| Latitude (°S) | 75 | 75 |
| Elevation (m a.s.l.) | 2892 | 3233 |
| Distance from the coast (km) | 550 | 900 |
| Mean snow accumulation (cm y⁻¹; w.e.) | [1]6 | [2]2.5 |
| Predominate wind direction (°) | 45 | 180-200 |
| e-folding depth (cm) | [3]2-5 | [4]10-20 |
| Average January nitrate concentration in skin layer (ng g⁻¹) | [3]230 | [3]600 |
| Average annual nitrate concentration in firn (ng g⁻¹) | [3]60 | [5]50 |
| Average January nitrate concentration in atmosphere (ng m⁻³) | [3]10 | [3]60 |

[1]Sommer et al. (2000);Hofstede et al. (2004)

[2]Le Meur et al. (2018)

[3]This study

[4]France et al. (2011)

[5]Frey et al. (2009)





**Table 2: Summary of observed and simulated archived, aerosol and skin layer NO₃⁻ mass concentrations and δ¹⁵N-NO₃⁻ composition at Dronning Maud Land (DML) and Dome C. n.d.: no data.**

| Archived (<30 cm) | NO₃⁻ (ng g⁻¹) | δ¹⁵N-NO₃⁻ (‰) | Flux (pg m⁻² s⁻¹) | Reference |
|---|---|---|---|---|
| DML Pit A | 60 | 50 | 110 | This study |
| DML Pit B | 50 | n.d. | 120 | This study |
| DML TRANSITS | 120 | 130 | 210 | This study |
| *DML expected | 100 | 100 | 140 | Erbland et al. (2015);Erbland et al. (2013) |
| Dome C | 50 | 280 | <140 | Erbland et al. (2013) |
| **Aerosol (January mean)** | **NO₃⁻ (ng m⁻²)** | **δ¹⁵N-NO₃⁻ (‰)** | **Flux (pg m⁻² s⁻¹)** | **Reference** |
| DML | 10 | -30 | 70 | This study |
| DML TRANSITS | 30 | -20 | 190 | This study; Weller and Wagenbach (2007) |
| Dome C | 60 | -10 | 90 | This study; Erbland et al. (2013) |
| **Skin layer (January mean)** | **NO₃⁻ (ng g⁻¹)** | **δ¹⁵N-NO₃⁻ (‰)** | **Flux (pg m⁻² s⁻¹)** | **Reference** |
| DML | 230 | -10 | 360 | This study |
| DML TRANSITS | 2800 | 10 | 4800 | This study |
| Dome C | 590 | 10 | 470 | This study; Erbland et al. (2013) |

*Expected values for a site with an accumulation rate of 6 cm yr⁻¹ (w.e.) based on the spatial transect of (Erbland et al., 2015).