# Peer review of "Deposition, recycling and archival of nitrate stable isotopes between the air-snow interface: comparison between Dronning Maud Land and Dome C, Antarctica"

_Atmospheric Chemistry and Physics, 2019_

## Referee Comment (RC1) · Anonymous Referee #1 · 5 Oct 2019

Winton et al report observations of the nitrogen isotopic composition of nitrate in Antarctic (DML) in the atmosphere, snow "skin layer", and depth profiles in snow pits. They use these observations combined with a snow chemistry model to understand what controls the variability in the nitrogen isotopes and nitrate concentrations and to assess the utility of such observations in ice cores as a proxy for past surface UV radiation. They conclude that although photolytic driven recycling and loss of reactive nitrogen is what determines the nitrate isotopes and abundances at this location, that variability in parameters such as snow accumulation rate have a large enough impact on the nitrogen isotopes so as to preclude the influence of variability in surface UV in determining the nitrogen isotopic composition of snow nitrate.

The observations and model-based interpretation are robust and important as it would be useful to have a paleo-UV proxy. As the authors state, it is important to assess the sensitivity of nitrogen isotopes at particular locations in order to determine its usefulness as a proxy for any given process. Unfortunately, the paper is frustrating to read because it is it so repetitive and spends so much time on introductory material throughout the manuscript that it is hard to find the actual interpretation of the results. It is as if the authors are afraid to state their interpretation. I suggest moving (and condensing) the introductory material that is spread throughout the manuscript (especially in section 4) to the introduction and making sure that the actual interpretation is presented up front instead of hidden. Because of this, it's sometimes hard to determine if the results support their conclusions. The paper as is reads as a first or second draft, not a final manuscript ready for submission for publication. This is particularly evident in the conclusions, which is not even written in paragraph form.

Additional comments:

I find the concept of the number of times nitrate is recycled difficult to wrap my head around. Does this mean that each molecule of nitrate is recycled on average three times before being archived? Is this averaged over the depth of the photic zone? I would imagine some molecules (like those that are close to the surface in summer) are recycled more than others, so that there is actually some distribution of recycling events on a molecule per molecule basis so as to average around the number 3. Is this interpretation correct? I suggest elaborating on this in the methods section.

Second paragraph of the introduction: It seems strange to say that the primary sources of nitrate are transport of nitrate from the stratosphere and transport of alkyl nitrates in the troposphere without mentioning transport of inorganic nitrate in the troposphere. Where is the evidence that inorganic nitrate is not transported in the troposphere to Antarctica? A model study suggests that it is certainly possible (Lee et al., 2014). Also, soil NOx should be mentioned as a NOx source in the troposphere.

Page 2 lines 61 – 62: Reprase to: "Model results from Zatko et al. (2016) suggest that…"

Page 4 line 116: Are you referring to skin layer nitrate here? Since you are measuring nitrate in three different locations, you should always be specific about which location you are talking about and not just say "nitrate".

Page 4 line 128: Perhaps you should say that you are referring to the e-folding depth of the snow photic zone, UV radiation, or something less vague.

Page 5 line 157: If you remove the word "While" from the beginning of this sentence it will be grammatically correct.

Page 8 line 259: By "lower" do you mean "shallower"? Lower could also mean deeper, so it would be better to use words like deeper and shallower when referring to the depth in the snow beneath the surface.

Section 2.7: It seems quite important to state what value you assume for the N-isotopic composition of primary nitrate, before it is impacted by photolysis.

Page 12 line 371: "values" of what? Concentration? Isotopes? Both? Since you measured more than one thing you need to be specific.

Page 13 Section 3.5: You should probably note that gas phase HNO3 and particulate nitrate have different dry deposition rates.

Section 3.6: I think you should elaborate on the difference between the actual (from photolysis) and apparent fraction factor. The latter is lower than the actual due to recycling. Also, related to this, can the difference between the actual and apparent fractionation factor be used to calculate fractional loss?

Page 15 line 455: Is this at DML?

Page 15 line 459: Which results? Concentration? Isotopes? Both?

Page 15 lines 472 – 473: Does the model simulate the influence of hoar frost and diamond dust on nitrate deposition? The way this sentence is written it seems that it does, but this is surprising to me as I didn't think the model was a meteorological/snow model that calculated such things. It seems that you are saying that diamond dust and hoar frost explain the difference between the model and the observations, and that the model has these things but they didn't happen in reality.

Section 4.2: This is a perfect example of the section 4 having lots of introductory material without any results. The entire first paragraph reads like an introduction except for the very last sentence. However, the very last sentence of the first paragraph is not specifically supported by your observations. By the time I finished the 5 pages of section 4.2, I have no idea what you learned or what you are concluding from your new observations.

Section 4.1: Again, I have no idea what you are concluding in this section.

Page 17 line 540: "is no exception" to what?

Page 18 lines 547 – 551: It seems that if you are collecting both gas and particulate phase nitrate that you cannot conclude that sea salt promotes conversion of HNO3 to nitrate. I'm sure that it does, however, I don't see how your observations provide evidence of this.

Page 18 lines 558-559: I suggest rephrasing this as "…Dome C where the underlying snowpack is the dominant… skin layer via photolytic recycling and redeposition.

First paragraph of section 4.3: Need a reference for this statement.

Page 20 line 617: It could also be transported away as nitrate, not just NO2.

Page 21 line 648: "The contribution *to what?* of natural sources"

Page 21 lines 659-660: If you say that denitrification causes O-isotopes to increase, then you need cite a paper that provides evidence of this. However, I would delete this sentence since this paper does not address O-isotopes.

Page 22 line 690: How was the apparent fractionation factor calculate? There should be an equation for this in the methods section.

Page 22 line 706: are these FNO2 values a daily mean, a daily maximum?

Page 24 lines 742 – 746: What is the difference between the grain size and impurity concentrations at these two locations? The N-isotopes may be sensitive to something, but if that something shows no significant difference between these two locations than it won't be able to explain the differences in N-isotopes.

Page 25 lines 786-787: The e-folding depth is 20 years yet the nitrate is archived after 5-6 years. This seems conflicting. Why is nitrate archived when it is still in the photic zone?

Page 25 lines 792 and 796: Are you referring to observed or modeled values here?

Page 25 line 807: Snow optical properties are part of what determines the depth of the photic zone, so it seems weird to mention these two things separately here as if they aren't related.

Page 26 Line 821: "lower" than what?

Page 28, last sentence of section 4.5.3: Explicitly state here why the sensitivity is greater at Dome C than at DML.

Page 28 line 899: What would cause a change in the e-folding depth?

Page 29 line 919: What does "unpicking" mean in this context? Perhaps choose a better word.

Page 29 line 931: This is percent. Should it be permil?

Figure 1 caption: State the difference between HiVol 2 and 3.

Figure 2: The boxes in this figure are totally unclear to me. What are the gray boxes trying to convey?

Figure 4: Mark the seasons (e.g., shade winter) in c) and f). It's hard to see the seasonal cycles.

Figure 4: Why does the x-axis scales to all the way to 1500 and 300 when the observations are much lower than this? The axes should be scaled by the range of the data.

Figure 5: Why is "recycling factor" listed as a process occurring below the snow photic zone? How is "tropospheric input" different from "net lateral advection"?

Figure 7: Why are these values for accumulation rates chosen as sensitivity studies in the model? What is the base case that you are changing each parameter around?

Figure 8: This is showing the change in TCO around what value?

Reference:

Lee, H.-M., Henze, D. K., Alexander, B., and Murray, L. T.: Investigating the sensitivity of surface-level nitrate seasonality in Antarctica to primary sources using a global model, Atm. Env., 89, 757-767, 10.1016/j.atmosenv.2014.03.003, 2014.

---

## Referee Comment (RC2) · Anonymous Referee #2 · 13 Dec 2019

This work present new observations of the isotopic composition of nitrate in atmospheric, skin layer, surface snow and snowpit samples from Dronning Maud Land, Antarctica. The goal of the work is to understand the primary driver(s) of post-depositional processing of nitrate in this environment, based upon a snowpack box model, and determine whether it would be possible to uncover a d15N-nitrate signal that is dependent upon total column ozone. A great deal of work has been done on interpreting the d15N-nitrate in surface snow and the atmosphere at Dome C, and this work seeks to expand the application of similar tools to another site with different environmental conditions (i.e. accumulation rate, snowfall timing, and e-folding depth).

While the data and methods in this work appear sound, and the results are interesting, the manuscript reads as a first draft. With so many authors on this paper, and several who have led work at Dome C, it is surprising how disorganized and filled with typos this work is. Overall, it is difficult to connect the results with the interpretation. Much of the discussion section reads as introduction, and the introduction itself is overly repetitive.

Most critically on the interpretation front, is that the authors must first consider the e-folding depth results THEN the results considering changes in the snow accumulation, timing, etc. The model (shown in Figure 7) is simply not at all good enough to draw the conclusions the authors are drawing UNTIL it is shown that with the reduced e-folding depth the model can actually reproduce the observations. This is done as a final step and negates all of the previous discussion that suggests that the model output is robust, and therefore negates the conclusions being drawn.

In section 4.2.3, confidence is built in that TRANSITS can reproduce the seasonal cycle, not at the site here that we are comparing with, but at other sites in Antarctica? This makes no sense.

The introduction should be rewritten to better frame where the paper is headed, after the discussion section is reorganized and edited.

It is well established that photolysis is a major driver of nitrate loss across East Antarctica, between the vast literature at Dome, Erbland's work and Shi's work. At this point, this should be a starting point, not something that is derived step-wise in the manuscript. Furthermore, it would help tighten up and shorten the manuscript. Finally, take a closer look at and include Shi et al., Investigation of post-depositional processing of nitrate in East Antarctic snow: isotopic constraints on photolytic loss, re-oxidation, and source inputs, Atmos. Chem. Phys., 15, 9435–9453, 2015, https://doi.org/10.5194/acp-15-9435-2015.

The evaporation (volatilization) of nitrate needs more discussion in the introduction. It is stated that that is negligible several times and then, finally, it is explained at lines 680-685 that this is temperature dependent process and THEREFORE not important AT THIS SITE. This should be detailed much earlier on.

More evidence should be provided that the collection method is robust for representing the isotopic composition of nitrate in the air. The authors state that is "assumed that the atmospheric NO3- collected on glass fibre filters represents the sum of atmospheric particulate NO3- and HNO3 (gas phase)" and then says this is described in Frey et al. (2009). Except Frey et al. makes this exact same ASSUMPTION without providing any evidence that this is the case. Later in this manuscript it is suggested that differences found from earlier work by Weller and Wagenbach may be bcasue different filters were used (Teflon) – which filters are robust? Might the Weller and Wagenbach filters only represent one phase? and if so the authors should understand what the implications of this is for the data comparison.

In section 3.5 and in figure 5, it is never explained what scenario 1 vs scenario 2 is, and where does the 296 for wet deposition come from?

The authors make a claim regarding d18O-NO3- data late in the manuscript (line 660) – this is inappropriate given that none of that data is shown. Further, the claim is that the d18O rises due to denitrification, but there is not previous validity to this statement in the literature.

It is stated that the poorly constrained "quantum yield of NO3- photolysis in natural snow" yields a flux of NOx that is 50 times too high. Can this not be tested in TRANSITS? And shouldn't this affect the TRANSITS results as well and not just the estimated calculations here? Finally, consider better comparing with Zatko et al., 2016 throughout the discussion – the equations used here are very similar to that paper and that work does in fact quantify the recycling despite the line later that only two studies have done so and then Zatko's work is compared with.

Also Zatko's earlier work (2013) on impurities should be better considered. Later it is attributed to Geng et al. for some reason. Can impurities in the snow not help account for some of the difference in the photolysis results? i.e. difference in impurities in the snow at DML and Dome C could help to account for the significantly lower photolysis rates at DML.

The timing and rate of snowfall CANNOT explain the misalignment between the observations and model results (Line 845). The e-folding depth is critical to right FIRST, then test the sensitivity of these other parameters to determine how to make the fit better. Literally none of the model results presented in Figure 7 before the e-folding depth results come close to overlapping with the observations. Also, you should consider having the model results on a different x-axis so that the depth profile, especially for d15N, can be seen. At this point, the idea of seasonality and the possibility of interannual interpretation is difficult to see.

Technical comments:

I do my best to point out a lot of simple errors, but it would behoove the authors to take a closer read on the next version of the manuscript.

Line 10: d15N-NO3- should be (d15N-NO3-)

Line 20: photochemical processes cannot drive the archiving of nitrate; it drives the loss of nitrate or recycling of nitrate from the snow.

Line 34: TOC should be TCO

Line 38: rephrase this line – it appears as if you are suggesting that NO3- is formed from oxidation of N2

Line 57: J should be (J)

Line 64: add a comma after Greenland

Line 97: this is the first use of PSC, spell it out and explain their purpose here

Line 169-170: the additional skin layer samples for comparison should be represented in the data figures.

Line 219: a references for the seawater ratio (I assume this means sea salt ratio) should be included.

Line 300: followed should be follows

Line 303: add the word in after changes

Line 370: remove and before archived

Line 398: as a year round does not make sense

Line 457: inter-decadal should be interannual

Line 475: Fig 5 should not be cited here

Line 490: remove are

Line 530: up taken should be taken up

Line 536: the idea that "NO3- peaks are substantially modified after burial" undermines so much of the current manuscript that suggests that NO3- is archived once buried. Rephrase.

Line 566: rephrase "form of nitrogen to skin layer"

Line 612: back to THE skin later with a day, or IS transported away

Line 624-625: see comment above about better explaining the evaportation results; and Shi et al 2019 reflect field conditions, not laboratory conditions.

Line 643-653: I have serious issues here with the interpretation of previous literature. First and foremost, the assumption that anthropogenic emissions of NOx are positive in d15N and natural emissions are negative in d15N is false and not up to date. Recent work shows that vehicle emissions are, in fact, negative in d15N (Miller et al.,

2017, Vehicle NOx emission plume isotopic signatures: Spatial variability across the eastern United States, J. Geophys. Res. Atmos., 122, doi:10.1002/2016JD025877)/ At least three works show that fertilized soil emissions (which are not considered a natural source in emission inventories) are very negatives in d15N (Yu & Elliott, 2017, Novel method for nitrogen isotopic analysis of soil-emitted nitric oxide. Environmental Science & Technology, 51(11), 6268–6278, https://doi.org/10.1021/acs.est.7b00592; Miller et al., 2018, Isotopic composition of in situ soil NOx emissions in manure fertilized cropland, Geophysical Research Letters, 45(21), 12058-12066, https://doi.org/10.1029/2018GL079619.; Li & Wang, 2008, Nitrogen isotopic signature of soil-released nitric oxide (NO) after fertilizer application. Atmospheric Environment, 42(19), 4747–4754. https://doi.org/10.1016/j.atmosenv.2008.01.042.). Geng et al. make the exact same false interpretation that anthropogenic sources are positive and therefore are ruled out in interpretation of a Summit, Greenland core – do not cite this is evidence when it is simply an unwarranted assumption. Finally, please be more precise in the language here – emission sources emit NO (except diesel engines, which can also emit NO2) or you can call it NOx; they do not "have positive d15N-NO3-" values since they do not emit nitrate, it is secondarily formed and subject to partitioning in the atmosphere, which Geng et al. invoke as a major mechanism to explain Greenland ice results, and this is wholly ignored in the current work.

Line 690-695: a range of -19 per mil to +12 per mil does not at all "nicely match" with the expected -59 per mil to -16 per mil.

Line 698: add an before annual

Section 4.3.3 – seems odd to switch to section i, ii, iii, etc here when earlier subsections are numbered in series (i.e. 4.3.3.1, 4.3.3.2, etc).

Line 780: Concluding that photolysis is an important driver is not an interesting result given the vast evidence for this throughout the EAIS. The other conclusions are still interesting but this should not be the primary focus. Furthermore, the fact that there is

less photolysis than expected is really very interesting.

Line 850-851: Interestingly, Geng et al. 2015 ignores surface snow work at Summit, Greenland to invoke that post-depositional processing can explain everything (Fibiger et al. (2016), Analysis of nitrate in the snow and atmosphere at Summit, Greenland: Chemistry and transport, J. Geophys. Res. Atmos., 121, 5010–5030, doi:10.1002/2015JD024187; Fibiger et al. (2013), The preservation of atmospheric nitrate in snow at Summit, Greenland, Geophys. Res. Lett., 40, 3484–3489, doi:10.1002/grl.50659.). Here, you are showing that is critical to use surface work to best determine how to interpret ice cores. I suggest you reconsider including comparison with Geng's work here.

Line 855: depending ON the timing

Line 902: due TO photolysis-driven

Line 905: This should reference Zatko et al. not Geng et al.

Line 913: I strongly disagree with the statement that "TRANSITS doe such a good job of simulating NO-3 recycling in Antarctica" unless you do the e-folding fit first and then explore sensitivities. ( I do agree that it is an excellent recommendation to use TRANSITS to assess sites that would be useful for interpreting nitrate isotopic records!)

Line 919: unpicking is a strange word here – distinguishing?

Line 922: resulting IN an enrichment

Line 929: this conclusion would make more sense if the e-folding depth model experiments were done first.

Line 945: THE ground

Figure 3: how is the data averaged here?

Figure 4: x-axis on right side is misspelled

Figure 5: what is scenario 1 vs 2, this is never explained

Table 2: Archived should be >30 cm not <30 cm, correct?

[Figure]

---

## Author Response (AR1)

**Authors' Response to ACP MS No.: acp-2019-669**

We thank the reviewer for their comments and suggestions. In the text below, we outline our responses in blue. Line numbers refer to the revised manuscript.

**Anonymous Referee #1**

Winton et al report observations of the nitrogen isotopic composition of nitrate in Antarctic (DML) in the atmosphere, snow "skin layer", and depth profiles in snow pits. They use these observations combined with a snow chemistry model to understand what controls the variability in the nitrogen isotopes and nitrate concentrations and to assess the utility of such observations in ice cores as a proxy for past surface UV radiation. They conclude that although photolytic driven recycling and loss of reactive nitrogen is what determines the nitrate isotopes and abundances at this location, that variability in parameters such as snow accumulation rate have a large enough impact on the nitrogen isotopes so as to preclude the influence of variability in surface UV in determining the nitrogen isotopic composition of snow nitrate.

We agree that variability in the snow accumulation rate precludes the use of using the  $\delta^{15}$ N-NO3- composition as a surface UV proxy in the short-term. However, longer-term UV trends may be inferred from ice cores at the site taking into account or constraining factors masking the UV-impact such as e-folding depth and accumulation rate.

The observations and model-based interpretation are robust and important as it would be useful to have a paleo-UV proxy. As the authors state, it is important to assess the sensitivity of nitrogen isotopes at particular locations in order to determine its usefulness as a proxy for any given process. Unfortunately, the paper is frustrating to read because it is it so repetitive and spends so much time on introductory material throughout the manuscript that it is hard to find the actual interpretation of the results. It is as if the authors are afraid to state their interpretation. I suggest moving (and condensing) the introductory material that is spread throughout the manuscript (especially in section 4) to the introduction and making sure that the actual interpretation is presented up front instead of hidden. Because of this, it's sometimes hard to determine if the results support their conclusions. The paper as is reads as a first or second draft, not a final manuscript ready for submission for publication. This is particularly evident in the conclusions, which is not even written in paragraph form.

We agree with the reviewer that too much background information is spread throughout the discussion rather than the introduction. We have condensed and moved the most relevant material to the introduction. Overall, we have revised the manuscript to provide better clarity on our interpretation of the results and conclusions.

**Additional comments:**

I find the concept of the number of times nitrate is recycled difficult to wrap my head around. Does this mean that each molecule of nitrate is recycled on average three times before being archived? Is this averaged over the depth of the photic zone? I would imagine some molecules (like those that are close to the surface in summer) are recycled more than others, so that there is actually some distribution of recycling events on a molecule per molecule basis so as to average around the number 3. Is this interpretation correct? I suggest elaborating on this in the methods section.

The number of recylings is the "average number of recyclings" undergone by the archived nitrate, i.e., below the zone of active photochemistry. Recycling includes the following processes: the combination of NOx production from nitrate photolysis in snow, venting to the air above the snowpack and subsequent atmospheric re-oxidation of NOx to form atmospheric nitrate, the deposition (dry and/or wet) of a fraction of the product, and the export of another fraction (Erbland et al., 2015). In TRANSITS, there is a tracer/counter called CYCL, which, in a given box (snow layer or atmosphere), represents the average number of recyclings undergone by nitrate in that box. The CYCL value for primary nitrate is set to 0, and CYCL variables in the boxes are incremented by 1 each time NO2 molecules cross the air-snow interface. The average number of recyclings is calculated as a mass weighted average of the CYCL values of the 52 snow layers (representing one week of snowfall) which are archived below 1 m over the course of 1 year, in order to average out any seasonal variability.

The average number of recyclings undergone by the archived nitrate at our study site in Dronning Maud Land (DML) is 2 for the last layer before leaving the photic zone, which means that, on average, the archived nitrate at DML has undergone 2 recyclings (i.e., loss, local re-oxidation, deposition). Erbland et al. (2015) notes that this number of recyclings represents an average value for the archived nitrate. Considering individual ions in the archived nitrate, the number of recyclings could be variable as some ions may have travelled through the entire snowpack zone of active photochemistry without being recycled, while some underwent many recyclings.

We have expanded the explanation of the number of recycling's in the methodology (lines 337-345) as follows:

Modified text: "TRANSITS calculates the average number of recyclings undergone by the archived  $NO_3^-$ , i.e., below the zone of active photochemistry. In TRANSITS, the average number of recyclings undergone by  $NO_3^-$  in a given box (snow layer or atmosphere) is represented by a tracer (or counter) called CYCL. The CYCL value for primary  $NO_3^-$  is set to 0, and CYCL variables in the boxes are incremented by 1 each time  $NO_2$  molecules cross the air-snow interface. The average number of recyclings is calculated as a mass weighted average of the CYCL values of the 52 snow layers (representing one week of snowfall) which are archived below 1 m over the course of 1 year, in order to average out any seasonal variability. Erbland et al. (2015) notes that the number of recyclings represents an average value for the archived  $NO_3^-$ , i.e., considering individual ions in the archived  $NO_3^-$ , the number of recyclings could be variable as some ions may have travelled through the entire snowpack zone of active photochemistry without being recycled, while some underwent many recyclings."

Second paragraph of the introduction: It seems strange to say that the primary sources of nitrate are transport of nitrate from the stratosphere and transport of alkyl nitrates in the troposphere without mentioning transport of inorganic nitrate in the troposphere. Where is the evidence that inorganic nitrate is not transported in the troposphere to Antarctica? A model study suggests that it is certainly possible (Lee et al., 2014). Also, soil NOx should be mentioned as a NOx source in the troposphere.

We have added the transport of inorganic nitrate (fossil fuel combustion, soil, and lightning) and referenced Lee et al. (2014) in lines 53-56 as follows:

Modified text: "Primary sources of reactive nitrogen species to the Antarctic lower atmosphere and snow pack include the sedimentation of polar stratospheric clouds (PSC) in late winter (Savarino et al., 2007) and, to a minor extent, advection of oceanic methyl nitrate (CH3NO3) and peroxyacyl nitrates (PAN) (Jacobi et al., 2000;Jones et al., 1999;Beyersdorf et al., 2010), in addition to tropospheric transport of inorganic NO3- from lightning, biomass burning and soil emissions (Lee et al., 2014)."

Page 2 lines 61 – 62: Reprase to: "Model results from Zatko et al. (2016) suggest that..."

**Done**

Page 4 line 116: Are you referring to skin layer nitrate here? Since you are measuring nitrate in three different locations, you should always be specific about which location you are talking about and not just say "nitrate".

No, we are referring to ice core nitrate as mentioned in the sentence. We have added "ice core nitrate" for clarity.

Page 4 line 128: Perhaps you should say that you are referring to the e-folding depth of the snow photic zone, UV radiation, or something less vague.

**Done**

Page 5 line 157: If you remove the word "While" from the beginning of this sentence it will be grammatically correct.

**Done**

Page 8 line 259: By "lower" do you mean "shallower"? Lower could also mean deeper, so it would be better to use words like deeper and shallower when referring to the depth in the snow beneath the surface.

"Lower" has been replaced with "shallower".

Section 2.7: It seems quite important to state what value you assume for the N-isotopic composition of primary nitrate, before it is impacted by photolysis.

The  $\delta^{15}$ N-NO3- value of primary nitrate is set to 19 ‰ as estimated by Savarino et al. (2007). This value is reported in Table S3.

Page 12 line 371: "values" of what? Concentration? Isotopes? Both? Since you measured more than one thing you need to be specific.

We have added  $\delta^{15}$ N-NO3- values to this sentence.

Page 13 Section 3.5: You should probably note that gas phase HNO3 and particulate nitrate have different dry deposition rates.

Agreed, we have noted this in lines 418-419 as follows:

Modified text: "Although gas phase  $HNO_3$  and particulate  $NO_3^-$  have different dry deposition rates..."

Section 3.6: I think you should elaborate on the difference between the actual (from photolysis) and apparent fraction factor. The latter is lower than the actual due to recycling. Also, related

to this, can the difference between the actual and apparent fractionation factor be used to calculate fractional loss?

We have discussed the difference in lines 88-91. We found that the single loss Raleigh model does not work well at sites with an annual signal in  $\delta^{15}$ N-NO3-, and therefore suggest that at DML it is not useful for calculating fractional loss. The text was modified as follows:

Modified text: "As this approach may oversimplify the processes occurring at the air-snow interface, Erbland et al. (2013) referred to the quantity as an "apparent" fractionation constant ( $^{15}\varepsilon_{app}$ ). Thus, the apparent fractionation constant represents the integrated isotopic effect of the processes involving NO3- in the surface of the snow pack and in the lower atmosphere."

Page 15 line 455: Is this at DML?

Yes, the Weller et al. (2004) study is at DML. We have added "at DML" for clarity.

Page 15 line 459: Which results? Concentration? Isotopes? Both?

Both. We have added "NO3- mass concentrations and  $\delta^{15}$ N-NO3-" to the sentence.

Page 15 lines 472 - 473: Does the model simulate the influence of hoar frost and diamond dust on nitrate deposition? The way this sentence is written it seems that it does, but this is surprising to me as I didn't think the model was a meteorological/snow model that calculated such things. It seems that you are saying that diamond dust and hoar frost explain the difference between the model and the observations, and that the model has these things but they didn't happen in reality.

No, the model does not simulate the influence of hoar frost and diamond dust on nitrate deposition. We suggest that the difference between the simulated and observed nitrate concentrations in the skin layer is due to a sampling artefact where we are diluting the high nitrate concentrations of diamond dust and hoar frost observed in new deposition. We have modified the sentence in lines 564-566 as follows:

Modified text: "Here, extremely high concentrations of  $NO_3^-$  from new deposition from diamond dust and hoar frost are also found. In summary, it is likely that we do not measure such high  $NO_3^-$  mass concentrations in hoar frost and diamond dust in the skin layer because of sampling artefacts or blowing snow, which can dilute or remove the diamond dust and hoar frost."

Section 4.2: This is a perfect example of the section 4 having lots of introductory material without any results. The entire first paragraph reads like an introduction except for the very last sentence. However, the very last sentence of the first paragraph is not specifically supported by your observations. By the time I finished the 5 pages of section 4.2, I have no idea what you learned or what you are concluding from your new observations.

Section 4.2 has been condensed to 2 pages and the interpretation of results clarified.

Section 4.1: Again, I have no idea what you are concluding in this section.

We conclude that overall our nitrate conclusion measurements agree well with the literature and that the simulated results from TRANISTS match our observations with the exception of the skin layer concentrations. We provide possible reasons for this difference, and have modified Section 4.1 for clarity.

Page 17 line 540: "is no exception" to what?

We have deleted "is no exception".

Page 18 lines 547 - 551: It seems that if you are collecting both gas and particulate phase nitrate that you cannot conclude that sea salt promotes conversion of HNO3 to nitrate. I'm sure that it does, however, I don't see how your observations provide evidence of this.

We agree and have removed this section.

Page 18 lines 558-559: I suggest rephrasing this as "...Dome C where the underlying snowpack is the dominant... skin layer via photolytic recycling and redeposition.

**Done**

First paragraph of section 4.3: Need a reference for this statement.

Section of text has been removed.

Page 20 line 617: It could also be transported away as nitrate, not just NO2.

We have added nitrate to the sentence.

Page 21 line 648: "The contribution to what? of natural sources"

The "contribution of nitrate from natural sources" has been added to the sentence.

Page 21 lines 659-660: If you say that denitrification causes O-isotopes to increase, then you need cite a paper that provides evidence of this. However, I would delete this sentence since this paper does not address O-isotopes.

Agreed, we have delete the sentence.

Page 22 line 690: How was the apparent fractionation factor calculate? There should be an equation for this in the methods section.

Please see equation 7 in the methods in section 3.6.

Page 22 line 706: are these FNO2 values a daily mean, a daily maximum?

The model estimates are mean values for the 1-14 January 2017 period as stated in line 465.

Page 24 lines 742 - 746: What is the difference between the grain size and impurity concentrations at these two locations? The N-isotopes may be sensitive to something, but if that something shows no significant difference between these two locations than it won't be able to explain the differences in Nisotopes.

As far as we are aware, at Dome C and DML, there are only published values of major ions in the snow pack in the top 30 cm which is the depth relevant for influencing the e-folding depth. For example, in the top 30 cm of the snow pack nitrate concentrations at Dome C are  $\sim$ 75 ppb (Frey et al., 2009) while they are  $\sim$ 55 ppb at DML (this study). Insoluble dust concentrations are higher at DML than Dome C in the Holocene (Delmonte et al., 2019), while no black carbon data is available at DML for comparison. There is considerate variability in the snow grain size

across Antarctica (Brucker et al., 2010). In particular, wind crust layers which occur in the snowpack have larger grain sizes and these have been observed at Dome C (France et al., 2011).

Based on the available data, we don't have a clear understanding of why the e-folding depth is lower at DML. Station pollution is less than at Dome C (Helmig et al., 2020), thus it is unlikely related to black carbon. However, other impurities are deposited in DML snow due to the closer proximity to marine sources. Snow grain sizes may be smaller, which will increase scattering, at DML than Dome C (Brucker et al., 2010). The larger e-folding depth at Dome C is in part due the presence of windcrust layers which comprise larger grain sizes. Sensitivity studies show that nitrate impurities have only a small contribution on the e-folding depth compared to scattering by snow grains which dominate (Chan et al., 2015;France et al., 2011;Zatko et al., 2013). Further work is required to determine why the e-folding depth is lower at DML. We have discussed this in lines 692-700 as follows:

Modified text: "The e-folding depth depends on the density and grain size of snow crystals, and the concentration of impurities. In terms of published values, impurity concentrations are generally higher at DML, for example dust and major ion concentrations (Delmonte et al., 2019;Legrand and Delmas, 1988), due to proximity to marine sources. Yet station pollution is greater at Dome C (Helmig et al., 2020), and thus the lower e-folding depth is unrelated to black carbon concentrations. Furthermore, there is considerate variability in snow grain size across Antarctica. The larger e-folding depth in windcrust layers at Dome C is due to larger grain sizes in those layers (France et al., 2011). Snow grain size may be smaller at DML, which will increase scattering (Brucker et al., 2010), but further work is required to confirm if this is the dominate factor influencing the lower e-folding depth at DML. Sensitivity studies show that  $NO_3^-$  impurities make a small contribution to the e-folding depth compared to scattering by snow grains which dominate (France et al., 2011;Chan et al., 2015;Zatko et al., 2013)."

Page 25 lines 786-787: The e-folding depth is 20 years yet the nitrate is archived after 5-6 years. This seems conflicting. Why is nitrate archived when it is still in the photic zone?

Weller et al. (2004) determine archived nitrate as the typical mean concentrations representative for the last 100 years. We have deleted the e-folding time of 20 years from sentence.

Page 25 lines 792 and 796: Are you referring to observed or modeled values here?

Observed values. We have added "observed" sentence.

Page 25 line 807: Snow optical properties are part of what determines the depth of the photic zone, so it seems weird to mention these two things separately here as if they aren't related.

We have deleted snow optical properties.

Page 26 Line 821: "lower" than what?

Lower than the base case profile. We have added "base case" to the sentence.

Page 28, last sentence of section 4.5.3: Explicitly state here why the sensitivity is greater at Dome C than at DML.

Done

Page 28 line 899: What would cause a change in the e-folding depth?

The e-folding depth could change due to a change in dust or black carbon concentrations (which are light absorbing impurities) or a change in the snow morphology in a particular snow layer. This has been added to lines 843-844 as follows:

Modified text: "The e-folding depth could change over time due to higher or lower dust or black carbon concentrations or a change in the snow grain size in a particular snow layer."

Page 29 line 919: What does "unpicking" mean in this context? Perhaps choose a better word.

Unpicking been replaced with disentangling.

Page 29 line 931: This is percent. Should it be permil?

Yes, percent symbols have been changed to permil symbols.

Figure 1 caption: State the difference between HiVol 2 and 3.

Done.

Figure 2: The boxes in this figure are totally unclear to me. What are the gray boxes trying to convey?

The box a) is in insert of the Kohnen Station and boxes b) and c) our inserts of our two sampling sites at the station. We have edited the caption to make this clearer.

Figure 4: Mark the seasons (e.g., shade winter) in c) and f). It's hard to see the seasonal cycles.

We have shaded the seasons in panel c) and updated the caption. The snow pit in Frey et al. (2009) was not dated and thus we cannot shade the seasons for Dome C.

Figure 4: Why does the x-axis scales to all the way to 1500 and 300 when the observations are much lower than this? The axes should be scaled by the range of the data.

We deliberately chose to keep the x-axis the same for each plot rather than scaling by the range of the data. This is so the reader can see the difference in concentration and isotopic values between the atmosphere, skin layer and snowpack profiles and between the two sites. In particular, we wanted to highlight the denitrification of the snowpack from enriched values in the snow pits to depleted values in the atmosphere.

Figure 5: Why is "recycling factor" listed as a process occurring below the snow photic zone? How is "tropospheric input" different from "net lateral advection"?

We have moved recycling to encompass the atmosphere and snow boxes. We have renamed net lateral advection to tropospheric input.

Figure 7: Why are these values for accumulation rates chosen as sensitivity studies in the model? What is the base case that you are changing each parameter around?

The justification for the range of accumulation rates used in the sensitivity study can be found in section 4.5.1.

The base case is explained in section 4.5 and in the caption, we refer the reader to that section.

Figure 8: This is showing the change in TCO around what value?

We used present day TCO values that were used in all our calculations. These values vary weekly and can be found the supplement (Table S3). For each week, a constant amount of ozone (e.g. 100 DU) was added or subtracted from these values. We have added this information to the caption and text in lines 816-817 as follows:

Modified text: "For each week, a constant amount of ozone (e.g. 100 DU) was added or subtracted from these present day values."

We thank the reviewer for the comments and suggestions. In the text below, we outline our responses in blue. Line numbers refer to the revised manuscript.

**Anonymous Referee #2**

This work present new observations of the isotopic composition of nitrate in atmospheric, skin layer, surface snow and snowpit samples from Dronning Maud Land, Antarctica. The goal of the work is to understand the primary driver(s) of post-depositional processing of nitrate in this environment, based upon a snowpack box model, and determine whether it would be possible to uncover a d15N-nitrate signal that is dependent upon total column ozone. A great deal of work has been done on interpreting the d15N-nitrate in surface snow and the atmosphere at Dome C, and this work seeks to expand the application of similar tools to another site with different environmental conditions (i.e. accumulation rate, snowfall timing, and e-folding depth).

While the data and methods in this work appear sound, and the results are interesting, the manuscript reads as a first draft. With so many authors on this paper, and several who have led work at Dome C, it is surprising how disorganized and filled with typos this work is. Overall, it is difficult to connect the results with the interpretation. Much of the discussion section reads as introduction, and the introduction itself is overly repetitive.

We agree with reviewer, and have reorganised and edited the entire manuscript with particular focus on the introduction and discussion. Please also see our response to referee #1's main comment.

Most critically on the interpretation front, is that the authors must first consider the efolding depth results THEN the results considering changes in the snow accumulation, timing, etc. The model (shown in Figure 7) is simply not at all good enough to draw the conclusions the authors are drawing UNTIL it is shown that with the reduced e-folding depth the model can actually reproduce the observations. This is done as a final step and negates all of the previous discussion that suggests that the model output is robust, and therefore negates the conclusions being drawn.

We agree with the reviewer that the e-folding depth results need to be accounted for before suggesting that the model output fits the observations. We reran TRANISTS with a 5 cm e-folding depth scenario as suggested by reviewer #2. First, we modified the methods section to include a 5 cm e-folding depth scenario (section 2.7 Air-snow transfer modelling). Second, we have added a section on the e-folding depth in the results section (section 3.8 Light attenuation through the snow pack) so the reader can see the observed results of a lower measured e-folding depth at DML upfront. We have also compared the new 5 cm e-folding depth case to the base case in section 3.9 Simulated nitrate mass concentrations and isotopic ratios from TRANSITS modelling. Third, we compare the lower e-folding depth to Dome C and discuss the impact of the lower e-folding depth on post-depositional processes (section 4.3.2.2 Nitrate recycling). Fourth, we have rerun the snow accumulation rate and snowfall timing sensitivity tests with an e-folding depth of 5 cm and modified Figs. 6, 7 and 8 and Table 2 with the new sensitivity test results. An observed e-folding depth of 5 cm was used as it has good fit with observations down to 30 cm depth. In light of the new sensitivity results from TRANSITS, we have reorganised section "4.5 Sensitivity of  $\delta^{15}$ N-NO3- to deposition parameters and implications for interpreting

ice core records of  $\delta^{15}$ N-NO3- at DML". Here, we discuss the TRANSITS modelling results by first showing that the base case scenario cannot reproduce the observations and that a reduced e-folding depth is required. Next, we discuss the sensitivity results of a variable snow accumulation rate and snowfall timing with an e-folding depth of 5 cm. With an e-folding depth of 5 cm we are able to reproduce the observations and thus our original conclusions that TRANSITS does a good job are valid.

In section 4.2.3, confidence is built in that TRANSITS can reproduce the seasonal cycle, not at the site here that we are comparing with, but at other sites in Antarctica? This makes no sense.

There are no year-round measurements of atmospheric or skin layer  $\delta^{15}$ N-NO3- at DML to compare to the TRANSITS seasonality simulations. This section has been rewritten to show that of the available year-round observations and seasonality simulations of atmospheric and skin layer  $\delta^{15}$ N-NO3- and nitrate mass concentrations in Antarctica, the seasonal pattern is the same at all Antarctic sites. The section has been renamed 4.2.2 Temporal variability of nitrate deposition.

The introduction should be rewritten to better frame where the paper is headed, after the discussion section is reorganized and edited.

**Done.**

It is well established that photolysis is a major driver of nitrate loss across East Antarctica, between the vast literature at Dome, Erbland's work and Shi's work. At this point, this should be a starting point, not something that is derived step-wise in the manuscript. Furthermore, it would help tighten up and shorten the manuscript. Finally, take a closer look at and include Shi et al., Investigation of post-depositional processing of nitrate in East Antarctic snow: isotopic constraints on photolytic loss, re-oxidation, and source inputs, Atmos. Chem. Phys., 15, 9435–9453, 2015, https://doi.org/10.5194/acp-15-9435-2015.

We agree, and this is reflected in the revised introduction and discussion. In addition, we have cited the Shi et al. (2015) reference in the appropriate places throughout the manuscript, and added the key findings of the paper in lines 137-140 as follows:

Modified text: "Erbland et al. (2013) suggest that  $NO_3^-$  loss at the coast reflects both photolysis and evaporation processes, while Shi et al. (2015) proposes that  $NO_3^-$  loss at the coast cannot be fully explained by local post-deposition processes and that seasonal cycles in the snowpack reflect stratospheric and troposphere  $NO_3^-$  sources during the cold and warm seasons respectively."

The evaporation (volatilization) of nitrate needs more discussion in the introduction. It is stated that that is negligible several times and then, finally, it is explained at lines 680-685 that this is temperature dependent process and THEREFORE not important AT THIS SITE. This should be detailed much earlier on.

As part of the revised introduction, the evaporation of nitrate is discussed in lines 86-98 as follows:

Modified text: "Fractionation constants, which assume a Rayleigh single loss and irreversible process of NO3- removal from the snow between phases during evaporation-condensation

processes, have been calculated to separate the isotopic signature of evaporation and photolysis processes. As this approach may oversimplify the processes occurring at the air-snow interface, Erbland et al. (2013) referred to the quantity as an "apparent" fractionation constant ( $^{15}\varepsilon_{app}$ ). Thus, the apparent fractionation constant represents the integrated isotopic effect of the processes involving NO3- in the surface of the snow pack and in the lower atmosphere. Nitrate evaporation from the snow pack has a  $^{15}\varepsilon_{app}$  of ~0 as determined by two independent studies (Erbland et al., 2013;Shi et al., 2019). This indicates that during NO3- evaporation, the air above the snow is not replenished and thus there is only a small NO3- mass loss. In comparison, fractionation constants associated with laboratory studies and field observations of NO3- photolysis are large:  $^{15}\varepsilon_{app} = -34$  ‰ (Berhanu et al., 2014;Meusinger et al., 2014) and -54 <  $^{15}\varepsilon_{app} < -60$  ‰ (Frey et al., 2009;Erbland et al., 2013), respectively. The negative fractionation constant obtained from photolysis implies that the remaining NO3- in the skin layer snow is enriched in  $\delta^{15}$ N-NO3-. In turn, the atmosphere is left with the source of NOx that is highly depleted in  $\delta^{15}$ N-NO3-. It follows that evaporation of NO3- is negligible on high-elevation Antarctic sites (Erbland et al., 2013;Shi et al., 2019)."

More evidence should be provided that the collection method is robust for representing the isotopic composition of nitrate in the air. The authors state that is "assumed that the atmospheric NO3- collected on glass fibre filters represents the sum of atmospheric particulate NO3- and HNO3 (gas phase)" and then says this is described in Frey et al. (2009). Except Frey et al. makes this exact same ASSUMPTION without providing any evidence that this is the case. Later in this manuscript it is suggested that differences found from earlier work by Weller and Wagenbach may be because different filters were used (Teflon) – which filters are robust? Might the Weller and Wagenbach filters only represent one phase? and if so the authors should understand what the implications of this is for the data comparison.

The glass fiber filters used in this study were employed and tested previously at Dome C, i.e., Frey et al. (2009) do not state an assumption but report evidence from tests with second stage filters. Accordingly, the atmospheric nitrate collected on glass fiber filters represents the sum of atmospheric particulate nitrate ( $p-NO_3^-$ ) and gaseous nitric acid ( $HNO_3$ ). The bulk of  $HNO_3$  present in the gas phase adsorbed most likely to aerosols on the filter. This is supported by the observation that second-stage filters (Whatman 41), known to trap  $HNO_3$  quantitatively (Morin et al., 2007), showed either very low nitrate concentrations or none at all.

In section 3.5 and in figure 5, it is never explained what scenario 1 vs scenario 2 is, and where does the 296 for wet deposition come from?

The value for wet deposition (296 pg m-2 s-1) in scenario two is calculated using equations 3-5 (total deposition – dry deposition). Scenario one and scenario two are now described in the caption of Fig. 5 and lines 429-443 as follows:

Modified text: "Taking this simple mass balance approach, a schematic of  $NO_3^-$  mass fluxes for two scenarios are illustrated in Fig. 5. Scenario 1 is an average annual budget for DML (Fig. 5a). As the atmospheric campaign did not cover an entire annual cycle, we use estimates of atmospheric NO3- fluxes at DML reported by Pasteris et al. (2014) and Weller and Wagenbach (2007) of 43 and 45 pg m-2 s-1, respectively, as year round dry deposition fluxes. Due to the linear relationship of ice core  $NO_3^-$  mass concentrations with the inverse accumulation, the authors assume that the magnitude of the dry deposition flux is homogenous over the DML region. Mean annual mass concentrations of  $NO_3^-$  in our snow pits suggest a total NO3- deposition mass flux of 110 pg m-2 s-1 and therefore a wet deposition mass flux of 65 pg m-2 s-1.

However, at relatively low snow accumulation sites where photolysis drives the fractionation of  $NO_3^-$  from the surface snow to atmosphere (Frey et al., 2009), it is necessary to take into account the skin layer in the  $NO_3^-$  flux budget as this air-snow interface is where air-snow transfer of  $NO_3^-$  takes place. In scenario 2, we utilise the available  $NO_3^-$  mass concentrations measured in aerosol, skin layer, and snow pits from the ISOL-ICE campaign to estimate the mass flux budget for January 2017 (Fig. 5b). The dry deposition mass flux of atmospheric  $NO_3^-$  during January 2017 at DML averages  $64 \pm 38$  pg m-2 s-1 (Table S5). The  $NO_3^-$  mass flux to the skin layer is 360 pg m-2 s-1, however only 110 pg m-2 s-1 of  $NO_3^-$  is archived. Considering the active skin layer, only 30 % of deposited  $NO_3^-$  is archived in the snow pack while 250 pg m-2 s-1 is re-emitted to the overlaying atmosphere."

The authors make a claim regarding d18O-NO3- data late in the manuscript (line 660) – this is inappropriate given that none of that data is shown. Further, the claim is that the d18O rises due to denitrification, but there is not previous validity to this statement in the literature.

**We have deleted this sentence as this manuscript does not address O-isotopes.**

It is stated that the poorly constrained "quantum yield of NO3- photolysis in natural snow" yields a flux of NOx that is 50 times too high. Can this not be tested in TRANSITS? And shouldn't this affect the TRANSITS results as well and not just the estimated calculations here?

We agree that it would be useful to further test the sensitivity of  $NO_x$  fluxes to quantum yield in TRANSITS given the large uncertainty of this quantity. However, this has been done previously and we therefore refer to the literature and clarify the statement in lines 471-487 as follows:

Modified text: "It should be borne in mind that the above simple model estimates (Eq. (8)) may significantly underestimate the real emission flux. Previous comparisons of  $F_{NO2}$  computed with Eq. (8) and  $F_{NOx}$  measured at Dome C showed that observations can exceed model predictions by up to a factor 50 (Frey et al., 2015;Frey et al., 2013). While NO3- mass concentrations in snow, the surface actinic flux and e-folding depth were measured at the DML field site, quantum yield of NO3- photolysis in surface snow ( $\Phi$ NO3-) was not, but introduces significant uncertainty in the model estimates. Previous lab measurements on natural snow samples collected at Dome C showed  $\Phi$ NO3- to vary between 0.003 and 0.05 (Meusinger et al., 2014). As described above (section 2.6) JNO3- used in Eq. (8) was calculated with  $\Phi$ NO3- at -30 °C (= 2 x 10-3) after Chu and Anastasio (2003), which is near the lower end of the observed range. Thus, up to half of the mismatch between Eq. (8) and Dome C observations can be explained by adjusting  $\Phi$ NO3-. Another factor contributing to larger fluxes and not included in Eq. (8) is forced ventilation.

In the more sophisticated TRANSITS model, Erbland et al. (2015) found that the photolytic quantum yield was one of the major controls on archived flux and primary input flux at Dome C. Erbland et al. (2015) initially used a quantum yield of 2.1 x  $10^{-3}$  at 246 K (France et al., 2011) but it underestimated NO3- recycling and overestimated primary NO3- trapped in snow. Adjusting the quantum yield to 0.026, within the range observed in the lab (Meusinger et al., 2014), gave more realistic archived  $\delta^{15}$ N-NO3- values. However, at Dome C TRANSITS simulated FNO2 fluxes were about a factor of 9 - 18 higher than observed FNOx. Erbland et al.

(2015) suggested that the discrepancy could result from the simplifications made in the TRANSITS model regarding the fate of  $NO_3^-$  photolysis products."

Finally, consider better comparing with Zatko et al., 2016 throughout the discussion – the equations used here are very similar to that paper and that work does in fact quantify the recycling despite the line later that only two studies have done so and then Zatko's work is compared with.

What we meant to say is that there are only two methods in the literature to quantify the number of recyclings (Erbland et al., 2015;Davis et al., 2008). Zatko et al. (2016) uses the Davis approach. We have edited the section on recycling, stating there are in fact three studies and have included the Zatko et al. (2016) paper in our comparison in lines 671-672 as follows:

Modified text: "Only three studies have attempted to quantify the degree of NO3- recycling between the air and snow (Davis et al., 2008;Erbland et al., 2015;Zatko et al., 2016)."

Also Zatko's earlier work (2013) on impurities should be better considered. Later it is attributed to Geng et al. for some reason. Can impurities in the snow not help account for some of the difference in the photolysis results? i.e. difference in impurities in the snow at DML and Dome C could help to account for the significantly lower photolysis rates at DML.

The impact of impurities on e-folding depth is addressed in section 4.3.2.2. Please see response to referee #1 concerning the impact of grain size and impurities on e-folding depth.

The timing and rate of snowfall CANNOT explain the misalignment between the observations and model results (Line 845). The e-folding depth is critical to right FIRST, then test the sensitivity of these other parameters to determine how to make the fit better. Literally none of the model results presented in Figure 7 before the e-folding depth results come close to overlapping with the observations. Also, you should consider having the model results on a different x-axis so that the depth profile, especially for d15N, can be seen. At this point, the idea of seasonality and the possibility of interannual interpretation is difficult to see.

We agree and thank the reviewer for the valuable comment to improve the manuscript. Please see our response to the e-folding depth comment above. We carried out the TRANISTS runs as suggested by referee #1. An e-folding depth of 5 cm has a much better fit with the observations. With the new TRANSITS runs, our conclusions reinforce the importance of accounting for the e-folding depth measurements across Antarctica. Regarding the x-axis on Fig. 7, the new TRANSITS runs with an e-folding depth of 5 cm move the simulated  $\delta^{15}$ N-NO3- values to more negative values. The better fit with the observations means it is much easier to see the interannual variability in the  $\delta^{15}$ N-NO3-. In addition, the seasonal variability is clearly visible in Fig. 4.

Technical comments:

I do my best to point out a lot of simple errors, but it would behoove the authors to take a closer read on the next version of the manuscript.

Line 10: d15N-NO3- should be (d15N-NO3-)

Done.

Line 20: photochemical processes cannot drive the archiving of nitrate; it drives the loss of nitrate or recycling of nitrate from the snow.

"Photochemical processes" has been replaced with "nitrate recycling".

Line 34: TOC should be TCO

Done.

Line 38: rephrase this line – it appears as if you are suggesting that NO3- is formed from oxidation of N2  $\,$

Done.

Line 57: J should be (J)

Done.

Line 64: add a comma after Greenland

Done.

Line 97: this is the first use of PSC, spell it out and explain their purpose here

Done.

Line 169-170: the additional skin layer samples for comparison should be represented in the data figures.

The samples representing spatial variability are already plotted in Fig. S6 and Fig 3. To increase visibility of these samples, we have added an error bar representing the spatial variability to Fig. 3. This is considerably lower than the instrumental variability (error bars are smaller than sample points).

Line 219: a references for the seawater ratio (I assume this means sea salt ratio) should be included.

Keene et al. (1986) reference added for the sea salt ratio.

Line 300: followed should be follows

Done.

Line 303: add the word in after changes

Done.

Line 370: remove and before archived

Done.

Line 398: as a year round does not make sense

Done.

Line 457: inter-decadal should be interannual

Done.

Line 475: Fig 5 should not be cited here

Changed to Fig. 7.

Line 490: remove are

This sentence has been removed following the main comments.

Line 530: up taken should be taken up

This sentence has been removed following the main comments.

Line 536: the idea that "NO3- peaks are substantially modified after burial" undermines so much of the current manuscript that suggests that NO3- is archived once buried. Rephrase.

This sentence has been removed following the main comments.

Line 566: rephrase "form of nitrogen to skin layer"

Done.

Line 612: back to THE skin later with a day, or IS transported away

Done.

Line 624-625: see comment above about better explaining the evaportation results; and Shi et al 2019 reflect field conditions, not laboratory conditions.

Section 4.3.1 Evaporation has been removed following the main comments of referees #1 and #2.

Line 643-653: I have serious issues here with the interpretation of previous literature. First and foremost, the assumption that anthropogenic emissions of NOx are positive in d15N and natural emissions are negative in d15N is false and not up to date. Recent work shows that vehicle emissions are, in fact, negative in d15N (Miller et al., 2017, Vehicle NOx emission plume isotopic signatures: Spatial variability across the eastern United States, J. Geophys. Res. Atmos., 122, doi:10.1002/2016JD025877)/ At least three works show that fertilized soil emissions (which are not considered a natural source in emission inventories) are very negatives in d15N (Yu & Elliott, 2017, Novel method for nitrogen isotopic analysis of soilemitted nitric oxide. Environmental Science & Technology, 51(11), 6268-6278, https://doi.org/10.1021/acs.est.7b00592; Miller et al., 2018, Isotopic composition of in situ soil NOx emissions in manure fertilized cropland, Geophysical Research Letters, 45(21), 12058-12066, https://doi.org/10.1029/2018GL079619.; Li & Wang, 2008, Nitrogen isotopic signature of soil-released nitric oxide (NO) after fertilizer application. Atmospheric Environment, 42(19), 4747–4754. https://doi.org/10.1016/j.atmosenv.2008.01.042.). Geng et al. make the exact same false interpretation that anthropogenic sources are positive and therefore are ruled out in interpretation of a Summit, Greenland core – do not cite this is evidence when it is simply an unwarranted assumption. Finally, please be more precise in the language here – emission sources emit NO (except diesel engines, which can also emit NO2) or you can call it NOx; they do not "have positive d15N-NO3-" values since they do not emit nitrate, it is secondarily formed and subject to partitioning in the atmosphere, which Geng et al. invoke as a major mechanism to explain Greenland ice results, and this is wholly ignored in the current work.

We thank the reviewer for providing additional references concerning the negative isotopic signature of anthropogenic emissions of NOx. We have edited this section to include the recent work on vehicle NOx emissions and fertilised soil emissions and provided the references suggested by the reviewer. We have removed the Geng et al. reference which discards anthropogenic nitrate as a potential source to Greenland snow. When referring to the isotopic signature of emission sources, we have replaced  $\delta^{15}$ N-NO3- with  $\delta^{15}$ N-NOx. In light of the negative source signature of NOx emissions, our interpretation that anthropogenic sources do not contribute to the atmospheric  $\delta^{15}$ N-NO3- at DML remains unchanged based on i) the well-established literature in which photolysis is the major driver of atmospheric  $\delta^{15}$ N-NO3- values over low accumulation sites in East Antarctica (e.g. Frey et al., 2009;Erbland et al., 2015;Erbland et al., 2013;Shi et al., 2015;Shi et al., 2018), and ii) modelling study by Lee et al. (2014) that shows fertilised soil NOx emissions to Antarctica are minor. We modified the text as follows (lines 637-642):

Modified text: "The  $\delta^{15}$ N-NOx source signature of the main natural NOx sources (biomass burning, lightning, soil emissions;  $\delta^{15}$ N-NOx <0 ‰) is lower than anthropogenic NOx sources, which generally have positive  $\delta^{15}$ N-NOx values (-13 <  $\delta^{15}$ N-NOx < 13 ‰; e.g. (Hastings et al., 2013;Kendall et al., 2007;Hoering, 1957) except in the case of vehicle and fertilised soil NOx emissions which have negative  $\delta^{15}$ N-NOx values (-60 <  $\delta^{15}$ N-NO3- <12 ‰; Miller et al. (2017);Yu and Elliott (2017);Miller et al. (2018);Li and Wang (2008). However, a NO3- source contribution from fertilised soil NOx emissions to Antarctica is thought to be minor (Lee et al., 2014)."

Line 690-695: a range of -19 per mil to +12 per mil does not at all "nicely match" with the expected -59 per mil to -16 per mil.

Replaced "nicely match" with "falls within the range".

Line 698: add an before annual

Done.

Section 4.3.3 – seems odd to switch to section i, ii, iii, etc here when earlier subsections are numbered in series (i.e. 4.3.3.1, 4.3.3.2, etc).

Replaced i, ii, iii with 1, 2, 3.

Line 780: Concluding that photolysis is an important driver is not an interesting result given the vast evidence for this throughout the EAIS. The other conclusions are still interesting but this should not be the primary focus. Furthermore, the fact that there is less photolysis than expected is really very interesting.

This section has been removed following the main comments of referee #1.

Line 850-851: Interestingly, Geng et al. 2015 ignores surface snow work at Summit, Greenland to invoke that post-depositional processing can explain everything (Fibiger et al. (2016), Analysis of nitrate in the snow and atmosphere at Summit, Greenland: Chemistry and transport, J. Geophys. Res. Atmos., 121, 5010–5030, doi:10.1002/2015JD024187; Fibiger et al. (2013), The preservation of atmospheric nitrate in snow at Summit, Greenland, Geophys. Res. Lett., 40, 3484–3489, doi:10.1002/grl.50659.). Here, you are showing that is critical to use surface

work to best determine how to interpret ice cores. I suggest you reconsider including comparison with Geng's work here.

Done.

Line 855: depending ON the timing

Done.

Line 902: due TO photolysis-driven

Done.

Line 905: This should reference Zatko et al. not Geng et al.

Done.

Line 913: I strongly disagree with the statement that "TRANSITS doe such a good job of simulating NO-3 recycling in Antarctica" unless you do the e-folding fit first and then explore sensitivities. (I do agree that it is an excellent recommendation to use TRANSITS to assess sites that would be useful for interpreting nitrate isotopic records!)

Please see our response to the e-folding depth comment above. Based on the improved fit of the new sensitivity tests using the e-folding depth of 5 cm, we have kept this statement in the manuscript.

Line 919: unpicking is a strange word here – distinguishing?

Conclusions were rewritten following the suggestion of referee #1. "Unpicking" is no longer used.

Line 922: resulting IN an enrichment

Done.

Line 929: this conclusion would make more sense if the e-folding depth model experiments were done first.

**Done.**

Line 945: THE ground

Done.

Figure 3: how is the data averaged here?

The RACMO precipitation data is published as daily values. See the data publication for further information https://data.bas.ac.uk/full-record.php?id=GB/NERC/BAS/PDC/01137. The wind data from the AWS is hourly. We have added this to the caption.

Figure 4: x-axis on right side is misspelled

**Done.**

Figure 5: what is scenario 1 vs 2, this is never explained

This is now explained in text (lines 429-443; see comment above) and in the caption of Fig. 5.

Table 2: Archived should be >30 cm not <30 cm, correct?

Yes, this has been corrected.

**Abstract**

- 10 The nitrogen stable isotopic composition in nitrate (δ15N-NO3-) measured in polar ice cores has the potential to provide constraints on past ultraviolet (UV) radiation and thereby total column ozone (TCO) due to the sensitivity of nitrate (NO3-) photolysis to UV radiation. However, understanding the transfer of reactive nitrogen at the air-snow interface in Polar Regions is paramount for the interpretation of ice core records of δ15N-NO3- and NO3- mass concentrations. As NO3- undergoes a number of post-depositional processes before it is archived in ice cores, site-specific observations of δ15N-NO3- and air-snow transfer
- 15 modelling are necessary in order to understand and quantify the complex photochemical processes at play. As part of the Isotopic Constraints on Past Ozone Layer Thickness in Polar Ice (ISOL-ICE) project, we report new measurements of  $NO_3^$ mass concentration and  $\delta^{15}N$ -NO3- in the atmosphere, skin layer (operationally defined as the top 5 mm of the snow pack), and snow pit depth profiles at Kohnen Station, Dronning Maud Land (DML), Antarctica. We compare the results to previous studies and new data, presented here, from Dome C, East Antarctic Plateau. Additionally, we apply the conceptual one-
- 20 dimensional model of TRansfer of Atmospheric Nitrate Stable Isotopes To the Snow (TRANSITS) to assess the impact of photochemical processes NO3- recycling on that drive the archival of δ15N-NO3- and NO3- mass concentrations archived in the snow and firmpack. We find clear evidence of NO3- 
[revised manuscript text omitted]
 NO3-concentration and an enrichment in the 845N-NO3-composition with depth (Erbland et al., 2013). Once NO4 is produced by NO2-photolysis, it is expected to have a lifetime in the polar troposphere of <1 day before it is oxidised to nitrie acid (HNO2) at Dome C and South Pole (Davis et al., 2004b), and can then be redeposited to the skin layer (e.g. Mulvaney et al 1998)

100

This research at Dome C laid the foundation for Erbland et al. (2015) to derive a conceptual model of UV-photolysis induced post-depositional processes of NO3- at the air-snow interface. TRANSITS is a conceptual multi-layer 1D model which aims to represent NO2-recycling at the air-snow interface including processes relevant for NO2-snow photochemistry (UV-photolysis of NO2, emission of NO2, local oxidation, deposition of HNO2) and explicitly calculates NO2 mass concentrations and 845N

- 105 NO3-in snow. "Nitrate recycling" is the combination of NO\* production from NO3-photolysis in snow, the subsequent atmospheric processing and oxidation of NOx to form atmospheric HNOs, the deposition (dry and/or wet) of a fraction of the HNO3, and the export of another fraction. In NO2- recycling, the skin layer is an active component of the atmosphere. This recycling can occur multiple times before NOr is eventually archived below the active photic zone in ice cores (Davis et al., 2008;Erbland et al., 2015;Zatko et al., 2016;Sofen et al., 2014). We refer to atmospheric NO2-as the combination (i.e., total)
- 110 of HNO2 (gas phase) and particulate NO2 We refer to atmospheric NO2 as the combination (i.e., total) of HNO2 (gas phase) and particulate NO2 ... The desorption of HNO2 from the snow crystal reduces the NO2 concentration in the snow in coastal Antarctica (Mulvaney et al., 1998). The evaporation of HNO2 is a two-step process, which involves the recombination of NO2 HNO1 followed by a phase change to HNO2 (gas phase). First, theoretical estimates indicated that evaporation of HNO3 should preferentially remove 45N from the snow and release to the atmosphere leading to depletion in 845N-NO3- in the
- 115 residual snow pack (Frey et al., 2009). Furthermore, recent laboratory experiments showed that evaporation imposes a negligible fractionation of 815N-NOx-(!!! INVALID CITATION !!! (Erbland et al., 2013;Shi et al., 2019)). However, we find that the snow pack is enriched in δ45N-NO2- relative to the atmosphere at DML (Figs. 3 and 6) and at Dome C (section 4.3.2). This fractionation observed in field studies cannot therefore be explained by evaporation, and must be attributed to different processes. It therefore follows that evaporation must be only a minor process in the redistribution of NO2- between atmosphere
- 120 and the snow pack above the Antarctic plateau. Nitrate evaporation from the snow pack has a 15 came of -0 as determined by two independent studies (!!! INVALID CITATION !!! (Erbland et al., 2013;Shi et al., 2019)). This indicates that during NO₄⊤evaporation, the air above the snow is not replenished
  - and thus there is only a small NO2-mass loss. The isotopic fractionation of NO2-evaporation is negligible across most of Antarctica at cold temperatures of < 24 °C (Shi et al., 2019) which is the case for DML. However, evaporation of NOg at
- 125 warmer temperatures ( 4 °C) depletes the heavy isotopes of NO2-remaining in the snow, and decreases the  $\delta^{\pm S}$ NO2-and the remaining snow by a few ‰ contrary to isotope effects of photolysis. In comparison, fractionation constants associated with laboratory studies and field observations of NO4-photolysis are large: 45Gapp = -34 ‰ (Berhanu et al., 2014;Meusinger et al., 2014) and 54 <-45 emp <- 60 % (Frey et al., 2009; Erbland et al., 2013), respectively. The negative fractionation constant obtained from photolysis implies that the remaining NO2- in the skin layer snow is enriched in  $\delta^{45}$ N-NO2-. In turn, the
- atmosphere is left with the source of NO2 that is highly depleted in δ45N-NO2. This enrichment (depletion) is exactly what we 130

4

**observe in the snow pack (atmosphere) at DML (Figs. 4 and 6). The marked difference in values from the evaporation experiments and those observed in snow at Dome C allows us to separate out the isotopic signature of evaporation and photolysis processes:**

Year round measurements of NO4--mass concentrations and ö45N NO4-- in the skin layer and atmosphere at Dome C have provided insights into the annual NO4-- cycle in Antarctica (Fig. 1) Erbland et al. (2013). In the early winter, the stratosphere undergoes denitrification via formation of PSC. As PSC sediment slowly, there is a delay between the maximum stratospheric NO4-- concentration and the maximum NO4-- concentration deposited in the skin layer in late winter (Mulvaney and Wolff, 1993;Savarino et al., 2007). In spring, surface UV increases and initiates photolysis-driven post-depositional processes, which
redistribute NO4-- between the snow pack and overlying air throughout the sunlit summer season. This results in the 845N-NO4-- isotopic enrichment of the NO4-- skin layer reservoir, and maximum atmospheric NO4-- mass concentrations in October-

November. In summer, NO2-resembles a strongly asymmetric distribution within the atmosphere-snow column with the bulk residing in the skin layer and only a small fraction in the atmospheric column above.

Over longer time scales, UV-driven post-depositional processing of NO2- is also driven by changes in the degree of postdepositional loss of NO2- with greater NO2- loss during the glacial period relative to the Holocene. The observed glacialinterglacial difference in post-depositional processing of NO2- is dominated by variations in snow accumulation rate (Geng et al., 2015).

Nitrate is not preserved in the snow pack at sites with very low snow accumulation rates (i.e., Dome C: 2.5-3 cm yr-1) because snow layers remain close to the surface and in contact with the overlaying atmosphere for a relatively long time enhancing the

- 150 effect of post-depositional processes. At sites with low snow accumulation rates, the source signature of δ15N-NO2- is erased by post-depositional processe. Therefore, photolysis induced NO3- loss and δ45N-NO2- fractionation is dependent on snow accumulation. Three distinct transects from coastal Antarctica to the East Antarctic Plateau show that NO3- fractionation is strongest with decreasing snow accumulation (Shi et al., 2018;Erbland et al., 2013;Noro et al., 2018). Skin layer NO3- mass concentrations are significantly higher at low snow accumulation sites, for example ~160 ng g+ (winter) to 1400 ng g+
- 155 (summer) at Dome C compared to 50 ng g+ (winter) to 300 ng g+ (summer) at Dumont d'Urville (DDU) on the Antarctice coast. (Shi et al., 2015;Erbland et al., 2013)Furthermore, the strong inverse linear relationship between NO3- 
[revised manuscript text omitted]
 NO3- snow photochemistry (UV-photolysis of NO3-, emission of NO3, local re-oxidation, deposition of
- 245 HNO3) and explicitly calculates NO3- mass concentrations and δ15N-NO3- in snow. The term "NO3- recycling" refers to the following processes. Nitrate on the surface of a snow crystal can be lost from the snow pack (Dubowski et al., 2001), either by UV-photolysis or evaporation. UV-photolysis produces NO, NO2 and HONO while only HNO3 can evaporate. Both of these processes produce reactive nitrogen that can be released from snow crystal into the interstitial air and rapidly transported out of the snow pack to the overlaying air via wind pumping (Zatko et al., 2013;Jones et al., 2000;Honrath et al., 1999;Jones et al.,
- 2001). Here, NO2 is either oxidised to HNO3, which undergoes wet or dry deposition back to the skin layer within a day, or is transported away from the site (Davis et al., 2004a). If HNO3 is re-deposited to the skin layer, it is available for NO3- photolysis and/or evaporation again. Any locally produced NO2 and NO3- that is transported away from the site of emission represents a loss of NO3- from the snow pack. Nitrate recycling can occur multiple times before NO3- is eventually archived below the active photic zone in ice cores (Davis et al., 2008;Erbland et al., 2015;Zatko et al., 2016;Sofen et al., 2014).
- 255 Year round measurements of NO3- mass concentrations and δ15N-NO3- in the skin layer and atmosphere at Dome C have provided insights into the annual NO3- cycle in Antarctica (Fig. 1; Erbland et al. (2013). In the early winter, the stratosphere undergoes denitrification via formation of PSC. As PSC sediment slowly, there is a delay between the maximum stratospheric NO3- mass concentration and the maximum NO3- mass concentration deposited in the skin layer in late winter (Mulvaney and Wolff, 1993;Savarino et al., 2007). In spring, surface UV increases and initiates photolysis-driven post-depositional processes,
- 260 which redistribute NO3- between the snow pack and overlying air throughout the sunlit summer season. This results in the δ15N-NO3- isotopic enrichment of the NO3- skin layer reservoir, and maximum atmospheric NO3- mass concentrations in October-November. In summer, NO3- resembles a strongly asymmetric distribution within the atmosphere-snow column with the bulk residing in the skin layer and only a small fraction in the atmospheric column above.

- Nitrate is not preserved in the snow pack at sites with very low snow accumulation rates (i.e., Dome C: 2.5 3 cm yr-1) because
  snow layers remain close to the surface and in contact with the overlaying atmosphere for a relatively long time enhancing the effect of post-depositional processes. At sites with low snow accumulation rates, the source signature of δ15N-NO3- is erased by post-depositional processe. Therefore, photolysis induced NO3- loss and δ15N-NO3- 
[revised manuscript text omitted]
 NO3" as the combination (i.e., total) of HNO3 (gas phase) and particulate NO3" and is represented by the NO3" mass concentrations measured on aerosol filters.

The high-volume aerosol sampler was located 1 m above the snow surface at the flux site at the DML site (Fig. 2c), where a total of 35 aerosol filters were sampled daily between 3 and 27 January 2017. In addition, we coordinated an intensive 4-hour sampling campaign in phase with Dome C, East Antarctica (Fig. 2) between 21 and 23 January 2017. At Dome C, a high-volume aerosol sampler wais located on the roof of the atmospheric shelter (6 m above the snow surface), where a total of 12 samples were collected. At DML, loading and changing of aerosol collection substrates was carried out in a designated clean area. Aerosol laden filters were transferred into individual double zip-lock plastic bags immediately after collection and stored frozen until analysis at the British Antarctic Survey (BAS; major ions) and IGE (NO3- isotopic composition). For the

360 atmospheric NO3- work, three types of filter blanks were carried out; i) laboratory filter blanks (n = 3; Whatman GF/A filters that underwent the laboratory procedures without going into the field), ii) procedural filter blanks (DML: n = 4; Dome C: n = 1; filters that had been treated as for normal samples but which were not otherwise used; once a week, during daily filter change-over, a procedural blank filter was mounted in the aerosol collector for 5 min without the collector pump in operation – this type of filter provides an indication of the operational blank associated with the sampling procedure), and iii) 24 h exposure filter blanks sampled at the beginning and end of the field campaign (DML: n = 2; Dome C: n = 1; filters treated like

a procedural blank but left in the collector for 24 h-without switching the collector on). All samples were kept frozen below - 20 °C during storage and transport prior to analysis.

In addition, skin layer and aerosol samples whave beenere sampled continuously at Dome C over the period 2009-2015 following Erbland et al. (2013);Frey et al. (2009). The sampling resolution for skin layer wais every 2-4 days, and weekly for aerosol samples. Data from 2009-2010 have previously been published by Erbland et al. (2013), and we report the 2011\_2015 data here (Fig. 1).

**2.3 Major ion mass concentrations in snow and aerosol**

Aerosol Atmospheric NO3- and other major ions were extracted in 40 mL of ultra-pure water (resistivity of 18.2 MΩ; Milli-Q water) by centrifugation using Millipore Centricon® Plus-70 Filter Units (10 kD filters) in a class-100 clean room at the BAS.
Major ion mass concentrations in DML snow samples were determined in an aliquot of melted snow from skin layer and snow pit samples, and aerosol extracts by suppressed ion chromatography (IC) using a DionexTM ICS-4000 Integrated Capillary HPICTM System ion chromatograph. A suite of anions, including NO3-, chloride (Cl-), methanesulfonic acid (MSA) and sulphate (SO42-), were determined using an AS11-HC column and a CES 500 suppressor. Cations, including sodium (Na+), were determined using a CS12A column and a CES 500 suppressor. During the course of the sample sequence, instrumental blank solutions and certified reference materials (CRM; ERM-CA616 groundwater standard and ERM-CA408 simulated

rainwater standard; Sigma-Aldrich) were measured regularly for quality control and yielded an accuracy of 97 % for NO3-. Nitrate mass concentrations in Dome C samples were determined by colorimetry at IGE following the procedure described in Frey et al. (2009). Blank concentrations for exposure blank, procedural blank and laboratory blank and detection limits are reported in Table S1. The non-sea-salt sulphate (nss-SO42-) fraction of SO42- was obtained by subtracting the contribution of sea-salt-derived SO42- from the measured SO42- mass concentrations (nss-SO42- = SO42- - 0.252 × Na+, where Na+ and SO42- are

the measured concentrations in snow pit samples and 0.252 is the SO42-/Na+ ratio in bulk seawater (Keene et al., 1986).

**2.4 Nitrate isotopic composition in snow and aerosol**

390

Samples were shipped frozen to IGE where the NO3- isotope analysis was performed. The denitrifier method was used to determine the stable NO3- isotopic composition in samples at IGE following Morin et al. (2008). Briefly, samples were preconcentrated due to the low NO3- mass concentrations found in the atmosphere and snow over Antarctica. To obtain 100 nmol of NO3- required for NO3- isotope analysis, the meltwater of snow samples and aerosol extracts were were sorbed onto 0.3 mL

[revised manuscript text omitted]

**2.6 Nitrate photolysis rate coefficient**

Hemispheric or  $2\pi$  spectral actinic flux from 270 to 700 nm was measured at 2.1 m above the snow surface using an actinic flux spectroradiometer (Meteorologieconsult GmbH; Hofzumahaus et al. (2004).  $2\pi$  NO3- photolysis rate coefficients J(NO3-) were then computed using the NO3- absorption cross section and quantum yield on ice estimated for -30 °C from Chu and Anastasio (2003). The mean  $2\pi$  J-NO3- value at DML during January 2017 was 1.02 x 10-8 s-1, and 0.98 x 10-8 s-1 during the 1

- to 14 January 2017 period. The observed 2π J(NO3-) at DML was a factor of three lower than Dome C (2.97 x 10-8 s-1; 1 to 14 January 2012) which was previously measured using the same instrument make and model, and at the same latitude (Kukui et al., 2013). Only ~5 % of the apparent inter-site difference can be attributed to TCO being ~25 DU larger at DML (306 DU) than at Dome C (287 DU) during the comparison period. The remainder was possibly due to greater cloudiness at DML and differences in calibration. In this study, the observed 2π J(NO3-) is used to estimate the snow emission flux of NO2.

**2.7 Air-snow transfer modelling**

Table S3.

435

In order to evaluate the driving parameters of isotope air-snow transfer at DML we used the TRANSITS model (Erbland et al., 2015) to simulate snow depth profiles of NO3- mass\_concentration and δ15N-NO3- and compare them to our observations. TRANSITS is a conceptual multi-layer 1D model which aims to represent NO2--recycling at the air snow interface including processes relevant for NO2--snow photochemistry (UV photolysis of NO3-, mission of NO3- local oxidation, deposition of HNO2) and explicitly calculates NO2- mass concentrations and δ4+N-NO3- 
[revised manuscript text omitted]
 δ15N-NO3- is about 25 ‰ higher than atmospheric δ15N-NO3-. Nitrate mass concentration and δ15N-NO3- composition data for aerosol, skin layer and

**550 3.4 Archived nitrate mass concentration and isotopic composition**

snow pit samples are available in Winton et al. (2019b).

We calculate archived values of NO3- mass concentration and  $\delta^{15}$ N-NO3- which represent the archived mass fraction and isotopic composition reached below the photic zone. Archived values were calculated by averaging the NO3- mass concentration and  $\delta^{15}$ N-NO3- values below the photic zone, i.e., 30–15 cm (section 4.44.4). The archived NO3- mass

concentration and δ15N-NO3- values for snow pit A were 60 ng g-1 and 50 ‰,-the and the archived NO3- mass concentration for snow pit B was 50 ng g-1. Note that no δ15N-NO3- values were measured below 30 cm in snow pit B. ObservedThese measured δ15N-NO3- values are half of those expected for a site with a snow accumulation rate of 6 cm yr-1 (w.e.) in the spatial survey from Erbland et al. (2013) (Table 2).

**3.5 Nitrate mass flux estimates**

The total deposition flux (F) of  $NO_3^-$  is partitioned into wet and dry deposition fluxes (Fwet and Fdry respectively; Eq. (3)), and 560 can be estimated using the measured mass concentration of  $NO_3$  in the snow pack ( $C_{snow}$ ) and the local snow accumulation rate (A; Eq. (4)). Estimates of the dry deposition rate (Fdry) of NO3- were calculated using Eq. (5) using the atmospheric mass concentrations of NO3- (Caerosol) and a dry deposition velocity (Vdry deposition) of 0.8 cm s-1, and are reported in Table S5. This deposition velocity is based on the dry deposition of HNO3 at South Pole (Huey et al., 2004) which has a similar snow accumulation rate (6.4 cm yr-1 (w.e.); Mosley-Thompson et al. (1999)) to DML. Other estimates of dry deposition velocities include 0.05\_-\_0.5 cm s-1 for HNO3 over snow (Hauglustaine et al., 1994;Seinfeld and Pandis, 1998), 1.0 cm s-1 for NO3- over the open ocean (Duce et al., 1991), and an apparent deposition velocity of 0.15 cm s-1 for summer HNO3 at Dome C (Erbland et al., 2013). The estimated apparent NO3- deposition velocity at Dome C is low because of the strong recycling of NO3- on the polar plateau in summer, i.e., reactive nitrogen is re-emitted from the skin layer to the atmosphere. Although gas phase HNOg and particulate NO3 have different dry deposition rates Thus, the dry deposition velocity at DML is likely to lie between \_ 570 0.15 and 0.8 cm s-1. We assume that a constant deposition velocity throughout the campaign is appropriate for DML.  $F = F_{wet} + F_{dry}$ (3)

| $C_{snow} = F / A$                                          | (4) |
|-------------------------------------------------------------|-----|
| $F_{dry} = \mathbf{C}_{acrosol} \mathbf{V}_{drydeposition}$ | (5) |

575 Note that Eq. (4) does not take into account post-depositional processes of non-conservative ions, such as NO3-. We follow the approach of Erbland et al. (2013) who use an archived NO3- mass flux (Fa) to represent the downward NO3- mass flux which escapes the photic zone towards deeper snow layers. Using simple mass balance, we can then estimate the mass flux of NO3- (Fre-emit), which is re-emitted from the snow pack to the overlaying atmosphere (Eq. (6)).  $F_{re-emit} = F - F_a$  (6)

580 Taking this a simple mass balance approach, a schematic of NO3- mass fluxes for two scenarios are illustrated in Fig. 5a. Scenario 1 is an average annual budget for DML (Fig. 5a). As the atmospheric campaign did not cover an entire annual cycle, we use estimates of atmospheric NO3- mass fluxes at DML reported by Pasteris et al. (2014) and Weller and Wagenbach (2007) of 43 and 45 pg m-2 s-1, respectively, as a-year round dry deposition fluxes. Due to the linear relationship of ice core NO3- mass concentrations with the inverse accumulation, the authors assume that the magnitude of the dry deposition flux is homogenous over the DML region. Mean annual mass concentrations of NO3- in our snow pits suggest a total NO3- deposition mass flux of

585 over the DML region. Mean annual mass concentrations of  $NO_3^-$  in our snow pits suggest a total  $NO_3^-$  deposition mass flux of 110 pg m-2 s-1 and therefore a wet deposition mass flux of 65 pg m-2 s-1.

18

However, at relatively low snow accumulation sites where photolysis drives the fractionation of NO3- from the surface snow to atmosphere (Frey et al., 2009), it is necessary to take into account the skin layer in the NO3- mass\_flux budget as this airsnow interface is where air-snow transfer of NO3- takes place. In scenario 2, wWe usutilisee the available NO3- 
[revised manuscript text omitted]
. TAlthough we only have measurements of  $\delta^{15}N$ -NO3-in January, the simulated atmospheric  $\delta^{15}N$ -NO3 values in the base case for January, are greater than the measurements available from this study, while the  $\delta^{15}N$ -NO3 values in the 5 cm EFD case fall within the range of observations. TThe annual cycle of simulated atmospheric  $\delta^{15}N$ -NO3 for the 5 cm EFD case shows a 40-50 ‰ dip in spring to -43-2 ‰ from winter values which coincides with the simulated atmospheric NO3 mass concentration increase in spring (Fig. 6c). The highest simulated atmospheric  $\delta^{15}N$ -
- NO3- values (7 ‰) occur in winter, for both scenarios. In the skin layer, the simulated NO3- mass concentrations are an order of magnitude greater than our observations in January and we outline possible reasons for this discrepancy in the discussion (section 4.1). The simulated annual cycle of NO3- mass concentrations in the skin layer steadily rise in spring and reach a peak in January when they begin to decline to the lowest mass concentration in winter (Fig. 6d). Simulated skin layer δ15N-NO3- values in January for the base case are ~10 ‰ higher than our highest observations for that month but the average January
- 660 value in the 5 cm EFD case (c7 ‰) falls in the range of observed values (-10 ‰) (Fig. 6e). For the 5 cm EFD case, tFhey begin to decrease by 24,30 ‰- in spring at the same time as atmospheric δ15N-NO3- values decrease. In October and November, the skin layer δ15N-NO3- values begin to rise up to c1144 ‰ in February in the 5 cm EFD case in February. The seasonality of simulated NO3- mass concentrations and δ15N-NO3- values in the atmosphere and skin layer at DML isis
- consistent with Dome C (Fig. 1). Similar to Dome C, simulated NO3- mass concentrations in the skin layer start to rise two 665 months earlier than atmospheric NO3- mass concentrations and the summer maximum is later. While the seasonality of  $\delta^{15}$ N-
- NO3- in the skin layer and atmosphere co-vary, simulated skin layer δ15N-NO3- values are enriched relative to atmospheric values on average by R(1-ss.
   The simulated NO3- mass concentrations and δ15N-NO3- values in the snow pitdepth profiles are illustrated in Fig. 7. The e-folding depth sensitivity tests show that a deeper e-folding depth i) increases the δ15N-NO3- enrichment in the photic zone, and

- 670 ii) increases in the mean annual archived δ15N-NO3- value (Fig. 7a). Out of the e-folding depths explored in the sensitivity analysis, an e-folding depth in the range of that observed at DML, i.e., 2 5 cm, has the closest mean annual δ15N-NO3- value to the observations (Fig. 7a). -BBoth the simulated-depth profile of simulated NO3- mass concentration and δ15N-NO3- in the base case for an accumulation rate of 6 cm yr-1 (w.e.) show seasonal variability in the first year with a range of of 380 ng g-1 and 20 ‰, which decreases with depth to a range of 95 ng g-1 and 10 ‰ in the fourth year. In comparison, in the 5 cm EFD
- 675 case, the seasonality of δ15N-NO3- and NO3 mass concentrations in the first year ranges from 290 ng g-1 and 40 ‰ to 75 ng g-1 and 20 ‰ in the fourth year (Fig. 7a). For the base case scenario, t the simulated archived (i.e., annual average of the first / year below 1 m) NO3- mass concentration, δ15N-NO3-, and NO3- mass flux values are 120 ng g-1, 130 ‰, and 210 pg m-2 yr-1, respectively. The simulated annual average 15εapp is -19 ‰ for the top 30 cm (i.e., active photic zone with an e-folding depth of 10 cm). In comparison, in the 5 cm EFD case, the simulated archived NO3- mass concentration, δ15N-NO3-, and NO3- mass /
- 680 flux values are 280 ng g-1, 50 ‰, and 480 pg m-2 yr-1, respectively. The simulated annual average 15 $\epsilon_{app}$  is -11 ‰ for the top 30 / cm. The 5 cm EFD case falls within the range of observations for  $\delta^{15}$ N-NO3- (Figs. 7a) but is significantly higher than the / observed NO3- mass concentrations (Fig. 7c). Also plotted in Figs. 7b-c are the simulated NO3- mass concentration and  $\delta^{15}$ N-

21

Formatted: Not Highlight Formatted: Pattern: Clear Formatted: Not Highlight Formatted: Not Highlight Formatted: Not Highlight Formatted: Pattern: Clear Formatted: Not Highlight Formatted: Pattern: Clear Formatted: Not Highlight Formatted: Not Highlight Formatted: Not Superscript/ Subscript  $NO_3^-$  depth profiles for accumulation rates of 2.5 cm yr-1 (w.e.) and 100-11 cm yr-1 (w.e.) for the 5 cm EFD case. As the accumulation rate increases, the annual layers of  $\delta^{15}N$ -NO3- become thicker, the seasonal amplitude increases, the mean annual

- $\delta^{15}$ N-NO3- value decreases, and there is less  $\delta^{15}$ N-NO3- enrichment in the photic values in the top 10 one decreases zone (Fig. 7b). At very low snow accumulation rates, the seasonal cycle is smoothed, as in the case of Dome C (Fig. 7b). A similar pattern is observed for the simulated NO3- mass concentrations with depth: seasonal cycles of NO3- mass concentrations are more pronounced at higher snow accumulation rates, while inter-annual variability is smoothed at very low accumulation rates such as Dome C (Fig. 7c). The relationship between the snow accumulation rate and  $\delta^{15}$ N-NO3- is non-linear (Figs b-c).
- 690 Overall the TRANSITS modelling shows that the i) simulated values in the base case scenario are higher than the 5 cm EFD case, and ii) TRANSITS modelling simulations using the observed e-folding depth of 5 cm are good fit with observations. Differences between the simulated  $\delta^{15}N-NO_3^-$  depth profiles for the two cases and observed  $\delta^{15}N-NO_3^-$  could be due to uncertainties in a number of factors, for example: i) a shallower e-folding depth than modelled (section 4.5.1), ii) lower JNO3- values (NO3- photolysis rate), which are related to a lower e-folding depth, and would lead to less enrichment of  $\delta^{15}N-NO_3^-$  in
- 695 the snow pack (section 4.3.2), iii) higher atmospheric NO3- input, however δ15N-NO3- values are not sensitive to variable atmospheric NO3- mass concentrations (Erbland et al., 2015), and/or iv) variable snow accumulation which would shift the oscillations to the correct depth and lower the mean δ15N-NO3- values below the photic zone (section 4.5.2). These differences are further addressed in section 4.5.
- The simulated archived (i.e., annual average of the first year below 1 m) NO2=mass concentration, 815N-NO2; and NO2=mass 700 flux values are 120 ng g+, 130 ‰, and 210 pg m-2 yr-1, respectively. The simulated annual average 15eapp is -19 ‰ for the top 30 cm (i.e., active photic zone with an e-folding depth of 10 cm).

**4 Discussion**

**4.1 Validation of results**

- Our January 2017-NO3- measurements at DML agree well with-values reported in the literature, and largely with largely with
  the simulated 5 cm EFD case results from the TRANSITS model with the exception of except for the skin layer NO3- mass concentrations. While we made the first measurements of atmospheric, skin layer and snow pit 845N-NO3-, and skin layer NO3- mass concentrations at DML, there are published measurements of NO3- mass concentrations in snow pits and In particular, oeur NO3- concentrations observations in snow pits agree well with published measurements of ANO3- mass concentrations of NO3- mass concentrations in snow pits and In particular, oeur NO3- concentrations observations in snow pits agree well with published measurements of NO3- mass concentrations of atmospheric NO3- mass concentrations in snow pits at DMLthose (Weller et al., 2004). While our January 2017 observations of atmospheric NO3- mass concentrations in January 2017 are lower than those observed in -Our atmospherie mass NO3- concentrations in January 2017 are lower than those observed in -Our atmospherie mass NO3- concentrations in January 2017 are lower than those observed in -Our atmospherie mass NO3- concentrations in January 2017 are lower than those observed in -Our atmospherie mass NO3- concentrations in January 2017 are lower than those observed in -Our atmospherie mass NO3- concentrations in January 2017 are lower than those observed in -Our atmospherie mass NO3- concentrations in January 2017 are lower than those observed in -Our atmospherie mass NO3- concentrations in January 2017 are lower than those observed in -Our atmospherie mass NO3- concentrations in January 2017 are lower than those observed in -Our atmospherie mass NO3- concentrations in January 2017 are lower than those observed in -Our atmospherie mass NO3- concentrations in January 2017 are lower than those observed in -Our atmospherie mass

22

| Farmer Marked, Mark I Balak Balak |  |
|-----------------------------------|--|
| Formatted: Not Highlight          |  |
| Formatted: Not Highlight          |  |
|                                   |  |
| Formatted: Highlight              |  |
| Formatted: Pattern: Clear         |  |
| Formatted: Pattern: Clear         |  |
|                                   |  |

|     | Overall, For the skin layer, the simulated NO3 mass concentrations results from TRANSITS are greater than our January                                  |
|-----|--------------------------------------------------------------------------------------------------------------------------------------------------------|
| 715 | observations, particularly the skin layer NO2-mass concentrations (Fig. 6d). The discrepancy between the significantly higher                          |
| 1   | simulated NO3 - mass concentrations than observations in the skin layer was also found at Dome C. Erbland et al. (2015)                     |
|     | suggested that this discrepancy could be related to either a sampling artefact, snow erosion or a modelled time response to                            |
|     | changes in past primary inputs. We provide an alternative explanation for the extremely high simulated $\mathrm{NO}_3^-$ mass                          |
|     | concentrations in the skin layer using daily measurements of NO3 - mass concentration in diamond dust and hoar frost collected              |
| 720 | daily from Polyvinyl chloride (PVC) sheets at Dome C in summer 2007/08, i.e. new deposition. New deposition of diamond                                 |
| I   | dust had $NO_3$ mass concentrations up to 2000 ng g -1 , which is four times greater than that observed in natural snow from the            |
|     | skin layer at the same time (Fig. S4). Similarly, new deposition of hoar frost had NO3 - mass concentrations up to 900 ng g -1 , |
| I   | which is three times greater than the skin layer snow. The formation of surface hoar frost occurs by co-condensation, i.e. the                         |
|     | simultaneous condensation of water vapour and NO3 - at the air-snow interface. Recent modelling suggests that co-condensation               |
| 725 | is the most important process explaining NO3 - incorporation in snow undergoing temperature gradient metamorphism at Dome                   |
|     | C (Bock et al., 2016). Diamond dust can also scavenge high concentrations of HNO3 at Dome C (Chan et al., 2018).                                       |
|     | Furthermore, the top layer of the snow pack is only 1 mm thick in the TRANSITS model, whereas our observations of the skin                             |
|     | layer are 5 mm thick. Due to the photochemical loss of NO3- mass concentrations with depth, the highest NO3- mass                                      |
|     | concentrations are expected in the top 1 mm layer which is the layer best in equilibrium with the atmosphere. Here, extremely                          |
| 730 | high mass concentrations of NO3 - from new deposition from diamond dust and hoar frost are also found. In summary, In                       |
|     | summary, it is likely that we do not measure such high NO3 mass concentrations in hoar frost and diamond dust values in the                            |
|     | skin layer because of sampling artefacts or blowing snow, which can dilute or remove the diamond dust and hoar frost-which                             |
|     | is where we would expect the highest concentrations due to the exponential decay of NO3 - with depth (Fig. S4). If indeed the               |
|     | higher simulated values in the skin layer can be explained by hoar frost and diamond dust, then we can have greater confidence                         |
| 735 | in the depth profile of NO3 - concentration. It is interesting to note that these higher simulated values in the skin layer do not          |
|     | impact the simulated depth profiles (Fig. 57). In summary, it is likely that we do not measure such high hoar frost and diamond                        |
|     | dust values in the skin layer because of sampling artefacts or blowing snow, which can dilute or remove the diamond dust and                           |

While not yet observed elsewhere on the Antarctic continent, over the short intensive sampling period at DML we 740 observe significant variability in NO₂-mass concentrations and δ15N NO₂-values that resembles a diurnal cycle. Over 4 hours, the skin layer NO₂-mass concentrations varied by 46 ng g+, the skin layer δ15N NO₂-by 21 ‰, and the atmospheric δ14N NO₂- by 18 ‰. Other coastal studies have attributed daily variability to individual storm events

hoar frost.

(Mulvaney et al., 1998;Weller et al., 1999). We note that the sampling duration is too short to confirm any diurnal patterns but it would be interesting to investigate this further in future work.

**745 4.2 Nitrate deposition**

Here we discuss the various processes in which NO3- can be deposited to the skin layer at DML. As we have just one month of atmospheric and skin layer data, our ability to look at the deposition on seasonal seales is limited, however we provide new insights into the austral summer deposition processes.

- 750 While it is common to measure nitrogen species in snow and aerosol samples as the NO3- ion using ion chromatography, nitrogen species can be deposited in various forms either by wet or dry deposition to the skin layer. We note that organic NO3- plays are little role in determining snow concentrations (Jones et al., 2007; Wolff et al., 2008), and as such we focus our discussion on inorganic NO3-. The various nitrogen species include, i) a neutral salt (NO3- co-deposition with sea our discussion on literative (2009). NO 3- is (ICDE) and the salt (NO3- co-deposition with sea our discussion of the sea out of the salt (NO3- co-deposition with sea out of the salt (NO3- co-deposition sea out of the salt (NO3- co-deposition with sea out out of the salt (NO3- co-deposition with sea out of the salt (NO3- co-deposition sea out of the salt (NO3- co-
- salt or mineral dust; Wolff et al. (2008)), ii) NO3- in air (HNO3 in gas phase plus particulate NO3-). Following the terminology of Erbland et al. (2013), this is referred to as "atmospheric NO3- and is represented by the NO3- mass concentrations measured on our aerosol filters. Atmospheric NO3- can either be deposited as dry deposition by adsorption to the snow surface as HNO3 has a strong affinity for ice surfaces (Abbatt, 2003;Huthwelker et al., 2006) or seavenged by precipitation as wet deposition, and iii) co-condensation of HNO3 and water vapour onto snow crystals (Thibert and Domine, 1998).
- 760 Depending on the deposition pathway, NO3- can either be predominantly incorporated into the bulk snow crystal or be adsorbed onto the surface of the snow crystal. Deposition pathways include co-condensation (formation of surface hoar frost), riming (deposition of supercooled fog droplets), and adsorption of HNO3-onto the snow crystal surface (dry deposition) (Röthlisberger et al., 2002). Both co-condensation (Bock et al., 2016) and dry deposition of HNO3, at very cold temperatures, can elevate NO3--mass concentrations in the skin layer. Furthermore, trace nitrogen impurities present in the interstitial air in the norous snow nack may be incorporated in snow crystal. While can engine of NO3-
- 765 present in the interstitial air in the porous snow pack may be incorporated in snow crystals. While scavenging of NO3- by snow (wet deposition) occurs sporadically throughout the year, dry deposition of particulate NO3- or surface adsorption may take place continuously throughout the year. We see both of these deposition processes taking place during January 2017.

**4.2.1 Wet and dry deposition**

- 770 Here we discuss the various processes in which NO3- can be deposited to the skin layer at DML. Firstly, we first look at atmospheric NO3- deposition in relation to the source region of the air mass. The mean annual wind direction at the site is 65° (Figs. 3 and S5). There is an excursion from this predominant wind direction between 19 22 January, where the wind direction switches to the southwest, i.e., atmosphere transport from the plateau. We do not see elevated NO3- mass concentrations during this period nor do we see a marked difference in isotopic signature that is similar to Dome C at this time (Fig. 4). This, in line
- 775 with air mass back trajectories (not shown) suggests that transport of NO3- re-emitted from inland sites of the Antarctic, carrying a distinctively enriched δ15N-NO3- signature, did not influence DML during our campaign. We can also rule out any downwind contamination from the station.

Secondly, we use Precipitation at DML can occur either through sporadic cyclonic intrusions of marine air masses from the adjacent ocean associated with large amounts of precipitation, or clear sky diamond dust that contributes smaller amounts to

780 the total precipitation (Schlosser et al., 2010). Overall, extreme precipitation events dominate the total precipitation (Turner et

al., 2019). In austral summer, the transport of marine aerosol to DML is mediated by two synoptic situations, i) low-pressure systems from the eastern South Atlantic associated with high marine aerosol concentrations, and ii) persistent long-range transport that provides background aerosol deposition during clear sky conditions (Weller et al., 2018). Weller et al. (2018) suggest that dry deposition of marine aerosol is dominant over wet deposition at DML. In contrast, Dome C receives predominantly diamond dust, and thus aerosol deposition is different there.

- More specifically, precipitation during our sampling campaign in January 2017 was relatively low compared to previous years.
   Modelled daily precipitation at the nearest Regional Atmospheric Climate Model (RACMO2; Van Meijgaard et al. (2008))
   grid point (75.0014°S, 0.3278°W) is illustrated in Fig. 3a. The largest precipitation event of the month was on 1 January (0.27 mm) resulting from a low-pressure system in the South Atlantic (Fig. S5). For the rest of the month, half of the days had zero
   790 precipitation and the other half had very little precipitation (~0.05 mm per day).
- We use mthe odelled daily precipitation at the nearest Regional Atmospheric Climate Model (RACMO2; Van Meijgaard et al. (2008) grid point (75.0014°S, 0.3278°W; Fig. 3a) RACMO2 daily precipitation data to identify whether the influence of cyclonic intrusions of marine air masses <del>provide to</del> wet deposition of NO3- to the site in January. We observe that .- In the skin layer, we observe that NO2- mass concentrations and other sea salt ions co-vary (Fig. S6) suggesting similar deposition
- 795 pathways of these ions\_Some peaks in the skin layer NO3- mass concentration are accompanied by fresh snow laden with relatively high sea salt aerosol mass concentrations and atmospheric NO3- mass concentrations, for example on 1, 13, and 18 January 2017 (Fig. S6). In the skin layer, we observe that NO3- mass concentrations and other sea salt ions co-vary (Fig. S6) suggesting similar deposition pathways of these ions. Such deposition events have also been observed on the Antarctic coast (Wolff et al., 2008). During the formation of precipitation, essentially all HNO3 is removed from the gas phase due to its high
- 800 solubility in liquid clouds (Seinfeld and Pandis, 1998). Therefore, HNO2 can be scavenged from the atmosphere and deposited as NO2- in the skin layer. The uptake of HNO2 onto the snow and ice crystal surface during and after precipitation can also contribute further to the NO2- mass concentrations found in the skin layer. On someWhereas on other precipitation days, we observe lower atmospheric NO3- mass concentrations and higher skin layer NO3- mass concentrations that could be a result of HNO3 scavenging. Mulvaney et al. (1998) observed higher skin layer concentrations in days when there was little snow
- 805 accumulation and concluded that NO2- is directly up taken onto the surface by dry deposition of particulate NO2- and surface adsorption of HNO2 (gas-phase) (e.g. Mulvaney et al., 1998). With only one month of data it is difficult to see the impact of wet deposition on the NO3- mass concentration in the skin layer; i.e. whether fresh snowfall dilutes the NO3- mass concentration in the skin layer or whether it scavenges HNO3 (gas-phase) resulting in higher mass concentrations of NO3- in the skin layer. Most likely both processes are occurring. We note that due to post-depositional processes (section 3) any short-term signals
- 810 observed in the skin layer are unlikely to be preserved. Even at the South Pole where the snow accumulation rate is slightly higher (8.5 cm yr+(w.e.); (Mosley-Thompson et al., 1999) than DML deposition, NO2-peaks are substantially modified after burial (Dibb and Whitlow, 1996).

Thirdly, we

785

investigate daily changes in the atmospheric and skin layer NO3+ mass concentrations and δ15N-NO3- over the campaign to see
 the influence of dry deposition, by adsorption of atmospheric NO3+ to the snow surface, 4.2.2 Dry deposition NO3+ is directly up taken onto the surface by dry deposition of particulate NO3+ and surface adsorption of HNO2 (gas phase) (e.g. Mulvaney et al., 1998)

up taken onto the surface by dry deposition of particulate NO3 and surface adsorption of HNO3 (gas phase) (e.g. Mulvaney et al., 1998) In order to investigate dry deposition of NO3, we first look at atmospheric NO3 in relation to the wind direction and air mass back trajectories. The mean annual wind direction at the site is  $65^{\circ}$ , and January 2017 is no exception (Figs. 3 and S7). There

820 is an excursion from the predominant wind direction between 19-22 January, where the wind direction switches to the southwest. Although there are no studies indicating fractionation of δ15N-NO3- in the atmosphere during atmospheric transport from the plateau to the coast, we do not see elevated NO3- mass concentrations during this period nor do we see a marked difference in isotopic signature that is similar to Dome C at this time (Fig. 4). This, in line with air mass back trajectories (not shown) suggests that long range transport of NO3- re-emitted from inland sites of the Antarctic did not reach DML during our campaign. We can also rule out any downwind contamination from the station.

High concentrations of sea salt and mineral dust can promote the conversion of HNO3 (gas-phase) to aerosol, as well as trapping NO3- (gas phase) on salty snow surfaces. We see a relationship between sea salt aerosol and atmospheric NO3- (R2= 0.59; p=<0.001) suggesting that even 550 km inland from the coast sea salt could promote the conversion of HNO3- to atmospheric NO3-, although we acknowledge that our filters capture both aerosol NO3- and HNO3, and sea salt concentrations are much higher at Halley and coastal Antarctica where this mechanism sporadically occurs (Wolff et al., 2008).

- Seavenging of atmospheric NO3- is largely responsible for theon the high mass concentrations observed in the skin layer. Temporal vVariation in the mass concentration and isotopic signature of aerosol and surface snow at DML over January 2017 suggests atmospheric NO3- is the source of NO3- to the skin layer. Throughout the month, the increase in the skin layer mass concentration of summer NO3- appears to be closely related to the decrease in the atmospheric NO3- mass concentrations (Fig.
- 3). There is a lag between atmospheric and skin layer NO3- i.e. atmospheric NO3- mass concentrations precede skin layer NO3- mass concentrations by day or two, however a longer time series is required to confirm this. The lag suggests that atmospheric NO3- is a source of NO3- to the skin layer, in line with Dome C where the underlying snow pack is the dominant source of NO3- to the skin layer viaphotolytic recycling and re-depositionthe-overlying air in summer. Furthermore, as atmospheric NO3- is deposited to the snow surface, 15N is preferentially removed first leaving the air isotopically depleted relative to the isotopically enriched snow (Frey et al., 2009). Figs. 3-4 illustrates that the δ15N-NO3- in the atmosphere is depleted with respect
- to the δ15N-NO3- in the skin layer snow. In the short time series, there are some periods where the δ15N-NO3- in the snow and atmosphere are in phase, for example, 3--13 January 2017. During other periods, the δ15N-NO3- in the snow and atmosphere switch to being out of phase emphasising NO3- isotopic fractionation during those periods. Both HNO3 and peroxynitric acid (HNO4) can be adsorbed to the snow surface in tandem (Jones et al., 2014), and although we have no direct measurements of these during the campaign, based on previous studies we suggest that HNO3 is the dominantly adsorbed most likely form of
  - nitrogen to the skin layer (Jones et al., 2007;Chan et al., 2018),

**Formatted: Normal**

We conclude that HNO3+ scavenging, adsorption and cyclonic intrusions of marine air masses deliver NO3, to the skin layer at DML in summer. During the campaign, deposition is not influenced by the transport of airmasses from the polar plateau which carry a distinct atmospheric  $\delta^{15}$ N-NO3, signature. Furthermore, the adsorption of HNO2 on ice surfaces is temperature

- 850 dependent with higher uptake at lower temperatures (Abbatt, 1997;Jones et al., 2014). However, there is only a relatively small temperature difference between Dome C and DML (summer mean temperature -30 °C and -25 °C respectively) which is not enough to drive a large difference in HNO, uptake (Jones et al., 2014). In addition, the uptake is not dependent on the HNO, eoneentration in the air (Abbatt, 1997). However, the seasonal temperature difference at an individual site (i.e., DML or Dome C) is far greater, which could allow a seasonal dependence on the uptake and loss of NO2- in the skin layer, which results in the retention of a greater proportion of NO2- in summer (Chan et al., 2018). We note that due to post-depositional processes
- (section 3) any short-term signals observed in the skin layer are unlikely to be preserved. Even at the South Pole where the snow accumulation rate is slightly higher (8.5 cm yr+(w.e.); (Mosley Thompson et al., 1999) than DML deposition, NO2= peaks are substantially modified after burial (Dibb and Whitlow, 1996).

**860 4.2.23 Annual cycle Temporal variability of nitrate deposition**

The seasonality of simulated skin layer and atmospheric  $NO_3^-$  mass concentrations at DML matches observations at other Antarctic sites. We use the simulated annual cycle of  $NO_3^-$  from TRANSITS model to describe the seasonal evolution of  $NO_3^-$  deposition to DML. While  $NO_3^-$  deposited to DML can be sourced from the sedimentation of polar stratospheric clouds in winter and we assume the atmospheric  $NO_3^-$  loading is uniform under the polar vortex, in spring and summer  $NO_3^-$  net

- 865 deposition is related to local photochemistry and subsequent post-depositional processing rather than primary NO3- sources. At this time, deposition of NO3- can be through the transport of re-emitted NO3- from the surface snow at low accumulation regions of the polar plateau, or NO3- produced *in situ* at DML in spring and summer. Year-round measurements of The annual cycle of atmospheric and/or skin layer NO3\* mass deposition concentration have previously been made-observed at DML (Figs. 5 and 6; Weller and Wagenbach (2007), Halley Station (Mulvaney et al., 1998; Jones et al., 2011), and Neumayer
- 870 Stations (Wagenbach et al., 1998), and the low snow accumulation site at Dome C (Fig. 1). These measurements describe the seasonal evolution of NO3- deposition to the skin layer from the atmosphere.(Weller and Wagenbach, 2007) indicates how much NO3- is deposited to the skin layer from the atmosphere (Figs. 5 and 6). Year-round NO3- mass concentrations have been measured in surface snow at the coastal sites of Halley (Mulvaney et al., 1998;Jones et al., 2011) and Neumayer Stations (Wagenbach et al., 1998), and the low snow accumulation site at Dome C (Fig. 1). An agreement with our simulated results 3.
- 875 at all Antarctic sites the highest atmospheric NO2+ mass concentrations are found during summer when the solar radiation is close to its annual maximum and NO2+ photolysis is strongest. The summer maximum at Dome C results from co-condensation of NO2+ (Book et al., 2016). This intense uplake in the skin layer in summer is driven by the strong temperature gradient in the upper few centimetres of the snow pack, highlighting that both physical (deposition; Bock et al., 2016). Chan et al., 2018) and

**Formatted:** Not Superscript/ Subscript**

**Formatted:** Not Superscript/ Subscript**

| - | Formatted: Highlight |
|---|----------------------|
|   |                      |
| 1 | Formatted: Highlight |
| + | Formatted: Highlight |
| + | Formatted: Highlight |
| 1 | Formatted: Highlight |

**shemical (NO, re-emission Fribland et al. (2015) processes explain the cycling of NO, between the air and snow. The lowest NO, mass concentrations in the skin layer are found in winter. Year round atmospheric NO, data at DML and Dome C shows atmospheric NO, is at a minimum in April to June and reaches a maximum in late November, slightly out of phase with skin layer NO, Wagenbach et al., 1998. Folland et al., 2013) (Figs. 1 and 6) (Figs. 1 and 6). The fact that the seasonality of simulated skin layer and atmospheric NO, at DML matches observations at other sites in Antarctica gives confidence in our TRANSITS model results (Fig. 6).**

- 885 We also observe variability on shorter timescales. While not yet observed elsewhere on the Antarctic continent, over the short intensive sampling period at DML we observe significant variability in NO3- mass concentrations and δ15N-NO3- values that resembles a diurnal cycle. Over 4 hours, the skin layer NO3- mass concentrations varied by 46 ng g-1, the skin layer δ15N-NO3- by 21 ‰, and the atmospheric δ15N-NO3- by 18 ‰. Other coastal studies have attributed daily variability to individual storm events (Mulvaney et al., 1998; Weller et al., 1999). We note that Tthe sampling duration in this study is too short to confirm
- 890 any diurnal patterns but it would be interesting to investigate this further in future work. We note that due to post-depositional processes (section 4.3) any short-term signals observed in the skin layer are unlikely to be preserved.

**4.2.43 Nitrate mass fluxes**

Our two NO3- mass flux scenarios in Fig. 5 highlight the importance of the skin layer in the air-snow transfer of NO3-. Like Bome C, the greatest deposition flux of NO3- is to the skin layer. The January dry deposition flux is greater than the annual mean flux estimated by Pasteris et al. (2014) and Weller and Wagenbach (2007) which is to be expected given the higher atmospheric NO3- mass concentrations in summer (Fig. 6). The wet deposition flux, calculated for the greater DML region by Pasteris et al. (2014), falls within our two scenarios. Furthermore, the simulated archived NO3- mass flux at DML of 210 pg m-2 s-1 for the base case and 480 pg m-2 s-1 for the 5 cm EFD case over predict s-the observed NO3- archived mass flux of 110

900 pg m-2 s-1 due to the higher simulated archived NO3- mass concentrations.- Interestingly, the simulated archived mass flux at Dome C (88 pg m-2 s-1) is lower than DML, yet the NO3- deposition flux to the skin layer in January at Dome C is similar to DML. We continue our discussion focusing on the recycling and redistribution of NO3- that occurs in the active skin layer emphasising its importance.

**4.3 Recycling and Post-depositional processes**

**905 4.3.1 Nitrate redistribution**

In corroboration with earlier work on the East Antarctic plateau,

Post-depositional loss and redistribution of  $NO_3^{-1}$  is the dominant control on snow pack mass concentrations and  $\delta^{15}N \cdot NO_3^{-1}$  isotopic signature on the Antarctic Plateau(Erbland et al., 2015). Recycling of  $NO_3^{-1}$  at the air snow interface comprises the following processes. Nitrate on the surface of a snow crystal can be lost from the snow pack (Dubowski et al., 2001), either by

**28**

**Formatted: Highlight Formatted: Highlight Formatted: Highlight Formatted: Highlight Formatted: Not Highlight Formatted: Not Highlight Formatted: Not Highlight Formatted: Not Highlight**

- - Formatted: Heading 3

- 910 UV photolysis or evaporation. UV photolysis produces NO, NO2 and HONO while only HNO2 can evaporate. Both of these processes produce reactive nitrogen that can be released from snow crystal into the interstitial air and rapidly transported out of the snow pack to the overlaying air via wind pumping (Zatko et al., 2013;Jones et al., 2000;Honrath et al., 1999;Jones et al., 2001). Here, NO2 is either oxidised to HNO3, which undergoes wet or dry deposition back to skin layer within a day, or transported away from the site (Davis et al., 2004a). If HNO3 is re-deposited on the snow skin layer, it is available for NO3- photolysis and/or evaporation again. Nitrate can be recycled multiple times between the boundary layer and the skin layer before it is buried in deeper layers of the snow pack. Photolysis and/or evaporation of NO3- and subsequent recycling between the air and snow alters the concentration and δ45N-NO3- that is ultimately preserved in ice cores. Nitrate recycling therefore redistributes NO3- from the active snow pack column to the skin layer via the atmosphere. Any locally produced NO2 that is transported away from the site of emission represents a loss of NO3- from the snow pack.
- 920 4.3.1 Evaporation

The desorption of HNO2 from the snow crystal reduces the NO2-concentration in the snow in coastal Antarctica (Mulvancy et a<mark>l., 1998), The evaporation of HNO2 is a two-step process, which involves the recombination of NO2-+H\*---> HNO2 followed by a phase change to HNO3 (gas-phase). First, theoretical estimates indicated that evaporation of HNO2 should preferentially remove 19N from the snow and release to the atmosphere leading to depletion in 845N-NO2--in the residual snow pack (Frey et</mark>

- 925 al., 2009). Furthermore, recent laboratory experiments showed that evaporation imposes a negligible fractionation of 844N-NO₂\* (Erbland et al., 2013;Shi et al., 2019). However, we find that the snow pack is enriched in 845N-NO₂\* relative to the atmosphere at DML (Figs. 3 and 6) and at Dome C (section 4.3.2). This fractionation observed in field studies cannot therefore be explained by evaporation, and must be attributed to different processes. It therefore follows that evaporation must be only a minor process in the redistribution of NO₂\* between atmosphere and the snow pack above the Antaretic plateau.
- 930 4.3.2 Photolysis

We focus our discussion on photolysis, which is the dominant process responsible for NO2--loss- and redistribution and associated δ15N-NO2--isotopic fractionation at low accumulation sites in Antarctica (Erbland et al., 2013;France et al., 2011). Nitrate photolysis occurs in the photoehemically active zone of the snow pack. Inown as the snow photic zone. Below this NO2- is buried. Nitrate photolysis in the active snow pack results in the production of NO2-leading to a reduction in the NO2-
 generation with depth in the snow pack (Fig. 4). In the photolysis induced fractionation of NO2-, aeN is preferentially removed first resulting in an enrichment of δ15N-NO2- in the snow pack. An individual snow layers undergo strong δ16N-NO2- enrichment as they are exposed to UV near the surface for the longest; late summer and autumn layers experience less δ15N-NO2- enrichment as they are exposed for less time before sunlight disappears at the start of polar-winter, during which new

940 precipitation buries existing snowfall.

29

| Formatted: Highlight |  |
|----------------------|--|
| Formatted: Highlight |  |
| Formatted: Highlight |  |
| -                    |  |
| Formatted: Highlight |  |
| Field Code Changed   |  |
|                      |  |

We provide five lines of evidence that photolysis is the dominant process for NO3- recycling and redistribution at DML.-we find clear evidence of NO3- redistribution via photolysis at DML, and confirmation of our hypothesis that UV-photolysis is driving NO3- recycling at DML. Firstly, the highly enriched  $\delta^{15}$ N-NO3- values of snow at DML and other Antarctic sites(-3 to 99 ‰), and the highly depleted atmospheric  $\delta^{15}$ N-NO3- values at DML (-20 to -49 ‰) are among the most extreme observed

- 945 on earth (Fig. S8S7;) Savarino et al. (2007), and cannot be explained by any known anthropogenic, marine or other natural sources. The δ15N-NOx-NOx--source signature of the main natural NOx sources (biomass burning, lightning, soil emissions-is lower; δ15N-NOx3 < 0 ‰) is lower than anthropogenic NOx sources, which generally have positive δ15N-NOx-- values (-13< δ15N-NOx3 < 13 ‰; e.g.) Hastings et al. (2013);Kendall et al. (2007);Hoering (1957) except in the case of vehicle and fertilised soil NOx emissions which have negative δ15N-NOx values (-60< δ15N-NO3- < 12 ‰; Miller et al. (2017);Yu and Elliott
- 950 (2017);Miller et al. (2018);Li and Wang (2008). However, a NO3- source contribution from fertilised soil NO8 emissions to Antarctica is thought to be minor (Lee et al., 2014). Such low atmospheric δ15N-NO3- values at DML show a marked difference to other mid-latitude tropospheric aerosol (-10< δ15N-NO3- <10 ‰; Freyer (1991). We acknowledge that stratospheric NO3± contributes to NO2- mass concentrations in snow in Antarctica. Although its isotopic signature is uncertain, estimates of stratospheric δ15N-NO3- are 19 ± 3 ‰ (Savarino et al., 2007), and fall well outside of atmospheric observations at DML.
- 955 TTherefore, δ15N-NO3- observations of aerosol, skin layer and snow pit at DML (-49< δ15N-NO3-<99 ‰) lie outside-of the range of natural and anthropogenic source end members (with the exception of anthropogenic emissions NO3 from vehicle and fertilised soil which can be ignored as a source to Antarctica), and thus cannot be explained by a mixture of sources (Fig. S7) or attributed. Thus, our measurements at DML are unrelated to seasonal variations in mid-low latitude NO3 sources e.g. increased springtime agricultural emissions, which has been observed in the mid-latitudes. The contribution of natural sources
- 960 to the Greenland snow pack δ14N NO3- signature has also been discarded (Geng et al., 2014;Geng et al., 2015). Furthermore, the negative atmospheric δ14N NO3- values at DML (-20 to -49 ‰) are extremely low. Such low atmospheric δ14N NO3- values have only been observed in Antarctica, and show marked difference to other mid-latitude tropospheric aerosol (-10< δ14N NO3- values) have only been observed in Antarctica, and show marked difference to other mid-latitude tropospheric aerosol (-10< δ14N NO3- values) have only been observed in Antarctica, and show marked difference to other mid-latitude tropospheric aerosol (-10< δ14N NO3- values) have only been observed in Antarctica, and show marked difference to other mid-latitude tropospheric aerosol (-10< δ14N NO3- contributes); Freyer (1991); Li and Wang (2008); Miller et al. (2018); Yu and Elliott (2017). We aeknowledge that stratospheric NO3- contributes to NO3- mass concentrations in snow in Antarctica. Although its isotopic signature is uncertain, estimates of stratospheric δ14N NO3- are 19 ± 3 ‰ (Savarino et al., 2007), and fall well outside of atmospheric observations at DML. The

unique snow and aerosol  $\delta^{15}$ N-NO3- signature\_of low accumulation Antarctic snow and aerosol is thus related to postdepositional processes specific to low accumulation sites in Antarctica.

Secondly, denitrification of the snow pack is seen through the δ15N-NO3- signature which evolves from the enriched snow pack (-3 to 99 ‰), to the skin layer (-22 to 3 ‰), to the depleted atmosphere (-49 to -20 ‰) corresponding to mass loss from the snow pack (Figs. 4 and S7). Denitrification causes the δ15N-NO3- of the residual snow pack NO3- to increase exponentially as

NO3- mass concentrations decrease. Thirdly, sensitivity analysis with TRANSITS, where photolysis is the driving process, is able to explain the observed snow pit

 $\delta^{15}$ N-NO3 variability when the e-folding depth is taken into account (section 4.5).

30

|      | ThFourthly, enrichment of $\delta^{15}$ N-NO 3 - is observed in the top 30 cm of the snowpack at DML indicating NO 3 - photolytic                                                                                                        |      |                                          |
|------|--------------------------------------------------------------------------------------------------------------------------------------------------------------------------------------------------------------------------------------------------------------------------------------|------|------------------------------------------|
| 975  | redistribution at DML in the photic zone of the snow pack (Fig. 7).i                                                                                                                                                                                                                 |      |                                          |
|      | Nitrate isotope enrichment takes place in the top 25 cm, which is consistent with an e-folding depth of 10 cm used in the base                                                                                                                                                       |      |                                          |
|      | $ \underline{ case \ scenario.} \ \underline{In \ the \ photic \ zone} \\ \underline{Here}, \ the \ \delta^{15} N-NO_3^- \ observations \ closely \ match \ the \ simulated \ \delta^{15} N-NO_3^- \ values \ \underline{from} \ begin{tabular}{lllllllllllllllllllllllllllllllllll$ |      |                                          |
|      | TRANSITS and show enrichment to this depth indicating NO3-photolytic redistribution at DML in the active photic zone of                                                                                                                                                              |      |                                          |
|      | the snow pack (Fig. 7). Below the photic zone, 8 14 N-NO 2 -values oscillate around a mean of -125 5. The mean values of the                                                                                                                                   |      | Formatted: Highlight                     |
| 980  | 845 N-NO3* observations are lower than the simulated values, which could be related to uncertainties in a number of factors, for                                                                                                                                                     |      |                                          |
|      | example: i) a shallower e-folding depth than modelled. During our field measurements, we derived a lower e-folding depth of                                                                                                                                                          |      |                                          |
|      | 2-5 cm (Fig. S1) at DML which could explain the lower enrichment in δ 15 N-NO3 + (section 4.5.2), ii) lower JNO3 + values (NO3 +                                                                                                         |      |                                          |
|      | photolysis rate), which are related to a lower e-folding depth, and would lead to less enrichment of 8 45 N-NO 2 - in the snow                                                                                                                      |      |                                          |
|      | pack, iii) higher atmospheric NO37-input, however-8 44 N-NO37 values are not sensitive to variable atmospheric NO37-mass                                                                                                                                                  |      |                                          |
| 985  | concentrations (Frbland et al., 2015), and/or iv) variable accumulation which would shift the oscillations to the correct depth                                                                                                                                                      |      | Formatted: Highlight                     |
|      | and lower the mean 8 45 N-NO 2 -values below the photic zone (section 4.5.1). The difference between the simulated and snow                                                                                                                                    |      | Formatted: Highlight                     |
|      | pit values shows that DML site is less sensitive to photolysis than we expected from TRANSITS modelling of 8 15 N-NO 5 + along                                                                                                                      |      |                                          |
|      | an accumulation gradient (Erbland et al., 2015).                                                                                                                                                                                                                                     |      | Formatted: Highlight                     |
|      | -Additionally, although not the focus of the study, denitrification causes the $\delta^{18}$ O-NO 3 values to increase in the residual NO 3 -                                                                                                                  | 5.7- | Formatted: Highlight                     |
| 990  | snow pack.                                                                                                                                                                                                                                                                           |      | Formatted: Font: 10 pt, Font color: Auto |
|      | Thirdly, the application of TRANSITS to DML observations show that our observed atmospheric, skin layer and snow depth                                                                                                                                                               |      |                                          |
|      | profiles of 814N-NOg- are similar to the simulated values where photolysis is the driving process (Figs. 6-7) Sensitivity analysis                                                                                                                                                   |      |                                          |
|      | with TRANSITS is able to explain the observed snow pit 8

---

## Author Response (AR2)

**Authors' point-by-point response to ACP MS No.: acp-2019-669**

We thank the anonymous reviewer for helpful comments and particularly welcome the suggestions to streamline the paper. We have combined the results and discussion, and further compared the Zatko et al. (2016) modelling results to Dome C and DML observations. In the text below, we outline our responses in blue. Line numbers refer to the revised manuscript.

Summary:

Based upon the feedback of the reviewers and the authors' response, the authors have done a much better job of explaining their data and interpretation. The re-organization of the paper was necessary to make this something that is publishable in the peer-reviewed literature. There are some suggestions below for streamlining the paper – it is still overly long and repetitive. The authors have also added a good deal of new interpretation/discussion since the previous version, which brings in new caveats. Overall, the dataset is novel and interesting and expands our understanding of the use of the isotopic composition of nitrate for interpreting snow and ice results in Antarctica.

General Comments:

I think the authors should consider combining results and discussion and streamlining the manuscript. The results and discussion are still very repetitive and at times statements come up in the results that are not then fully discussed for 8 more pages!

Thank you for the suggestion. We have combined the results and discussion as suggested.

The authors should also consider synthesizing a bit more regarding the Zatko et al 2016 results. The GEOS-Chem model is taking into account the loss, transport and recycling of nitrate/NOx. So it seems relevant to see whether the features predicted by the model are represented in the Dome C and DML comparison.

We thank the reviewer for this suggestion. We have compared the model results by Zatko et al. (2016) to our DML results in lines 723-727 (transport of locally produced NOx), lines 590-591 (recycling factor), lines 614-619 (e-folding depth), and lines 700-703 (archival time). In general, the spatial trends predicted by the model are represented at Dome C and DML, in particular the transport of snow sourced nitrate. An exception is the spatial pattern of the e-folding depth, where we observed a lower e-folding at DML than Dome C opposite to what the model predicts. The model overestimates the archival time and recycling factor of nitrate at DML, and we suggest this is due to the lower observed e-folding depth than modelled. We had added text in the discussion in lines 723-727 as follows:

Additional text: "Although the spatial trends predicted by the modelling of Zatko et al. (2016) are represented at Dome C and DML, an exception is the spatial pattern of the e-folding depth, where we observed a lower e-folding at DML than Dome C opposite to what the model predicts. At DML, the model overestimates the archival time and recycling factor of $NO_3^-$, and we suggest this is due to the lower observed e-folding depth than modelled."

Specific comments:

Abstract:

Rephrase the first sentence. This study is specific to low accumulation sites in Antarctica. The sensitivity to UV and TCO is not universal as this has not been shown to be a strong predictor anywhere else but the East Antarctic ice sheet (and not at all at the North Pole).

Sentence modified in lines 10-12 as follows:

Modified text: "The nitrogen stable isotopic composition in nitrate ($\delta^{15}$N-NO$_3^-$) measured in ice cores from low snow accumulation regions in East Antarctica has the potential to provide constraints on past ultraviolet (UV) radiation and thereby total column ozone (TCO) due to the sensitivity of nitrate (NO$_3^-$) photolysis to UV radiation."

Line 25-28: Delete the sentence starting with "Secondly,…." And instead begin the next sentence with Based on the TRANSITS model, we find that NO3- is…"

Done.

Also this sentence should report the 2 times recycling as the average for the skin layer.

Done.

Introduction:

Lines 53-56: The phrasing here regarding nitrate (NOx) sources is a bit strange. Later (much later) in the manuscript, marine air masses are deemed important, yet couched here as if they are minor. Further, the model results of Lee et al. are not really explained. Rephrase here to indicate that all of these sources are shown to be important. In fact, Lee et al can explain Antarctic nitrate concentration at all times of the year except the seasonal increase due to PSC sedimentation b/c this is not included in the model. So the tropospheric transport COULD be important and could be more important at other times of the year than spring.

Organic nitrate from marine sources has been shown to be a minor source of total nitrate in Antarctica (Jacobi et al., 2000;Jones et al., 1999;Beyersdorf et al., 2010). Section modified in lines 52-56 as follows:

Modified text: "Primary sources of reactive nitrogen species to the Antarctic lower atmosphere and snow pack include the sedimentation of polar stratospheric clouds (PSC) in late winter (Savarino et al., 2007), in addition to tropospheric transport of inorganic NO$_3^-$ from lightning, biomass burning and soil emissions (Lee et al., 2014) and, to a minor extent, advection of oceanic organic nitrate such as methyl nitrate (CH$_3$NO$_3$) and peroxyacyl nitrates (PAN) (Jacobi et al., 2000;Jones et al., 1999;Beyersdorf et al., 2010)."

Lines 95-96: It does not at all follow the previous text that evaporation of nitrate is negligible. This is further explained, at too much length, in the discussion section. Here the loss of nitrate from the surface and its impact on the isotopic composition of nitrate is being discussed. The evaporation part needs to be rephrased to make clear that isotopically evaporation is negligible b/c of the very large fractionation associated with photolytic loss. And/or introduce the fact that based on the Shi et al study conditions at DML would warrant that evaporation should have a negligible effect b/c the isotopes effects reported in Shi et al are negligible.

Sentence modified in lines 95-97 as follows:

Modified text: "As the fractionation associated with photolytic loss is large and the isotope effects of evaporation are negligible (Shi et al., 2019), it follows that evaporation of $NO_3^-$ is negligible on high-elevation Antarctic sites (Erbland et al., 2013;Shi et al., 2019)."

Line 118: The nitrate is not archived necessarily in ice. The e-folding depth shown in this study is very shallow and clearly in the firn layer. Rephrease here.

This sentence refers to nitrate recycling which includes a number of processes e.g. photolytic loss in the photic zone of the snow pack and has been defined by Erbland et al. (2015). We have modified in sentence to "…in firn/ice cores" in line 124.

Lines 120-123: This completely disregards the previous paragraph on nitrate sources at lines 53-56. Here only the stratosphere is being considered as a source.

This sentence describes the interpretation of the annual cycle in atmosphere and skin layer nitrate observations by the authors who collected those data at Dome C. We also include tropospheric nitrate transport throughout year in lines 99-100 as follows:

Modified text: "Additional to year-round troposphere transport of $NO_3^-$ (Lee et al., 2014), in the early winter, the stratosphere undergoes denitrification via formation of PSC."

Line 128-132: This is repetitive. In the Introduction and in the Discussion section the authors flip flop between the importance of source and the importance of post-depositional processing and re-explain post-depositional processing multiple times. The primary deposition signal is only erased if there is post-depositional recycling. If there is only loss, then based upon the modeling results presented in this paper, the loss can be accounted for and a primary signal could be detected. But with lots of loss and recycling the original signal is clearly too overprinted.

The text in these lines is the first mention of the dependence of nitrate concentration and isotopes on the snow accumulation rate. We have edited the text to avoid repetition in lines 125-128 as follows:

Modified text: "The $NO_3^-$ signal in the snow pack is dependent on the snow accumulation rate. At sites with very low snow accumulation rates (i.e., Dome C: 2.5 - 3 cm yr$^{-1}$), $NO_3^-$ is not preserved in the snow pack because snow layers remain close to the surface and in contact with the overlaying atmosphere for a relatively long time enhancing the effect of post-depositional processes which erase the source signature of $\delta^{15}N-NO_3^-$."

Further, the nitrate isotope signal at DML and low accumulation sites reflects post-depositional processing rather than nitrate sources which we emphasise in section 4.3. We believe the more streamlined version, with results and discussion combined, reduces any confusion between sources and processes.

Line 131: The "Therefore, photolysis induced NO3- loss and…." Does not follow from the previous sentence.

Following the comment, this section has been modified above in lines 125-128 as follows:

Modified text: "The $NO_3^-$ signal in the snow pack is dependent on the snow accumulation rate. At sites with very low snow accumulation rates (i.e., Dome C: 2.5 - 3 cm yr$^{-1}$), $NO_3^-$ is not preserved in the snow pack because snow layers remain close to the surface and in

contact with the overlaying atmosphere for a relatively long time enhancing the effect of post-depositional processes which erase the source signature of $\delta^{15}$N-NO$_3^-$.”

Throughout the next two paragraphs I still found the text repetitive and much of the material is really more important for the discussion and is all repeated there.

The second to last paragraph of the introduction is about the influence of snow accumulation on the nitrate signal in snow and ice cores. We believe this is important information in the introduction section as the UV proxy is based on low snow accumulation Antarctic sites. We have condensed the paragraph to avoid repetition. While the last paragraph sets the objectives of the study. We have condensed background information about the ISOL-ICE project from the methods and moved it into the introduction.

Methods:

The beginning of section 2 is not Methods, it's really an overview of the project's purpose and also discusses some results. It might make more sense to move this into introduction and reduce the last two paragraphs as suggested as above. Further, it is strange that Figure 2 is introduced before Figure 1. Figure 1 is really results and should be introduced in that section.

The beginning of section 2 outlines the aims of the project and the data used in this manuscript as some data from Dome C is published and some is new. We have moved this section 2 into the last paragraph of the introduction as suggested by the reviewer.

Figure 1 is introduced first in line 99 (before Figure 2 in line 169) to help describe introductory material, i.e., the annual cycle of nitrate deposition to Dome C. It comprises published data in addition to new data which extends the Dome C record. Therefore, we refer to Figure 1 first in the introduction.

Similarly, lines 230-234 and lines 330-335 are results not methods.

Lines 221-224 describe the sampling resolution, sampling dates and refer to papers for the sampling methods for the new atmospheric and skin layer samples from Dome C. We believe section "2.2. Snow and aerosol sampling" in the methods is the best place for this information.

Lines 362-367 describe how we adjusted the e-folding depth in the TRANSITS model. As this parameter cannot simply be changed, we recalculated the J($^{14/15}$NO$_3^-$) profiles for DML which has not been done before for TRANISTS sensitivity runs. We believe the methods are the best place to describe our approach.

Line 337: This needs to be reworded to make clear that recycling occurs before the nitrate is archived.

Line 369 reworded as follows:

Modified text: "TRANSITS calculates the average number of recyclings before NO$_3^-$ is archived…"

Results:

Line 377: This is an overly broad statement that the annual cycle is similar "across Antarctica". It would be useful here to state explicitly what locations are being compared

to… this does not appear true for the coastal work nor for site on the West Antarctic Ice sheet. So this should be more specific rather than an overreach. Furthermore, this is introduced here as a fact and then re-discussed much later in the discussion in more detail.

Text has been modified in lines 546-548 as follows:

Modified text: "The annual cycle is consistent both i) spatially across a vast area of Antarctica, i.e., South Pole, Dome C, Halley Station, Neumayer Station (McCabe et al., 2007;Wolff et al., 2008;Erbland et al., 2013;Frey et al., 2009;Wagenbach et al., 1998)…"

Sections 3.5, 3.6 and part of 3.8 are really methods. Here you are introducing the method of calculation then doing the calculation and stating a result. But most of this belongs in methods as an explanation of the approach taken and then the results and discussion and discuss the results of these calculations referring back to the equations in the methods section. And section 3.7 is really discussion not results since you cannot explain the result and instead state a hypothesis that should be part of discussion rather than then referring to a much later section.

Equations in sections 3.2, 3.5, 3.6 and 3.8 have been moved to the methods. The text in section 3.7 has been moved to the discussion (section 4.3.2).

Most of section 3.8 is focused on results for Dome C and this is confusing since the purpose of the paper is to really understand air to snow transfer of nitrate at DML and it's potential for ice core nitrate interpretation at DML. The simple calculated flux is a huge overestimate; then the TRANSIT model is a large underestimate. Why are we spending so much time here on this when there actually is no real constraint on this in the region?

We refer the reviewer to the first round of reviews. Anonymous Referee #2 asked us to elaborate on the poorly constrained quantum yield of nitrate photolysis in snow which yields a very high NOx flux compared to TRANISTS. We have spent time discussing this in response to the comment made by referee #2.

Discussion:

The title of 4.1 is confusing. How is this a validation of results? Throughout the results section the DML results and additional Dome C results are "confirmed" by comparison with past results at Dome C. Here it is really the TRANSITS model that is being evaluated.

The title of section 4.1 has been renamed as "Evaluation of TRANSITS model results"

Lines 565-568: Notably, have you considered a volume/mass effect here? Presumably the deposition as diamond dust or hoar frost is occurring in a much more limited volume of water equivalent then the collections of the surface layer. Were the masses of each sample collected recorded? Could the nitrate concentrations be mass-weighted and compared to explain the difference in results? In fact at line 585 it is suggested that possible wet deposition of nitrate can explain the lower mass fractions in the skin layer? Why isn't this discussion tied together?

We agree with the reviewer than mass weighted concentration would be helpful to compare nitrate concentrations in hoar frost and skin layer samples. While the mass of DML samples was measured, we do not have masses for the hoar frost samples from Dome C. We refer the reviewer to lines 518-521 where we summarise that it is difficult to separate out the effect on

nitrate mass concentrations from nitric acid scavenging diamond dust/hoar from versus precipitation diluting the nitrate mass concentrations.

Section 4.2.1 is repetitive – this has all already been covered at least twice in the manuscript so far.

In section 4.2.1 we use information about the airmass source region, precipitation and dry deposition to explain the nitrate mass concentrations in the skin layer at DML during the ISOL-ICE campaign. The airmass fetch region and RACMO precipitation data is new information and has not been mentioned previously. By combining the results and discussion sections, repetition has been avoided.

Line 577: How is station contamination completely ruled out? State clearly that the station was ALWAYS downwind if that is the case.

Text modified in lines 510-512 as follows:

Modified text: "The second excursion occurs during a few short periods when the wind direction switches upwind of the station however, there are no spikes in the $NO_3^-$ mass concentration or a change in the $\delta^{15}N$-$NO_3^-$ signature and so we can also rule out any downwind contamination from the station."

Line 605: But how does this compare to Zatko et al 2016 model results which do account for transport of snow-sourced emissions and re-deposition?

We have added the following text to lines 541-543 as follows:

Modified text: "Interestingly, model results from Zatko et al. (2016), which account for transport of snow-sourced $NO_3^-$ emissions and deposition, show that the deposition of recycled $NO_3^-$ to snow is lowest on the East Antarctic Plateau including the high-elevation DML region."

Lines 637-650: This section on sources is oddly worded and does not reflect the most recent literature. It also is really not necessary. The paper has already established the importance and large isotopic imprint of post-depositional loss and recycling. It could easily be stated here what the range in measured sources are, or make a table in the supplement of the reported ranges and how they do not at all cover the observed range at DML. Fibiger et al. (ES&T, 2016) contains positive values for some types of biomass burning so stating that it is a negative d15N source is incorrect. And the large ranges reported here do not reflect the fact that the variability in those ranges can be explained. For instance, the d15N-NOx from biomass burning is dependent upon the d15N-biomass. The more recent vehicle measurements suggest that older measurements are outdated and the range for plumes (versus tailpipes) is much more narrowly defined. Overall, this paragraph is the same arguments made in other papers and could be made more simply earlier on in the manuscript.

We have modified the text on lines 560-569 as follows:

Modified text: "Firstly, the highly enriched $\delta^{15}N$-$NO_3^-$ values of snow at DML (-3 to 99 ‰), and the highly depleted atmospheric $\delta^{15}N$-$NO_3^-$ values at DML (-20 to -49 ‰) are unique to post-depositional processes at low accumulation sites in Antarctica (Fig. S7) and lie outside the range of known anthropogenic, marine or other natural source end members (e.g.

Hastings et al., 2013;Kendall et al., 2007;Hoering, 1957;Miller et al., 2017;Yu and Elliott, 2017;Miller et al., 2018;Li and Wang, 2008;Freyer, 1991;Savarino et al., 2007)."

The formatting in section 4.3.2 differs from all of the other sections (i.e. why is there 1., 2., etc. sections?).

We refer the reviewer to the first round of reviews. The formatting was modified in response to a comment made by Anonymous Referee #2.

Section 5 here is another opportunity to consider the Zatko et al. 2016 model predictions.

In the discussion, we have compared our results to those predicted by Zatko et al. (2016) in terms of:

- e-folding depth in lines 614-619 as follows:

Additional text: "Spatial patterns of modelled e-folding depths across Antarctica predict shallower e-folding depths in regions of relatively high black carbon concentrations located on the plateau in Antarctica (Zatko et al., 2016). In contrast, we observe a opposite pattern of higher black carbon concentrations and a deeper e-folding depth at Dome C compared to a shallower e-folding depth at DML. Therefore, the observed shallower e-folding depth at DML appears unrelated to black carbon concentrations as the modelling by Zatko et al. (2016) predicts a greater e-folding depth in the DML region where black carbon concentrations are lower."

- Recycling factor in lines 590-591 as follows:

Additional text: "Although these findings are consistent with spatial patterns of $NO_3^-$ recycling factors across Antarctica reported by Zatko et al. (2016), predictions for the DML region are almost double our estimates."

- Archival time in lines 700-703 as follows:

Additional text: "The greater residence time of $NO_3^-$ in the photic zone at Dome C relative to DML is consistent with modelled spatial patterns of the lifetime of $NO_3^-$ burial across Antarctica where $NO_3^-$ remains in the photic zone for the longest in the lower snow accumulation regions (Zatko et al., 2016). The model predicts $NO_3^-$ archival time to be 3-4 years at DML which is considerably greater than our estimates."

- More generally in lines 723-727 as follows:

Additional text: "Although the spatial trends predicted by the modelling of Zatko et al. (2016) are represented at Dome C and DML, an exception is the spatial pattern of the e-folding depth, where we observed a lower e-folding at DML than Dome C opposite to what the model predicts. At DML, the model overestimates the archival time and recycling factor of $NO_3^-$, and we suggest this is due to the lower observed e-folding depth than modelled."

Section 4.4 – why and what is the Weller approach as opposed to what is already been done using TRANSITS?

The Weller et al. (2004) approach does not take the high skin layer nitrate concentrations into account. We have modified the text in lines 672-673 as follows:

Modified text: "By modifying the approach of Weller et al. (2004) by taking the high observed skin layer $NO_3^-$ mass concentrations into account (average of 230 ng g$^{-1}$ in January for DML), we calculate…"

Section 4.5 – is is unclear to me why McCabe et al's work at South Pole is not be discussed here? This is the only work I am aware of that makes a quantitative link between TCO and d15N of nitrate.

McCabe et al. (2007) propose $\Delta^{17}O\text{-}NO_3^-$ at South Pole can be used as a proxy for stratospheric ozone. However, post depositional processes related to $\Delta^{17}O\text{-}NO_3^-$ need to be quantificatied to fully understand the sources and processes responsible for deposited and archiving $\Delta^{17}O\text{-}NO_3^-$ signature in Antarctica. We have discussed McCabe et al. (2007)'s work in lines 803-804:

Additional text: "In addition, the oxygen isotopic composition of $NO_3^-$ ($\Delta^{17}O\text{-}NO_3^-$) has been proposed as a proxy for stratospheric ozone at South Pole (McCabe et al., 2007), however post depositional processes related to $\Delta^{17}O\text{-}NO_3^-$ need to be quantified to fully understand the sources and processes responsible for depositing and archiving the $\Delta^{17}O\text{-}NO_3^-$ signature in Antarctica."

Section 4.5.1 and 4.5.2 are totally repetitive of earlier discussion and results. I suggest you combine results and discussion and eliminate the repeats.

Done.

Line 813-814: Variation in snow accumulation is clearly important, and you model this and the sensitivity of d15N to it. But you suggest that TRANSITS can explain and capture this so the statement here that this needs to be carefully accounted for seems counter to the fact that the model can reproduce the d15N so well (as suggested by the authors).

TRANSITS can explain the sensitivity of $\delta^{15}N\text{-}NO_3^-$ to the snow accumulation rate when the observed snow accumulation is used in the model. Therefore, it is necessary to know and account for past changes in the snow accumulation rate from ice cores to interpret the nitrate isotope signal at DML.

Line 839-841: Why is Domine's work on nitrate diffusion not considered here? (Domine et al., Atmos. Chem. Phys., 8, 171–208, 2008 www.atmos-chem-phys.net/8/171/2008/). Based upon the temperature and accumulation rate at DML, diffusion should be relatively straightforward to constrain rather than raise as a big open question.

We have added text in lines 811-813 as follows:

[revised manuscript text omitted]